# SINGLE-LEVEL ADVERSARIAL DATA SYNTHESIS BASED ON NEURAL TANGENT KERNELS

## ABSTRACT

Generative adversarial networks (GANs) have achieved impressive performance in data synthesis and have driven the development of many applications. However, GANs are known to be hard to train due to their bilevel objective, which leads to the problems of convergence, mode collapse, and gradient vanishing. In this paper, we propose a new generative model called the generative adversarial NTK (GA-NTK) that has a single-level objective. The GA-NTK keeps the spirit of adversarial learning (which helps generate plausible data) while avoiding the training difficulties of GANs. This is done by modeling the discriminator as a Gaussian process with a neural tangent kernel (NTK-GP) whose training dynamics can be completely described by a closed-form formula. We analyze the convergence behavior of GA-NTK trained by gradient descent and give some sufficient conditions for convergence. We also conduct extensive experiments to study the advantages and limitations of GA-NTK and propose some techniques that make GA-NTK more practical.[1]

## 1 INTRODUCTION

Generative adversarial networks (GANs) (Goodfellow et al., 2014; Radford et al., 2016), a branch of deep generative models based on adversarial learning, have received much attention due to their novel problem formulation and impressive performance in data synthesis. Variants of GANs have also driven recent developments of many applications, such as super-resolution (Ledig et al., 2017), image inpainting (Xu et al., 2014), and video generation (Vondrick et al., 2016).

A GANs framework consists of a discriminator network $\mathcal{D}$ and a generator network $\mathcal{G}$ parametrized by $\boldsymbol{\theta}_\mathcal{D}$ and $\boldsymbol{\theta}_\mathcal{G}$, respectively. Given a $d$-dimensional data distribution $\mathcal{P}_{\text{data}}$ and a $c$-dimensional noise distribution $\mathcal{P}_{\text{noise}}$, the generator $\mathcal{G}$ maps a random noise $\boldsymbol{z} \in \mathbb{R}^c$ to a point $\mathcal{G}(\boldsymbol{z}) \in \mathbb{R}^d$ in the data space, while the discriminator $\mathcal{D}$ takes a point $\boldsymbol{x}' \in \mathbb{R}^d$ as the input and tells whether $\boldsymbol{x}'$ is real or fake, i.e., $\mathcal{D}(\boldsymbol{x}') = 1$ if $\boldsymbol{x}' \sim \mathcal{P}_{\text{data}}$ and $\mathcal{D}(\boldsymbol{x}') = 0$ if $\boldsymbol{x}' \sim \mathcal{P}_{\text{gen}}$, where $\mathcal{P}_{\text{gen}}$ is the distribution of $\mathcal{G}(\boldsymbol{z})$ and $\boldsymbol{z} \sim \mathcal{P}_{\text{noise}}$. The objective of GANs is typically formulated as a bilevel optimization problem:

$$\arg\min_{\boldsymbol{\theta}_\mathcal{G}} \max_{\boldsymbol{\theta}_\mathcal{D}} \mathbb{E}_{\boldsymbol{x} \sim \mathcal{P}_{\text{data}}}[\log \mathcal{D}(\boldsymbol{x})] + \mathbb{E}_{\boldsymbol{z} \sim \mathcal{P}_{\text{noise}}}[\log(1 - \mathcal{D}(\mathcal{G}(\boldsymbol{z})))]. \tag{1}$$

The discriminator $\mathcal{D}$ and generator $\mathcal{G}$ aim to break each other through the inner $\max$ and outer $\min$ objectives, respectively. The studies by Goodfellow et al. (2014); Radford et al. (2016) show that this adversarial formulation can lead to a better generator that produces plausible data points/images.

However, GANs are known to be hard to train due to the following issues (Goodfellow, 2016). **Failure to converge.** In practice, Eq. (1) is usually only approximately solved by an alternating first-order method such as the alternating stochastic gradient descent (SGD). The alternating updates for $\boldsymbol{\theta}_\mathcal{D}$ and $\boldsymbol{\theta}_\mathcal{G}$ may cancel each other's progress. During each alternating training step, it is also tricky to balance the number of SGD updates for $\boldsymbol{\theta}_\mathcal{D}$ and that for $\boldsymbol{\theta}_\mathcal{G}$, as a too small or large number for $\boldsymbol{\theta}_\mathcal{D}$ leads to low-quality gradients for $\boldsymbol{\theta}_\mathcal{G}$. **Mode collapse.** The alternating SGD is attracted by stationary points and therefore is not good at distinguishing between a $\min_{\boldsymbol{\theta}_\mathcal{G}} \max_{\boldsymbol{\theta}_\mathcal{D}}$ problem and a $\max_{\boldsymbol{\theta}_\mathcal{D}} \min_{\boldsymbol{\theta}_\mathcal{G}}$ problem. When the solution to the latter is returned, the generator tends to always produce the points at modes that best deceive the discriminator, making $\mathcal{P}_{\text{gen}}$ of low

---

[1]Our code is available on GitHub at `https://github.com/ga-ntk/ga-ntk`.

diversity.[2] **Vanishing gradients.** At the beginning of a training process, the finite real and fake training data may not overlap with each other in the data space, and thus the discriminator may be able to perfectly separate the real from fake data. Given the cross-entropy loss (or more generally, any $f$-divergence measure (Rényi et al., 1961) between $\mathcal{P}_{\text{data}}$ and $\mathcal{P}_{\text{gen}}$), the value of the discriminator becomes saturated on both sides of the decision boundary, resulting in zero gradients for $\boldsymbol{\theta}_{\mathcal{G}}$.

In this paper, we argue that the above issues are rooted in the modeling of $\mathcal{D}$. In most existing variants of GANs, the discriminator is a deep neural network with explicit weights $\boldsymbol{\theta}_{\mathcal{D}}$. Under gradient descent, the gradients of $\boldsymbol{\theta}_{\mathcal{G}}$ in Eq. (1) cannot be back-propagated through the inner $\max_{\boldsymbol{\theta}_{\mathcal{D}}}$ problem because otherwise it requires the computation of high-order derivatives of $\boldsymbol{\theta}_{\mathcal{D}}$. This motivates the use of alternating SGD, which in turn causes the convergence issues and mode collapse. Furthermore, the $\mathcal{D}$ is a single network whose particularity may cause a catastrophic effect, such as the vanishing gradients, during training.

We instead model the discriminator $\mathcal{D}$ as a Gaussian process whose mean and covariance are governed by a kernel function called the neural tangent kernel (NTK-GP) (Jacot et al., 2018; Lee et al., 2019; Chizat et al., 2019). The $\mathcal{D}$ approximates an infinite ensemble of infinitely wide neural networks in a nonparametric manner and has no explicit weights. In particular, its training dynamics can be completely described by a closed-form formula. This allows us to simplify adversarial data synthesis into a single-level optimization problem, which we call the generative adversarial NTK (GA-NTK). Moreover, since $\mathcal{D}$ is an infinite ensemble of networks, the particularity of a single element network does not drastically change the training process. This makes GA-NTK less prone to vanishing gradients and stabilizes training even when an $f$-divergence measure between $\mathcal{P}_{\text{data}}$ and $\mathcal{P}_{\text{gen}}$ is used as the loss of $\mathcal{D}$. The following summarizes our contributions:

- We propose a single-level optimization method, named GA-NTK, for adversarial data synthesis. It can be solved by ordinary gradient descent, avoiding the difficulties of bi-level optimization in GANs.

- We prove the convergence of GA-NTK training under mild conditions. We also show that $\mathcal{D}$ being an infinite ensemble of networks can provide smooth gradients for $\mathcal{G}$, which stabilizes GA-NTK training and helps fight vanishing gradients.

- We propose some practical techniques to reduce the memory consumption of GA-NTK during training and improve the quality of images synthesized by GA-NTK.

- We conduct extensive experiments on real-world datasets to study the advantages and limitations of GA-NTK. In particular, we find that GA-NTK has much lower sample complexity as compared to GANs, and the presence of a generator is not necessary to generate images under the adversarial setting.

Note that the goal of this paper is not to replace existing GANs nor advance the state-of-the-art performance, but to show that adversarial data synthesis can be done via a single-level modeling. Our work has implications for future research. In particular, the low sample complexity makes GA-NTK suitable for applications, such as medical imaging, where data are personalized or not easily collectible. In addition, GA-NTK bridges the gap between kernel methods and adversarial data/image synthesis and thus enables future studies on the relationship between kernels and generated data.

## 2 RELATED WORK

### 2.1 GANS AND IMPROVEMENTS

Goodfellow et al. (2014) proposes GANs and gives a theoretical convergence guarantee in the function space. However, in practice, one can only optimize the generator and discriminator in Eq. (1) in the parameter/weight space. Many techniques have been proposed to make the bilevel optimization easier. **Failure to convergence.** To solve this problem, studies devise new training algorithms for GANs (Nagarajan & Kolter, 2017; Daskalakis et al., 2018) or more general minimax problems (Thekumparampil et al., 2019; Mokhtari et al., 2020). But recent works by Mescheder et al. (2018); Farnia & Ozdaglar (2020) show that there may not be a Nash equilibrium solution in GANs. **Mode**

---

[2] Mode collapse can be caused by other reasons, such as the structure of $\mathcal{G}$. This paper only solves the problem due to alternating SGD.

**collapse.** Metz et al. (2017) alleviates this issue by back-propagating the computation of $\boldsymbol{\theta}_{\mathcal{G}}$ through the discriminators trained with several steps to strengthen the $\min_{\boldsymbol{\theta}_{\mathcal{G}}} \max_{\boldsymbol{\theta}_{\mathcal{D}}}$ property. Other works mitigate mode collapse by diversifying the modes of $\mathcal{D}$ through regularization (Che et al., 2017; Mao et al., 2019), modeling $\mathcal{D}$ as an ensemble of multiple neural networks (Durugkar et al., 2017; Ghosh et al., 2018), or using additional auxiliary networks(Srivastava et al., 2017; Bang & Shim, 2021; Li et al., 2021). **Vanishing gradients.** Mao et al. (2017) tries to solve this problem by using the Pearson $\chi^2$-divergence between $\mathcal{P}_{\text{data}}$ and $\mathcal{P}_{\text{gen}}$ as the loss to penalize data points that are far away from the decision boundary. However, it still suffers from vanishing gradients as any $f$-divergence measure, including the cross-entropy loss and Pearson $\chi^2$-divergence, cannot measure the difference between disjoint distributions (Sajjadi et al., 2018). Later studies replace the loss with either the Wasserstein distance (Arjovsky et al., 2017; Gulrajani et al., 2017) or maximum mean discrepancy (Gretton et al., 2012; Li et al., 2015; 2017) that can measure the divergence of disjoint $\mathcal{P}_{\text{data}}$ and $\mathcal{P}_{\text{gen}}$. In addition, the works by Miyato et al. (2018); Qi (2020) aim to constrain the Lipschitz continuity of the discriminator to prevent its value from being saturated.

Despite that many efforts have been made to improve the training of GANs, most existing approaches address only one or two issues at a time with different assumptions, and in the meanwhile, they introduce new hyperparameters or side effects. For example, in the Wasserstein GANs (Arjovsky et al., 2017; Gulrajani et al., 2017) mentioned above, efficient computation of Wasserstein distance requires the discriminator to be Lipschitz continuous. However, realizing Lipschitz continuity introduces new hyperparameters and could limit the expressiveness of the discriminator (Anil et al., 2019). Until now, training GANs is still not an easy task because one has to 1) tune many hyperparameters and 2) strike a balance between the benefits and costs of different training techniques to generate satisfactory data points/images.

## 2.2 GAUSSIAN PROCESSES AND NEURAL TANGENT KERNELS

Consider an infinite ensemble of infinitely wide networks that use the mean square error (MSE) as the loss and are trained by gradient descent. Recent developments in deep learning theory show that the prediction of the ensemble can be approximated by a special instance of Gaussian process called NTK-GP (Jacot et al., 2018; Lee et al., 2019; Chizat et al., 2019). The NTK-GP is a Bayesian method, so it outputs a distribution of possible values for an input point. The mean and covariance of the NTK-GP prediction are governed by a kernel function $k(\cdot, \cdot)$ called the neural tangent kernel (NTK). Given two data points $\boldsymbol{x}^i$ and $\boldsymbol{x}^j$, the $k(\boldsymbol{x}^i, \boldsymbol{x}^j)$ represents the similarity score of the two points in a kernel space, which is fixed once the hyperparameters of the initial weights, activation function, and architecture of the networks in the target ensemble are determined.

Here, we focus on the mean prediction of NTK-GP as it is relevant to our study. Consider a supervised learning task given $\mathbb{D}^n = (\boldsymbol{X}^n \in \mathbb{R}^{n \times d}, \boldsymbol{Y}^n \in \mathbb{R}^{n \times c})$ as the training set, where there are $n$ examples and each example consists of a pair of $d$-dimensional input and $c$-dimensional output. Let $\boldsymbol{K}^{n,n} \in \mathbb{R}^{n \times n}$ be the kernel matrix for $\boldsymbol{X}^n$, i.e., $K_{i,j}^{n,n} = k(\boldsymbol{X}_{i,:}^n, \boldsymbol{X}_{j,:}^n)$. Then, at time step $t$ during gradient descent, the mean prediction of NTK-GP for $\boldsymbol{X}^n$ evolve as

$$(\boldsymbol{I}^n - e^{-\eta \boldsymbol{K}^{n,n} t}) \boldsymbol{Y}^n \in \mathbb{R}^{n \times c}, \tag{2}$$

where $\boldsymbol{I}^n \in \mathbb{R}^{n \times n}$ is an identity matrix and $\eta$ is a sufficiently small learning rate (Jacot et al., 2018; Lee et al., 2019).

The NTK used in Eq. (2) can be extended to support different network architectures, including convolutional neural networks (CNNs) (Arora et al., 2019; Novak et al., 2019b), recurrent neural networks (RNNs) (Alemohammad et al., 2021; Yang, 2019b), networks with the attention mechanism (Hron et al., 2020), and other architectures (Yang, 2019b; Arora et al., 2019). Furthermore, studies (Novak et al., 2019a; Lee et al., 2020; Arora et al., 2020; Geifman et al., 2020) show that NTK-GPs perform similarly to their finite-width counterparts (neural networks) in many situations and sometimes even better on small-data tasks.

A recent study by Franceschi et al. (2021) analyzes the behavior of GANs from the NTK perspective by taking into account the alternating optimization. It shows that, in theory, the discriminator can provide a well-defined gradient flow for the generator, which is opposite to previous theoretical interpretations (Arjovsky & Bottou, 2017). Our work, on the other hand, focuses on adversarial data

synthesis *without* alternating optimization.[3] We make contributions in this direction by (1) formally proving the convergence of the proposed single-level optimization, (2) showing that a generator network is not necessary to generate plausible images (although it might be desirable), and (3) proposing the batch-wise and multi-resolutional extensions that respectively improve the memory efficiency of training and global coherency of generated image patterns.

## 3 GA-NTK

We present a new adversarial data synthesis method, called the generative adversarial NTK (GA-NTK), based on the NTK theory (Jacot et al., 2018; Lee et al., 2019; Chizat et al., 2019). For simplicity of presentation, we let $\mathcal{G}(\boldsymbol{z}) = \boldsymbol{z} \in \mathbb{R}^d$ and focus on the discriminator for now. We will discuss the case where $\mathcal{G}(\cdot)$ is a generator network in Section 3.2. Given an unlabeled, $d$-dimensional dataset $\boldsymbol{X}^n \in \mathbb{R}^{n \times d}$ of $n$ points, we first augment $\boldsymbol{X}^n$ to obtain a labeled training set $\mathbb{D}^{2n} = (\boldsymbol{X}^n \oplus \boldsymbol{Z}^n \in \mathbb{R}^{2n \times d}, \boldsymbol{1}^n \oplus \boldsymbol{0}^n \in \mathbb{R}^{2n})$, where $\boldsymbol{Z}^n \in \mathbb{R}^{n \times d}$ contains $n$ generated points, $\boldsymbol{1}^n \in \mathbb{R}^n$ and $\boldsymbol{0}^n \in \mathbb{R}^n$ are label vectors of ones and zeros, respectively, and $\oplus$ is the vertical stack operator. Then, we model a discriminator trained on $\mathbb{D}^{2n}$ as an NTK-GP. Let $\boldsymbol{K}^{2n,2n} \in \mathbb{R}^{2n \times 2n}$ be the kernel matrix for $\boldsymbol{X}^n \oplus \boldsymbol{Z}^n$, where the value of each element $K_{i,j}^{2n,2n} = k((\boldsymbol{X}^n \oplus \boldsymbol{Z}^n)_{i,:}, (\boldsymbol{X}^n \oplus \boldsymbol{Z}^n)_{j,:})$ can be computed once we decide the initialization, activation function, and architecture of the element networks in the target infinite ensemble, i.e., the discriminator. By Eq. (2) and let $\lambda = \eta \cdot t$, the mean predictions of the discriminator can be written as

$$\mathcal{D}(\boldsymbol{X}^n, \boldsymbol{Z}^n; k, \lambda) = (\boldsymbol{I}^{2n} - e^{-\lambda \boldsymbol{K}^{2n,2n}})(\boldsymbol{1}^n \oplus \boldsymbol{0}^n) \in \mathbb{R}^{2n}, \tag{3}$$

where $\boldsymbol{I}^{2n} \in \mathbb{R}^{2n \times 2n}$ is an identity matrix. We formulate the objective of GA-NTK as follows:

$$\arg\min_{\boldsymbol{Z}^n} \mathcal{L}(\boldsymbol{Z}^n), \text{ where } \mathcal{L}(\boldsymbol{Z}^n) = \|\boldsymbol{1}^{2n} - \mathcal{D}(\boldsymbol{X}^n, \boldsymbol{Z}^n; k, \lambda)\|. \tag{4}$$

$\mathcal{L}(\cdot)$ is the loss function and $\boldsymbol{1}^{2n} \in \mathbb{R}^{2n}$ is a vector of ones. Statistically, Eq. (4) aims to minimize the Pearson $\chi^2$-divergence (Jeffreys, 1946), a case of $f$-divergence, between $\mathcal{P}_{\text{data}} + \mathcal{P}_{\text{gen}}$ and $2\mathcal{P}_{\text{gen}}$, where $\mathcal{P}_{\text{gen}}$ is the distribution of generated points. Please see Section 6 in Appendix for more details.

GA-NTK formulates an adversarial data synthesis task as a single-level optimization problem. On one hand, GA-NTK aims to find points $\boldsymbol{Z}^n$ that best deceive the discriminator such that it outputs wrong labels $\boldsymbol{1}^{2n}$ for these points. On the other hand, the discriminator is trained on $\mathbb{D}^{2n}$ with the correct labels $\boldsymbol{1}^n \oplus \boldsymbol{0}^n$ and therefore has the opposite goal of distinguishing between the real and generated points. Such an adversarial setting can be made single-level because the training dynamics of the discriminator $\mathcal{D}$ by gradient descent can be completely described by a closed-form formula in Eq. (3)—any change of $\boldsymbol{Z}^n$ causes $\mathcal{D}$ to be "retrained" instantly. Therefore, one can easily solve Eq. (4) by ordinary SGD.

**Training.** Before running SGD, one needs to tune the hyperparameter $\lambda$. We show in the next section that the value of $\lambda$ should be large enough but finite. Therefore, the complete training process of GA-NTK is to 1) find the minimal $\lambda$ that allows the discriminator to separate real data from pure noises in an auxiliary task, and 2) solve $\boldsymbol{Z}^n$ in Eq. (4) by ordinary SGD with the fixed $\lambda$. Please see Section 7.3 in Appendix for more details.

### 3.1 MERITS

As compared to GANs, GA-NTK offers the following advantages: **Convergence.** The GA-NTK can be trained by ordinary gradient descent. This gives much nicer convergence properties:

**Theorem 3.1** *Let $s$ be the number of the gradient descent iterations solving Eq. (4), and let $\boldsymbol{Z}^{n,(s)}$ be the solution at the $s$-th iteration. Suppose the following values are bounded: (a) $\boldsymbol{X}_{i,j}^n$ and $\boldsymbol{Z}_{i,j}^{n,(0)}$, $\forall i, j$, (b) $t$ and $\eta$, and (c) $\sigma$ and $L$. Also, assume that (d) $\boldsymbol{X}^n$ contains finite, non-identical, normalized rows. Then, for a sufficiently large $t$, we have*

$$\min_{j \leq s} \|\nabla_{\boldsymbol{Z}^n} \mathcal{L}(\boldsymbol{Z}^{n,(j)})\|^2 \leq O(\frac{1}{s-1}).$$

---

[3]From GAN perspective, our work can be regarded as a special case of the framework proposed by Franceschi et al. (2021), where the discriminator neglects the effect of historical generator updates and only distinguish between the true and currently generated data at each alternating step.

We prove the above theorem by showing that, with a large enough $\lambda$, $\nabla_{\boldsymbol{Z}^n} \mathcal{L}(\boldsymbol{Z}^{n,(s)})$ is smooth enough to lead to the convergence of gradient descent. For more details, please see Section 6 in Appendix. **Diversity.** GA-NTK avoids mode collapse due to the confusion between the min-max and max-min problems in alternating SGD. Given different initial values, the generated points in $\boldsymbol{Z}^n$ can be very different from each other. **No vanishing gradients, no side effects.** The hyperparameter $\lambda$ controls how much $\mathcal{D}$ should learn from the true and fake data during each iteration. Figure 5 shows the gradients of $\mathcal{D}$ with a finite $\lambda$, which do not saturate. This avoids the necessity of using a loss that imposes side effects, such as the Wasserstein distance (Arjovsky et al., 2017; Gulrajani et al., 2017) whose efficient evaluation requires Lipschitz continuity of $\mathcal{D}$.

## 3.2 GA-NTK IN PRACTICE

**Scalability.** To generate a large number of points, we can parallelly solve multiple $\boldsymbol{Z}^n$'s in Eq. (4) on different machines. On a single machine, the gradients of $\boldsymbol{Z}^n$ need to be back-propagated through the computation of $\boldsymbol{K}^{2n,2n}$, which has $O(n^2)$ space complexity. This may incur scalability issues for large datasets. Although recent efforts by Arora et al. (2019); Bietti & Mairal (2019); Han et al. (2021); Zandieh et al. (2021) have been made to reduce the time and space complexity of the evaluation of NTK and its variants, they are still at an early stage of development and the consumed space in practice may still be too large. To alleviate this problem, we propose the batch-wise GA-NTK with the objective

$$\arg \min_{\boldsymbol{Z}^n} \mathbb{E}_{\boldsymbol{X}^{b/2} \subset \boldsymbol{X}^n, \boldsymbol{Z}^{b/2} \subset \boldsymbol{Z}^n} \|\mathbf{1}^b - \mathcal{D}(\boldsymbol{X}^{b/2}, \boldsymbol{Z}^{b/2}; k, \lambda)\|, \tag{5}$$

that can be solved using mini-batches: during each gradient descent iteration, we 1) randomly sample a batch of $b$ rows in $\boldsymbol{X}^n \oplus \boldsymbol{Z}^n$ and their corresponding labels, and 2) update $\boldsymbol{Z}^n$ based on $\boldsymbol{K}^{b,b}$. Although the batch-wise GA-NTK is cosmetically similar to the original GA-NTK, it solves a different problem. In the original GA-NTK, the $\boldsymbol{Z}^n$ aims to fool a single discriminator $\mathcal{D}$ trained on $2n$ examples, while in the batch-wise GA-NTK, the $\boldsymbol{Z}^n$'s goal is to deceive *many* discriminators, each trained on $b$ examples only. Fortunately, Shankar et al. (2020); Arora et al. (2020) have shown that NTK-based methods perform well on small datasets. We will conduct experiments to verify this later.

**Generator Network.** So far, we let $\mathcal{G}(\boldsymbol{z}) = \boldsymbol{z}$ and show that a generator is *not* necessary in adversarial data synthesis.[4] Nevertheless, the presence of a generator network may be favorable in some applications to save time and memory at inference time. This can be done by extending the batch-wise GA-NTK as follows:

$$\arg \min_{\boldsymbol{\theta}_{\mathcal{G}}} \mathbb{E}_{\boldsymbol{X}^{b/2} \subset \boldsymbol{X}^n, \boldsymbol{Z}^{b/2} \sim \mathcal{N}(\mathbf{0}, \mathbf{I})} \|\mathbf{1}^b - \mathcal{D}(\boldsymbol{X}^{b/2}, \mathcal{G}(\boldsymbol{Z}^{b/2}; \boldsymbol{\theta}_{\mathcal{G}}); k, \lambda)\|, \tag{6}$$

where $\mathcal{G}(\cdot; \boldsymbol{\theta}_{\mathcal{G}})$ is a generator network parametrized by $\boldsymbol{\theta}_{\mathcal{G}}$, and $\boldsymbol{Z} \in \mathbb{R}^l$ where $l \leq d$. Note that this is still a single-level objective, and $\boldsymbol{\theta}_{\mathcal{G}}$ can be solved by gradient descent. We denote this variant GA-NTKg.

**Image Quality.** To generate images, one can pair up GA-NTK with a convolutional neural tangent kernel (CNTK) (Arora et al., 2019; Novak et al., 2019b; Garriga-Alonso et al., 2019; Yang, 2019a) that approximates a CNN with infinite channels. This allows the NTK-GP (discriminator) to distinguish between real and fake points based on local patterns in the pixel space. However, the images synthesized by this GA-NTK variant may lack global coherency, just like the images generated by the CNN-based GANs (Radford et al., 2016; Salimans et al., 2016). Many efforts have been made to improve the image quality of CNN-based GANs, and this paper opens up opportunities for them to be adapted to the kernel regime. In particular, we propose the multi-resolutional GA-CNTK based on the work by Wang et al. (2018), whose objective is formulated as:

$$\arg \min_{\boldsymbol{Z}^n} \sum_m \|\mathbf{1}^{2n} - \mathcal{D}^m(\text{pool}^m(\boldsymbol{X}^n), \text{pool}^m(\boldsymbol{Z}^n); k^m, \lambda^m)\|, \tag{7}$$

where $\mathcal{D}^m$ is an NTK-GP taking input at a particular pixel resolution and $\text{pool}^m(\cdot)$ is a downsample operation (average pooling) applied to each row of $\boldsymbol{X}^n$ and $\boldsymbol{Z}^n$. The generated points in $\boldsymbol{Z}^n$ aim to simultaneously fool multiple NTK-GPs (discriminators), each classifying real and fake images at a distinct pixel resolution. The NTK-GPs working at low and high resolutions encourage global coherency and details, respectively, and together they lead to more plausible points in $\boldsymbol{Z}^n$.

---

[4]In GANs, solving $\boldsymbol{Z}$ directly against a finite-width discriminator is infeasible because it amounts to finding adversarial examples (Goodfellow et al., 2015) whose gradients are known to be very noisy (Ilyas et al., 2019).

## 4 EXPERIMENTS

We conduct experiments to study how GA-NTK works in image generation.

**Datasets.** We consider the unsupervised/unconditional image synthesis tasks over real-world datasets, including MNIST (LeCun et al., 2010), CIFAR-10 (Krizhevsky, 2009), CelebA (Liu et al., 2015), CelebA-HQ (Liu et al., 2015), and ImageNet (Deng et al., 2009). To improve training efficiency, we resize CelebA images to $64 \times 64$ and ImageNet images to $128 \times 128$ pixels, respectively. We also create a 2D toy dataset consisting of 25-modal Gaussian mixtures of points to visualize the behavior of different image synthesis methods. **GA-NTK implementations.** GA-NTK works with different NTK-GPs. For the image synthesis tasks, we consider the NTK-GPs that model the ensembles of fully-connected networks (Jacot et al., 2018; Lee et al., 2019; Chizat et al., 2019) and convolutional networks (Arora et al., 2019; Novak et al., 2019b; Garriga-Alonso et al., 2019; Yang, 2019a), respectively. We implement GA-NTK using the Neural Tangents library (Novak et al., 2019a) and call the variants based on the former and latter NTK-GPs the GA-FNTK and GA-CNTK, respectively. In GA-FNTK, an element network of the discriminator has 3 infinitely wide, fully-connected layers with ReLU non-linearity, while in GA-CNTK, an element network follows the architecture of InfoGAN (Chen et al., 2016) except for having infinite filters at each layer. We tune the hyperparameters of GA-FNTK and GA-CNTK following the method proposed in Poole et al. (2016); Schoenholz et al. (2017); Raghu et al. (2017). We also implement their batch-wise, generator, and multi-resolutional variants described in Section 3.2. See Section 7 in Appendix for more details. **Baselines.** We compare GA-NTK with some popular variants of GANs, including vanilla GANs (Goodfellow et al., 2014), DCGAN (Radford et al., 2016), LSGAN (Mao et al., 2017), WGAN (Arjovsky et al., 2017), WGAN-GP (Gulrajani et al., 2017), SNGAN (Miyato et al., 2018) and StyleGAN2 (Karras et al., 2020). To give a fair comparison, we let the discriminator of each baseline follow the architecture of InfoGAN (Chen et al., 2016) and tune the hyperparameters using grid search. **Metrics.** We evaluate the quality of a set of generated images using the Fréchet Inception Distance (FID) (Heusel et al., 2017). The lower the FID score the better. We find that an image synthesis method may produce downgrade images that look almost identical to some images in the training set. Therefore, we also use a metric called the average max-SSIM (AM-SSIM) that calculates the average of the maximum SSIM score (Wang et al., 2004) between $\mathcal{P}_{\text{gen}}$ and $\mathcal{P}_{\text{data}}$:

$$\text{AM-SSIM}(\mathcal{P}_{\text{gen}}, \mathcal{P}_{\text{data}}) = \mathbb{E}_{\boldsymbol{x}' \sim \mathcal{P}_{\text{gen}}} \big[ \max_{\boldsymbol{x} \sim \mathcal{P}_{\text{data}}} \text{SSIM}(\boldsymbol{x}', \boldsymbol{x}) \big].$$

A generated image set will have a higher AM-SSIM score if it contains downgrade images. **Environment and limitations.** We conduct all experiments on a cluster of machines having 80 NVIDIA Tesla V100 GPUs. As discussed in 3.2, GA-NTK consumes a significant amount of memory on each machine due to the computations involved in the kernel matrix $\boldsymbol{K}^{2n,2n}$. With the current version of Neural Tangents library (Novak et al., 2019a) and a V100 GPU of 32GB RAM, the maximum sizes of the training set from MNIST, CIFAR-10, CelebA, and ImageNet are 1024, 512, 256, and 128, respectively (where the computation graph and backprop operations of $\boldsymbol{K}^{2n,2n}$ consume about 27.5 GB RAM excluding other necessary operations). Since our goal is not to achieve state-of-the-art performance but to compare different image synthesis methods, we train all the methods using up to 256 images randomly sampled from all classes of MNIST, the "horse" class CIFAR-10, the "male with straight hair" class of CelebA, and the "daisy" class of ImageNet, respectively. We will conduct larger-scale experiments in Section 9.2. For more details about our experiment settings, please see Section 7 in Appendix.

### 4.1 IMAGE QUALITY

We first study the quality of the images synthesized by different methods. Table 1 summarizes the FID and AM-SSIM scores of the generated images. LSGAN and DCGAN using $f$-divergence as the loss function give high FID and fail to generate recognizable images on CIFAR-10 and CelebA datasets due to the various training issues mentioned previously. StyleGAN, although being able to generate impressive images with sufficient training data, gives high FID here due to the high sample complexity of the style-based generator. Other baselines, including WGAN, WGAN-GP, and SN-GAN, can successfully generate recognizable images on all datasets, as shown in Figure 1. In particular, WGAN-GP performs the best among the GAN variants. However, WGAN-GP limits the Lipschitz continuity of the discriminator and gives higher FID scores than GA-CNTK. Also, it gives higher AM-SSIM values as the size of the training set decreases, implying there are many

Table 1: The FID and AM-SSIM scores of the images generated by different methods.

| | $n$ | Metric | DCGAN | LSGAN | WGAN | WGANGP | SNGAN | StyleGAN | GACNTK | GACNTKg |
|---|---|---|---|---|---|---|---|---|---|---|
| MNIST | 64 | FID | 27.43 | 69.76 | 50.69 | 32.49 | 57.89 | 91.82 | 31.10 | 32.43 |
| | | AMSSIM | 0.84 | 0.79 | 0.77 | 0.83 | 0.67 | 0.69 | 0.49 | 0.71 |
| | 128 | FID | 31.89 | 38.52 | 49.28 | 30.20 | 38.33 | 88.31 | 21.14 | 36.50 |
| | | AMSSIM | 0.85 | 0.80 | 0.74 | 0.76 | 0.67 | 0.66 | 0.52 | 0.72 |
| | 256 | FID | 69.76 | 35.33 | 50.33 | 24.37 | 29.49 | 84.7 | 14.96 | 51.21 |
| | | AMSSIM | 0.69 | 0.78 | 0.72 | 0.73 | 0.70 | 0.65 | 0.54 | 0.73 |
| CIFAR-10 | 64 | FID | 312.21 | 258.41 | 117.85 | 49.29 | 118.16 | 406.02 | 55.54 | 106.44 |
| | | AMSSIM | 0.22 | 0.25 | 0.29 | 0.74 | 0.28 | 0.64 | 0.41 | 0.44 |
| | 128 | FID | 229.94 | 339.27 | 101.90 | 68.53 | 128.65 | 484.36 | 39.98 | 61.19 |
| | | AMSSIM | 0.36 | 0.10 | 0.26 | 0.60 | 0.21 | 0.39 | 0.41 | 0.44 |
| | 256 | FID | 181.15 | 255.19 | 111.92 | 85.34 | 107.29 | 426.58 | 28.40 | 55.46 |
| | | AMSSIM | 0.27 | 0.22 | 0.22 | 0.46 | 0.20 | 0.26 | 0.42 | 0.44 |
| CelebA | 64 | FID | 489.82 | 83.71 | 122.36 | 83.71 | 169.04 | 323.37 | 30.83 | 95.91 |
| | | AMSSIM | 0.02 | 0.05 | 0.29 | 0.56 | 0.29 | 0.23 | 0.60 | 0.21 |
| | 128 | FID | 55.01 | 450.81 | 125.82 | 92.73 | 168.11 | 337.58 | 33.51 | 58.39 |
| | | AMSSIM | 0.03 | 0.11 | 0.28 | 0.54 | 0.28 | 0.21 | 0.51 | 0.38 |
| | 256 | FID | 461.95 | 403.79 | 108.07 | 79.36 | 161.20 | 333.16 | 63.15 | 78.46 |
| | | AMSSIM | 0.04 | 0.09 | 0.31 | 0.39 | 0.27 | 0.30 | 0.38 | 0.40 |

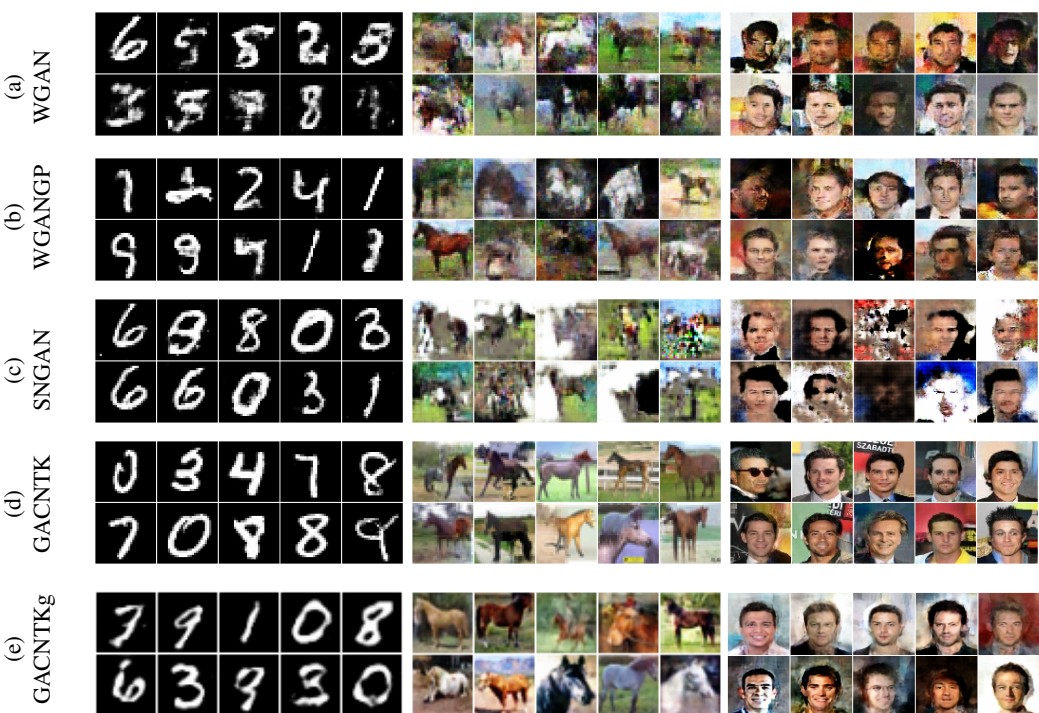

Figure 1: The images generated by different methods on MNIST, CIFAR-10, and CelebA datasets given only 256 training images.

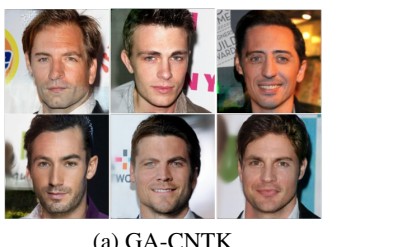
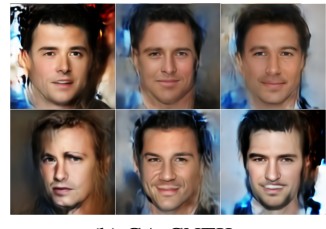

(a) GA-CNTK          (b) GA-CNTKg

Figure 2: The images generated by GA-CNTK (a) without and (b) with a generator given 256 CelebA-HQ training images.

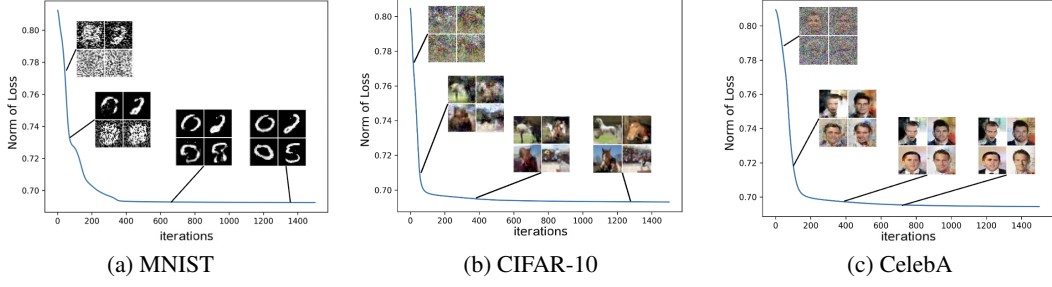

(a) MNIST         (b) CIFAR-10         (c) CelebA

Figure 3: The learning curve and image quality at different stages of a training process.

downgrade images that look identical to some training images. This is because the Wasserstein distance, which is also called the earth mover's distance, allows fewer ways of moving when there are less available "earth" (i.e., the density values of $\mathcal{P}_{\text{data}}$ and $\mathcal{P}_{\text{gen}}$) due to a small $n$, and thus the $\mathcal{P}_{\text{gen}}$ needs to be exactly the same as $\mathcal{P}_{\text{data}}$ to minimize the distance.[5] The GA-NTK variants, including GA-CNTK and GA-CNTKg ("g" means "with generator"), perform relatively well due to their lower sample complexity, which aligns with the previous observations (Shankar et al., 2020; Arora et al., 2020) in different context.

Next, we compare the images generated by the multi-resolutional GA-CNTK and GA-CNTKg (see Section 3.2) on the CelebA-HQ dataset. The multi-resolutional GA-CNTK employs 3 discriminators working at 256×256, 64×64, and 16×16 pixel resolutions, respectively. Figure 2 shows the results. We can see that the multi-resolutional GA-CNTK (without a generator) gives better-looking images than GA-CNTKg (with a generator) because learning a generator, which maps two spaces, is essentially a harder problem than finding a set of plausible $z$'s. Although synthesizing data faster at inference time, a generator may not be necessary to generate high-quality images under the adversarial setting.

## 4.2 TRAINING STABILITY

**Convergence.** Figure 3 shows the learning curve and the relationship between the image quality and the number of gradient descent iterations during a training process of GA-CNTK. We find that

---

[5]The problem of Lipschitz continuity may be alleviated when $n$ becomes larger.

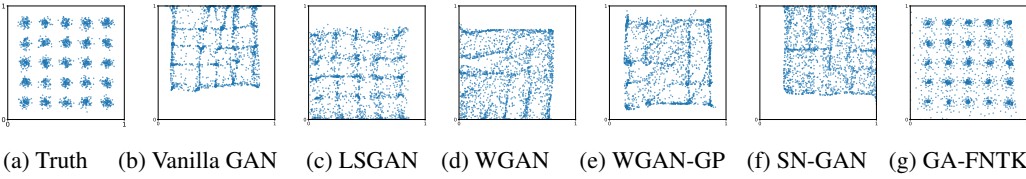

(a) Truth    (b) Vanilla GAN    (c) LSGAN    (d) WGAN    (e) WGAN-GP    (f) SN-GAN    (g) GA-FNTK

Figure 4: Visualization of distribution alignment and mode collapse on a 2D toy dataset.

GA-CNTK easily converges under various conditions, which is supported by Theorem 3.1. Furthermore, we can see a correlation between the image quality and the loss value—as the loss becomes smaller, the quality of the synthesized images improves. This correlation can save human labor from monitoring the training processes, which is common when training GANs. Note that the images generated in the latter stage of training contain recognizable patterns that change over training time. This is a major source of GA-CNTK creativity. Please see Section 9.4 for more discussions.

**Mode collapse.** To study how different methods align $\mathcal{P}_{\text{gen}}$ with $\mathcal{P}_{\text{data}}$, we train them using a 2D toy training set where $\mathcal{P}_{\text{data}}$ is a 25-modal Gaussian mixture. We use two 3-layer fully-connected neural networks as the generator and discriminator for each baseline and an ensemble of 3-layer, infinitely wide counterpart as the discriminator in GA-FNTK. For GANs, we stop the alternating SGD training when the generator receives 1000 updates, and for GA-FNTK, we terminate the GD training after 1000 iterations. Figure 4 shows the resultant $\mathcal{P}_{\text{gen}}$ of different methods. GA-FNTK avoids mode collapse due to the use of alternating SGD. **Gradient vanishing.** To verify that GA-NTK gives no vanishing gradients with a finite $\lambda$, we conduct an experiment using another toy dataset consisting of 256 MNIST images and 256 random noises. We replace the discriminator of GA-CNTK with a single parametric network of the same architecture but finite width. We train the finite-width

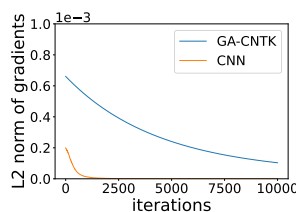

Figure 5: Comparison between the gradients of a $\mathbf{Z}_{i,:}$ in Eq. (4) obtained from different types of $\mathcal{D}$.

network on the toy dataset by minimizing the MSE loss using gradient descent. We set the training iteration to a large value (65536) to simulate the situation where the network value becomes saturated on both sides of the decision boundary. Figure 5 compares the gradients of a generated image $\mathbf{Z}_{i,:}^n$ in Eq. (4) obtained from 1) the finite-width network and 2) the corresponding GA-CNTK with a large $t$. As $\mathbf{Z}_{i,:}^n$ evolves through gradient descent iterations, the norm of its gradients obtained from the finite-width discriminator quickly shrinks to zero. On the other hand, the gradient norm obtained from the discriminator of GA-CNTK is always positive thanks to the infinite ensembling.

### 4.3 SCALABILITY

Unlike GA-CNTK, the GA-CNTKg is batch-wise and thus can be trained by more examples. Here, we scale up WGAN-GP and GA-CNTKg by training them on CelebA dataset consisting of 2048 images. The batch size is 256. Table 2 summarizes the FID and AM-SSIM scores of the generated images. On MNIST, WGAN-GP slightly outperforms GA-CNTKg. The training of WGAN-GP on MNIST is easy, so GA-CNTKg does not offer much advantage. However, in a more complex task like CIFAR-10 or CelebA, GA-CNTKg outperforms WGAN-GP, suggesting that our single-level modeling is indeed beneficial.

We have conducted more experiments. Please see Appendix for their results.

Table 2: The FID and AM-SSIM scores of the images output by WGAN-GP and GA-CNTKg trained on 2048 CelebA images with batch size 256.

| $n$=2048 | Metric | WGANGP | GACNTKg |
|---|---|---|---|
| MNIST | FID | 23.47 | 56.73 |
| | ASSIM | 0.786 | 0.787 |
| CIFAR-10 | FID | 110.70 | 78.85 |
| | ASSIM | 0.404 | 0.432 |
| CelebA | FID | 67.29 | 59.91 |
| | ASSIM | 0.337 | 0.411 |

## 5 CONCLUSION

We proposed GA-NTK and showed that adversarial data synthesis can be done via single-level modeling. It can be solved by ordinary gradient descent, avoiding the difficulties of bi-level training of GANs. We analyzed the convergence behavior of GA-NTK and gave sufficient conditions for convergence. Extensive experiments were conducted to study the advantages and limitations of GA-NTK. We proposed the batch-wise and multi-resolutional variants to improve memory efficiency and image quality, and showed that GA-NTK works either with or without a generator network. GA-NTK works well with small data, making it suitable for applications where data are hard to collect. GA-NTK also opens up opportunities for one to adapt various GAN enhancements into the kernel regime. These are matters of our future inquiry.

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

## 6 STATISTICAL INTERPRETATION OF GA-NTK

Statistically, minimizing Eq. (4) or (6) amounts to minimizing the Pearson $\chi^2$-divergence (Jeffreys, 1946), a case of $f$-divergence (Rényi et al., 1961), between $\mathcal{P}_{\text{data}} + \mathcal{P}_{\text{gen}}$ and $2\mathcal{P}_{\text{gen}}$, where $\mathcal{P}_{\text{data}}$ is the distribution of real data and $\mathcal{P}_{\text{gen}}$ is the distribution of generated points. To see this, we first rewrite the loss of our discrimonator $\mathcal{D}$, denoted by $\mathcal{L}(\mathcal{D})$, in expectation:

$$\arg\min_{\mathcal{D}} \mathcal{L}(\mathcal{D}) = \arg\min_{\mathcal{D}} \mathbb{E}_{\boldsymbol{x} \sim \mathcal{P}_{\text{data}}}\big[(\mathcal{D}(\boldsymbol{x}) - 1)^2\big] + \mathbb{E}_{\boldsymbol{x} \sim \mathcal{P}_{\text{gen}}}\big[(\mathcal{D}(\boldsymbol{x}) - 0)^2\big]. \tag{8}$$

Here, $\mathcal{P}_{\text{gen}}$ can represent either $\boldsymbol{Z}$ in Eq. (4) or the output of the generator $\mathcal{G}$ in Eq. (6). Similarly, the loss function for our $\mathcal{P}_{\text{gen}}$, denoted by $\mathcal{L}(\mathcal{P}_{\text{gen}}; \mathcal{D})$, can be written as follows:

$$\arg\min_{\mathcal{P}_{\text{gen}}} \mathcal{L}(\mathcal{P}_{\text{gen}}; \mathcal{D}) = \arg\min_{\mathcal{P}_{\text{gen}}} \mathbb{E}_{\boldsymbol{x} \sim \mathcal{P}_{\text{data}}}\big[(\mathcal{D}(\boldsymbol{x}) - 1)^2\big] + \mathbb{E}_{\boldsymbol{x} \sim \mathcal{P}_{\text{gen}}}\big[(\mathcal{D}(\boldsymbol{x}) - 1)^2\big]. \tag{9}$$

GA-NTK, in the form of Eqs. (8) and (9), is a special case of LSGAN (Mao et al., 2017). Let $\mathcal{D}^*$ be the minimizer of Eq. (8). We can see that Eqs. (4) and (6) effectively solve the problem:

$$\arg\min_{\mathcal{P}_{\text{gen}}} \mathcal{L}(\mathcal{P}_{\text{gen}}; \mathcal{D}^*) = \arg\min_{\mathcal{P}_{\text{gen}}} \mathbb{E}_{\boldsymbol{x} \sim \mathcal{P}_{\text{data}}}\big[(\mathcal{D}^*(\boldsymbol{x}) - 1)^2\big] + \mathbb{E}_{\boldsymbol{x} \sim \mathcal{P}_{\text{gen}}}\big[(\mathcal{D}^*(\boldsymbol{x}) - 1)^2\big]. \tag{10}$$

Mao et al. (2017) show that, under mild relaxation, minimizing Eq. (10) yields minimizing the Pearson $\chi^2$-divergence between $\mathcal{P}_{\text{data}} + \mathcal{P}_{\text{gen}}$ and $2\mathcal{P}_{\text{gen}}$:

$$\arg\min_{\mathcal{P}_{\text{gen}}} \mathcal{L}(\mathcal{P}_{\text{gen}}; \mathcal{D}^*) = \arg\min_{\mathcal{P}_{\text{gen}}} \chi^2_{\text{Pearson}}(\mathcal{P}_{\text{data}} + \mathcal{P}_{\text{gen}} \| 2\mathcal{P}_{\text{gen}})$$

$$= \arg\min_{\mathcal{P}_{\text{gen}}} \int (\mathcal{P}_{\text{data}}(\boldsymbol{x}) + \mathcal{P}_{\text{gen}}(\boldsymbol{x})) \left( \frac{2\mathcal{P}_{\text{gen}}(\boldsymbol{x})}{\mathcal{P}_{\text{data}}(\boldsymbol{x}) + \mathcal{P}_{\text{gen}}(\boldsymbol{x})} - 1 \right)^2 \mathrm{d}\boldsymbol{x}.$$

The loss becomes zero when $\mathcal{P}_{\text{data}}(\boldsymbol{x}) = \mathcal{P}_{\text{gen}}(\boldsymbol{x})$ for all $\boldsymbol{x}$. Therefore, minimizing Eq. (4) or (6) brings $\mathcal{P}_{\text{gen}}$ closer to $\mathcal{P}_{\text{data}}$.

# 7 PROOF OF THEOREM 3.1

In this section, we prove the convergence of a GA-NTK whose discriminator $\mathcal{D}$ approximates an infinite ensemble of infinitely-wide, fully-connected, feedforward neural networks. The proof can be easily extended to other network architectures such as convolutional neural networks.

## 7.1 BACKGROUND AND NOTATION

Consider a fully-connected, feedforward neural network $f : \mathbb{R}^d \rightarrow \mathbb{R}$,

$$f(\boldsymbol{x}; \boldsymbol{\theta}) = \frac{\sigma_w}{\sqrt{d^{L-1}}} \boldsymbol{w}^L \phi \left( \frac{\sigma_w}{\sqrt{d^{L-2}}} \boldsymbol{W}^{L-1} \phi \left( \cdots \phi \left( \frac{\sigma_w}{\sqrt{d}} \boldsymbol{W}^1 \boldsymbol{x} + \sigma_b \boldsymbol{b}^1 \right) \cdots \right) + \sigma_b \boldsymbol{b}^{L-1} \right) + \sigma_b \boldsymbol{b}^L, \tag{11}$$

where $\phi(\cdot)$ is the activation function (applied element-wisely), $L$ is the number of hidden layers, $\{d^1, \cdots, d^{L-1}\}$ are the dimensions (widths) of hidden layers, $\boldsymbol{\theta} = \cup_{l=1}^L \boldsymbol{\theta}^l = \cup_{l=1}^L (\boldsymbol{W}^l \in \mathbb{R}^{d^l \times d^{l-1}}, \boldsymbol{b}^l \in \mathbb{R}^{d^l})$ are trainable weights and biases whose initial values are i.i.d. Gaussian random variables $\mathcal{N}(0, 1)$, and $\sigma_w^2$ and $\sigma_b^2$ are scaling factors that control the variances of weights and biases, respectively. Suppose $f$ is trained on a labeled dataset $\mathbb{D}^{2n} = (\boldsymbol{X}^n \oplus \boldsymbol{Z}^n \in \mathbb{R}^{2n \times d}, \boldsymbol{1}^n \oplus \boldsymbol{0}^n \in \mathbb{R}^{2n})$ by minimizing the MSE loss using $t$ gradient-descent iterations with the learning rate $\eta$. Let $\boldsymbol{\theta}^{(0)}$ and $\boldsymbol{\theta}^{(t)}$ be the initial and trained parameters, respectively. As $d^1, \cdots, d^L \rightarrow \infty$, we can approximate the distribution of $f(\boldsymbol{x}; \boldsymbol{\theta}^{(t)})$ as a Gaussian process (NTK-GP) (Jacot et al., 2018; Lee et al., 2019; Chizat et al., 2019) whose behavior is controlled by a kernel matrix

$$\boldsymbol{K}^{2n,2n} = \nabla_{\boldsymbol{\theta}} f(\boldsymbol{X}^n \oplus \boldsymbol{Z}^n; \boldsymbol{\theta}^{(0)})^\top \nabla_{\boldsymbol{\theta}} f(\boldsymbol{X}^n \oplus \boldsymbol{Z}^n; \boldsymbol{\theta}^{(0)}) \in \mathbb{R}^{2n \times 2n}, \tag{12}$$

where $f(\boldsymbol{X}^n \oplus \boldsymbol{Z}^n; \boldsymbol{\theta}^{(0)}) \in \mathbb{R}^{2n}$ is the vector of in-sample predictions made by the initial $f$. The value of each element $K_{i,j}^{2n,2n} = k^L((\boldsymbol{X}^n \oplus \boldsymbol{Z}^n)_{i,:}, (\boldsymbol{X}^n \oplus \boldsymbol{Z}^n)_{j,:})$ presents the similarity score of two rows (points) of $\boldsymbol{X}^n \oplus \boldsymbol{Z}^n$ in a kernel space, and it can be expressed by a kernel function $k^L : \mathbb{R}^d \times \mathbb{R}^d \rightarrow \mathbb{R}$, called the neural tangent kernel (NTK). The NTK is deterministic as it depends only on $\phi(\cdot)$, $\sigma_w$, $\sigma_b$, and $L$ rather than the specific values in $\boldsymbol{\theta}^{(0)}$. Furthermore, it can be evaluated layer-wisely. Let $h_j^l(\boldsymbol{x}) \in \mathbb{R}^{d^l}$ be the pre-activation of the $j$-th neuron at the $l$-th layer of $f(\boldsymbol{x}; \boldsymbol{\theta}^{(t)})$. The distribution of $h_j^l(\boldsymbol{x})$ is still an NTK-GP, and its associated NTK is defined as $k^l : \mathbb{R}^d \times \mathbb{R}^d \rightarrow \mathbb{R}$,

$$k^l(\boldsymbol{x}, \boldsymbol{x}') = \nabla_{\boldsymbol{\theta}^{\leq l}} h_j^l(\boldsymbol{x})^\top \nabla_{\boldsymbol{\theta}^{\leq l}} h_j^l(\boldsymbol{x}'),$$

where $\boldsymbol{\theta}^{\leq l} = \cup_{i=1}^l \boldsymbol{\theta}^i$. Note that all $h_j^l(\boldsymbol{x})$'s, $\forall j$, are i.i.d. and thus share the same kernel. It can be shown that

$$\begin{aligned} k^l(\boldsymbol{x}, \boldsymbol{x}') &= \nabla_{\boldsymbol{\theta}^l} h_j^l(\boldsymbol{x})^\top \nabla_{\boldsymbol{\theta}^l} h_j^l(\boldsymbol{x}') + \nabla_{\boldsymbol{\theta}^{\leq l-1}} h_j^l(\boldsymbol{x})^\top \nabla_{\boldsymbol{\theta}^{\leq l-1}} h_j^l(\boldsymbol{x}') \\ &= \tilde{k}^l(\boldsymbol{x}, \boldsymbol{x}') + \sigma_w^2 k^{l-1}(\boldsymbol{x}, \boldsymbol{x}') \mathbb{E}_{(h_j^{(l-1)}(\boldsymbol{x}), h_j^{(l-1)}(\boldsymbol{x}')) \sim \mathcal{N}(\boldsymbol{0}^2, \tilde{\boldsymbol{K}}^{l-1})} \left[ \phi'(h_j^{(l-1)}(\boldsymbol{x})) \phi'(h_j^{(l-1)}(\boldsymbol{x}')) \right] \end{aligned} \tag{13}$$

and

$$k^1(\boldsymbol{x}, \boldsymbol{x}') = \frac{\sigma_w^2}{d} \boldsymbol{x}^\top \boldsymbol{x}' + \sigma_b^2 \tag{14}$$

where $\tilde{k}^l : \mathbb{R}^d \times \mathbb{R}^d \rightarrow \mathbb{R}$ is the NNGP kernel (Lee et al., 2018; Matthews et al., 2018) that controls the behavior of another Gaussian process, called NNGP, approximating the distribution of $f(\boldsymbol{x}; \boldsymbol{\theta}^{(0)})$, and

$$\tilde{\boldsymbol{K}}^{l-1} = \begin{bmatrix} \tilde{k}^{l-1}(\boldsymbol{x}, \boldsymbol{x}) & \tilde{k}^{l-1}(\boldsymbol{x}, \boldsymbol{x}') \\ \tilde{k}^{l-1}(\boldsymbol{x}, \boldsymbol{x}') & \tilde{k}^{l-1}(\boldsymbol{x}', \boldsymbol{x}') \end{bmatrix} \in \mathbb{R}^{2 \times 2}.$$

## 7.2 CONVERGENCE

The GA-NTK employs the above NTK-GP as the discriminator $\mathcal{D}$. So, the in-sample mean predictions of $\mathcal{D}$ can be written as a closed-form formula:

$$\mathcal{D}(\boldsymbol{X}^n, \boldsymbol{Z}^n) = (\boldsymbol{I}^{2n} - e^{-\eta t \boldsymbol{K}^{2n,2n}}) \boldsymbol{y}^{2n} \in \mathbb{R}^{2n}, \tag{15}$$

where $\boldsymbol{I}^{2n}$ is an identity matrix and $\boldsymbol{y}^{2n} = \boldsymbol{1}^n \oplus \boldsymbol{0}^n \in \mathbb{R}^{2n}$ is the "correct" label vector for training $\mathcal{D}$. We formulate the objective of GA-NTK as:

$$\arg\min_{\boldsymbol{Z}^n} \mathcal{L}(\boldsymbol{Z}^n) = \arg\min_{\boldsymbol{Z}^n} \frac{1}{2} \|\boldsymbol{1}^{2n} - \mathcal{D}(\boldsymbol{X}^n, \boldsymbol{Z}^n)\|^2, \tag{16}$$

where $\boldsymbol{1}^{2n} \in \mathbb{R}^{2n}$ in the loss $\mathcal{L}(\cdot)$ is the "wrong" label vector that guides us to find the points $(\boldsymbol{Z}^n)$ that best deceive the discriminator. We show that

**Theorem 7.1** *Let $s$ be the number of the gradient descent iterations solving Eq. (16), and let $\boldsymbol{Z}^{n,(s)}$ be the solution at the $s$-th iteration. Suppose the following values are bounded: (a) $\boldsymbol{X}_{i,j}^n$ and $\boldsymbol{Z}_{i,j}^{n,(0)}$, $\forall i, j$, (b) $t$ and $\eta$, and (c) $\sigma$ and $L$. Also, assume that (d) $\boldsymbol{X}^n$ contains finite, non-identical, normalized rows. Then, for a sufficiently large $t$, we have*

$$\min_{j \leq s} \|\nabla_{\boldsymbol{Z}^n} \mathcal{L}(\boldsymbol{Z}^{n,(j)})\|^2 \leq O(\frac{1}{s-1}).$$

### 7.3 PROOF

To prove Theorem 7.1, we first introduce the notion of $\beta$ smoothness:

**Definition 7.1** *A continuously differentiable function $g : \mathbb{R}^d \to \mathbb{R}$ is $\beta$-smooth if there exits $\beta \in \mathbb{R}$ such that*

$$\|\nabla_{\boldsymbol{a}} g(\boldsymbol{a}) - \nabla_{\boldsymbol{b}} g(\boldsymbol{b})\| \leq \beta \|\boldsymbol{a} - \boldsymbol{b}\|$$

*for any $\boldsymbol{a}, \boldsymbol{b} \in \mathbb{R}^d$.*

It can be shown that gradient descent finds a stationary point of a $\beta$-smooth function efficiently (Gower, 2022).

**Lemma 7.1** *Let $\boldsymbol{a}^{(s)}$ be the input of a function $g : \mathbb{R}^d \to \mathbb{R}$ after applying $s$ gradient descent iterations to an initial input $\boldsymbol{a}^{(0)}$. If $g$ is $\beta$-smooth, then $g(\boldsymbol{a}^{(s)})$ converges to a stationary point at rate*

$$\min_{j \leq s} \|\nabla_{\boldsymbol{a}} g(\boldsymbol{a}^{(j)})\|^2 \leq O(\frac{1}{s-1}).$$

So, our goal is to show that the loss $\mathcal{L}(\boldsymbol{Z}^n)$ in Eq. (16) is $\beta$-smooth w.r.t. any generated point $\boldsymbol{z} \in \mathbb{R}^d$.

**Corollary 7.1** *If all the conditions (a)-(d) in Theorem 7.1 hold, there exits a constant $c_1 \in \mathbb{R}^+$ such that $\|\nabla_{\boldsymbol{z}} \mathcal{L}(\boldsymbol{Z}^n)\| \leq c_1$ for each row $\boldsymbol{z} \in \mathbb{R}^d$ of $\boldsymbol{Z}^n$. This makes $\mathcal{L}(\boldsymbol{Z}^n)$ $\beta$-smooth.*

To prove Corollary 7.1, consider $\mathcal{D}_i(\boldsymbol{X}^n, \boldsymbol{Z}^n)$ and $\nabla_{z_j} \mathcal{L}(\boldsymbol{Z}^n)$, the $i$-th and $j$-th elements of $\mathcal{D}(\boldsymbol{X}^n, \boldsymbol{Z}^n) \in \mathbb{R}^{2n}$ and $\nabla_{\boldsymbol{z}} \mathcal{L}(\boldsymbol{Z}^n) \in \mathbb{R}^d$, respectively. We have

$$\begin{aligned}
\nabla_{z_j} \mathcal{L}(\boldsymbol{Z}^n) &= \nabla_{z_j} \frac{1}{2} \|\boldsymbol{1}^{2n} - \mathcal{D}(\boldsymbol{X}^n, \boldsymbol{Z}^n)\|^2 \\
&= \sum_{i=1}^{2n} (\mathcal{D}_i(\boldsymbol{X}^n, \boldsymbol{Z}^n) - 1) \cdot \nabla_{z_j} \mathcal{D}_i(\boldsymbol{X}^n, \boldsymbol{Z}^n)
\end{aligned} \tag{17}$$

Given a sufficiently large $t$, the $\mathcal{D}_i(\boldsymbol{X}^n, \boldsymbol{Z}^n)$ can be arbitrarily close to $y_i \in \{0, 1\}$ because $\boldsymbol{K}^{2n,2n}$ is positive definite (Jacot et al., 2018) and therefore $(\boldsymbol{I}^{2n} - e^{-\eta t \boldsymbol{K}^{2n,2n}}) \to \boldsymbol{I}^{2n}$ as $t \to \infty$ in Eq. (15). There exists $\epsilon \in \mathbb{R}^+$ such that

$$\begin{aligned}
|\nabla_{z_j} \mathcal{L}(\boldsymbol{Z}^n)| &\leq \epsilon \sum_{i=1}^n |\nabla_{z_j} \mathcal{D}_i(\boldsymbol{X}^n, \boldsymbol{Z}^n) + (1+\epsilon) \sum_{i=n+1}^{2n} |\nabla_{z_j} \mathcal{D}_i(\boldsymbol{X}^n, \boldsymbol{Z}^n)| \\
&\leq (1+\epsilon) \sum_{i=1}^{2n} |\nabla_{z_j} \mathcal{D}_i(\boldsymbol{X}^n, \boldsymbol{Z}^n)| \\
&= (1+\epsilon) \sum_{i=1}^{2n} |\nabla_{z_j} \sum_{p=1}^{2n} (I_{i,p}^{2n} - e_{i,p}^{-\eta t \boldsymbol{K}^{2n,2n}}) y_p^{2n}| \\
&= (1+\epsilon) \eta t \sum_{i,p,q=1}^{2n} e_{i,q}^{-\eta t \boldsymbol{K}^{2n,2n}} |\nabla_{z_j} k^L((\boldsymbol{X}^n \oplus \boldsymbol{Z}^n)_{q,:}, (\boldsymbol{X}^n \oplus \boldsymbol{Z}^n)_{p,:}) y_p^{2n}|.
\end{aligned}$$

Note that $e_{i,q}^{-\eta t \boldsymbol{K}^{2n,2n}} \in \mathbb{R}^+$ can be arbitrarily close to 0 with a sufficiently large $t$. Hence, Corollary 7.1 holds as long as $\nabla_{z_j} k^L((\boldsymbol{X}^n \oplus \boldsymbol{Z}^n)_{q,:}, (\boldsymbol{X}^n \oplus \boldsymbol{Z}^n)_{p,:})$ is bounded.

**Corollary 7.2** *If the conditions (a)-(d) in Theorem 7.1 hold, there exits a constant $c_2 \in \mathbb{R}^+$ such that $\nabla_{z_j} k^L(\boldsymbol{a}, \boldsymbol{b}) \le c_2$ for any two rows $\boldsymbol{a}$ and $\boldsymbol{b}$ of $\boldsymbol{X}^n \oplus \boldsymbol{Z}^n$.*

It is clear that $\nabla_{z_j} k^L(\boldsymbol{a}, \boldsymbol{b}) = 0$ if $\boldsymbol{a}, \boldsymbol{b} \ne \boldsymbol{z}$. So, without loss of generality, we consider $\nabla_{z_j} k^L(\boldsymbol{a}, \boldsymbol{z})$ only. From Eq. (13), we have

$$\frac{\partial k^L(\boldsymbol{a}, \boldsymbol{z})}{\partial z_j} = \frac{\partial k^L(\boldsymbol{a}, \boldsymbol{z})}{\partial k^{L-1}(\boldsymbol{a}, \boldsymbol{z})} \frac{\partial k^{L-1}(\boldsymbol{a}, \boldsymbol{z})}{\partial k^{L-2}(\boldsymbol{a}, \boldsymbol{z})} \cdots \frac{\partial k^1(\boldsymbol{a}, \boldsymbol{z})}{\partial z_j}.$$

For each $l = 2, \cdots, L$, we can bound $\partial k^l(\boldsymbol{a}, \boldsymbol{z})/\partial k^{l-1}(\boldsymbol{a}, \boldsymbol{z})$ by

$$
\begin{aligned}
\frac{\partial k^l(\boldsymbol{a}, \boldsymbol{z})}{\partial k^{l-1}(\boldsymbol{a}, \boldsymbol{z})} &= \sigma_w^2 \mathbb{E}_{(h_j^{(l-1)}(\boldsymbol{x}), h_j^{(l-1)}(\boldsymbol{x}')) \sim \mathcal{N}(\boldsymbol{0}^2, \tilde{\boldsymbol{K}}^{l-1})} \left[ \phi'(h_j^{(l-1)}(\boldsymbol{x})) \phi'(h_j^{(l-1)}(\boldsymbol{x}')) \right] \\
&\le (\sigma_w \max_h \phi'(h))^2
\end{aligned}
$$

provided that the maximum slope of $\phi$ is limited, which is true for many popular activation functions including ReLU and erf. Also, by Eq. (14), the value

$$\frac{\partial k^1(\boldsymbol{a}, \boldsymbol{z})}{\partial z_j} = \frac{\sigma_w^2}{d} a_j$$

is bounded. Therefore, Corollary 7.2 holds, which in turn makes $\mathcal{L}(\boldsymbol{Z}^n)$ $\beta$-smooth via Corollary 7.1. By Lemma 7.1, we obtain the proof of Theorem 7.1.

## 8 EXPERIMENT SETTINGS

This section provides more details about the settings of our experiments.

### 8.1 MODEL SETTINGS

The network architectures of the baseline GANs used in our experiments are based on InfoGAN (Chen et al., 2016). We set the latent dimensions, training iterations, and batch size according to the study (Lucic et al., 2018). The latent dimensions for the generator are all 64. The batch size for all baselines is set to 64. The training iterations are 80K, 100K, and 400K for MNIST, CelebA, and CIFAR-10 datasets, respectively. For the optimizers, we follow the setting from the respective original papers. Below we list the network architecture of the baselines for each dataset as well as the optimizer settings.

Table 3: The architectures of the discriminator and generator in the baseline GANs for the MNIST dataset.

| Discriminator | Generator |
|---|---|
| Input 28×28×1 Gray image | Input$\in \mathbb{R}^{64} \sim \mathcal{N}(\boldsymbol{0}, \boldsymbol{I})$ |
| 4×4 conv; 64 leaky ReLU; stride 2 | Fully Connected 1024 ReLU; batchnorm |
| 4×4 conv; 128 leaky ReLU; stride 2. batchnorm | Fully Connected $7 \times 7 \times 128$ ReLU; batchnorm |
| Fully Connected 1024 leaky ReLU; batchnorm | 4×4 deconv; 64 ReLU. stride 2; batchnorm |
| Fully Connected 1 output | 4×4 deconv; 1 sigmoid |

Table 4: The architectures of the discriminator and generator in the baseline GANs for the CIFAR-10 dataset.

| discriminator | generator |
|---|---|
| Input 32×32×3 Image | Input$\in \mathbb{R}^{64} \sim \mathcal{N}(\mathbf{0}, \boldsymbol{I})$ |
| 4×4 conv; 64 leaky ReLU; stride 2 | Fully Connected $2 \times 2 \times 448$ ReLU; batchnorm |
| 4×4 conv; 128 leaky ReLU; stride 2; batchnorm | 4×4 deconv; 256 ReLU; stride 2; batchnorm |
| 4×4 conv; 256 leaky ReLU; stride 2; batchnorm | 4×4 deconv; 128 ReLU; stride 2 |
| Fully Connected 1 output | 4×4 deconv; 64 ReLU; stride 2 |
| | 4×4 deconv; 3 Tanh; stride 2. |

Table 5: The architectures of the discriminator and generator in the baseline GANs for the CelebA dataset.

| discriminator | generator |
|---|---|
| Input 64×64×3 Image | Input$\in \mathbb{R}^{64} \sim \mathcal{N}(\mathbf{0}, \boldsymbol{I})$ |
| 4×4 conv; 64 leaky ReLU; stride 2 | Fully Connected $2 \times 2 \times 448$ ReLU; batchnorm |
| 4×4 conv; 128 leaky ReLU; stride 2; batchnorm | 4×4 deconv; 256 ReLU; stride 2; batchnorm |
| 4×4 conv; 256 leaky ReLU; stride 2; batchnorm | 4×4 deconv; 128 ReLU; stride 2 |
| 4×4 conv; 256 leaky ReLU; stride 2; batchnorm | 4×4 deconv; 64 ReLU; stride 2 |
| Fully Connected 1 output | 4×4 deconv; 32 ReLU; stride 2 |
| | 4×4 deconv; 3 Tanh; stride 2. |

Table 6: The optimizer settings for each GAN baseline. $n_{dis}$ denotes the training steps for discriminators in the alternative training process.

| | Optimizer type | Learning Rate | $\beta_1$ | $\beta_2$ | $n_{dis}$ |
|---|---|---|---|---|---|
| DCGAN | Adam | 0.0002 | 0.5 | 0.999 | 1 |
| LSGAN | Adam | 0.0002 | 0.5 | 0.999 | 1 |
| WGAN | RMSProp | 0.00005 | None | None | 5 |
| WGAN-GP | Adam | 0.0001 | 0.5 | 0.9 | 5 |
| SN-GAN | Adam | 0.0001 | 0.9 | 0.999 | 5 |

Note that we remove all the batchnorm layers for the discriminators in WGAN-GP. We architect the element network of the discriminator in our GA-NTK following InfoGAN (Chen et al., 2016), except that the width (or the number of filters) of the network is infinite at each layer and has no batchnorm layers.

The generator of GA-NCTKg consumes memory. To reduce memory consumption, we let $\mathcal{D}$ discriminates true and fake images in the code space of a pre-trained autoencoder $\mathcal{A}$ (Bergmann et al., 2019). After training, a code output by $\mathcal{G}$ is fed into the decoder of $\mathcal{A}$ to obtain an image. The architectures of the pre-trained $\mathcal{A}$ for different datasets are summarized as follows:

Table 7: The architectures of $\mathcal{A}$ for different datasets.

| MNIST | CIFAR-10 |
|---|---|
| Input 28×28×1 Image | Input 32×32×3 Image |
| 3×3 conv; 16 SeLU; stride 2 | 3×3 conv; 32 SeLU; stride 2 |
| 3×3 conv; 32 SeLU; stride 2 | 3×3 conv; 64 SeLU; stride 2 |
| 3×3 conv; 64 SeLU; stride 2 | 3×3 conv; 128 SeLU; stride 2 |
| Fully Connected; 128 tanh | Fully Connected; 1024 tanh |
| 3×3 transposeconv; 64 SeLU; stride 2 | 3×3 transposeconv; 128 SeLU; stride 2 |
| 3×3 transposeconv; 32 SeLU; stride 2 | 3×3 transposeconv; 64 SeLU; stride 2 |
| 3×3 transposeconv; 16 SeLU; stride 2 | 3×3 transposeconv; 32 SeLU; stride 2 |
| output | output |

| CelebA | CelebA-HQ |
|---|---|
| Input 64×64×3 Image | Input 256×256×3 Image |
| 3×(3×3 conv; 32 SeLU; stride 1) | 3×(3×3 conv; 64 SeLU; stride 1) |
| 3×3 conv; 32 SeLU; stride 2 | 3×3 conv; 64 SeLU; stride 2 |
| 3×(3×3 conv; 64 SeLU; stride 1) | 3×(3×3 conv; 128 SeLU; stride 1) |
| 3×3 conv; 64 SeLU; stride 2 | 3×3 conv; 128 SeLU; stride 2 |
| 3×(3×3 conv; 128 SeLU; stride 1) | 3×(3×3 conv; 256 SeLU; stride 1) |
| 3×3 conv; 128 SeLU; stride 2 | 3×3 conv; 256 SeLU; stride 2 |
| Fully Connected; 2048 tanh | 3×(3×3 conv; 512 SeLU; stride 1) |
| 3×(3×3 transposeconv; 128 SeLU; stride 1) | 3×3 conv; 512 SeLU; stride 2 |
| 3×3 transposeconv; 128 SeLU; stride 2 | Fully Connected; 2048 tanh |
| 3×(3×3 transposeconv; 64 SeLU; stride 1) | 3×(3×3 transposeconv; 512 SeLU; stride 1) |
| 3×3 transposeconv; 64 SeLU; stride 2 | 3×3 transposeconv; 512 SeLU; stride 2 |
| 3×(3×3 transposeconv; 32 SeLU; stride 1) | 3×(3×3 transposeconv; 256 SeLU; stride 1) |
| 3×3 transposeconv; 32 SeLU; stride 2 | 3×3 transposeconv; 256 SeLU; stride 2 |
| output | 3×(3×3 transposeconv; 128 SeLU; stride 1) |
| | 3×3 transposeconv; 128 SeLU; stride 2 |
| | 3×3(3×3 transposeconv; 64 SeLU; stride 1) |
| | 3×3 transposeconv; 64 SeLU; stride 2 |
| | output |

## 8.2 METRICS

The FID scores are computed using the code from the original paper (Heusel et al., 2017). We sample 2048 images to compute the FID scores. We calculate the AM-SSIM scores using the SSIM settings: filter size 4, filter sigma 1.5, $k_1$ 0.01, and $k_2$ 0.03 (Wang et al., 2004).

---

**Algorithm 1** Unidirectional search for the hyperparameter $\lambda$ of GA-NTK.

---

**Input:** Data $\boldsymbol{X}^n$, kernel $k$, and separation tolerance $\epsilon$
**Output:** $\lambda$ for GA-NTK

Randomly initiate $\boldsymbol{Z}^n \in \mathbb{R}^{n \times d}$
$\lambda \leftarrow 1$
**while** $\frac{1}{2n}\|\mathcal{D}(\boldsymbol{X}^n, \boldsymbol{Z}^n; k, \lambda) - (\mathbf{1}^n \oplus \mathbf{0}^n)\|^2 \leq \epsilon$ **do**
$\quad | \quad \lambda \leftarrow \lambda \cdot 2$
**end**
**return** $\lambda$

---

### 8.3 HYPERPARAMETER TUNING

For each data synthesis method, we tune its hyperparameter using grid search. **GA-NTK.** The computation of $\boldsymbol{K}^{2n,2n}$ requires one to determine the initialization and architecture of the element networks in the ensemble discriminator. Poole et al. (2016); Schoenholz et al. (2017); Raghu et al. (2017) have proposed a principled method to tune the hyperparameters for the initialization. From our empirical results, we also find that the quality of the images generated by GA-NTK is not significantly impacted by the choice of the architecture—a fully connected network with rectified linear unit (ReLU) activation suffices to generate recognizable image patterns. Once $\boldsymbol{K}^{2n,2n}$ is decided, there is only one hyperparameter $\lambda = \eta t$ to tune in Eq. (16). The $\lambda$ controls how well the discriminator is trained on $\mathbb{D}$, so either a too small or large value can lead to poor gradients for $\boldsymbol{Z}^n$ and final generated points. But since there is no alternating updates as in GANs, we can decide an appropriate value of $\lambda$ without worrying about canceling the learning progress of $\boldsymbol{Z}^n$. We propose a simple, unidirectional search algorithm for tuning $\lambda$, as shown in Algorithm 1. Basically, we search, from small to large, for a value that makes the discriminator nearly separate the real data from pure noises in an auxiliary learning task, and then use this value to solve Eq. (16). In practice, a small positive $\epsilon$ ranging from $10^{-3}$ to $10^{-2}$ suffices to give an appropriate $\lambda$. **Multi-resolutional GA-NTK.** We use 3 NTK-GP's as the discriminators, whose architectures are listed in Table 8.

Table 8: The architectures of the discriminators for multi-resolution GA-NTK.

| Discriminator small | Discriminator medium | Discriminator large |
|---|---|---|
| Input 16×16×3 Image | Input 64×64×3 Image | Input 256×256×3 Image |
| 4×4 conv; ReLU; stride 2 | 4×4 conv; ReLU; stride 2 | 4×4 conv; ReLU; stride 2 |
| 4×4 conv; ReLU; stride 2 | 4×4 conv; ReLU; stride 2 | 4×4 conv; ReLU; stride 2 |
| Fully Connected 1 output | 4×4 conv; ReLU; stride 2 | 4×4 conv; ReLU; stride 2 |
| | 4×4 conv; ReLU; stride 2 | 4×4 conv; ReLU; stride 2 |
| | Fully Connected 1 output | 4×4 conv; ReLU; stride 2 |
| | | 4×4 conv; ReLU; stride 2 |
| | | Fully Connected 1 output |

## 9 MORE EXPERIMENTS

### 9.1 GA-FNTK VS. GA-CNTK

Next, we compare the images generated by GA-FNTK, GA-CNTK, and the multi-resolutional GA-CNTK described in Section 3.2 on the CelebA and CelebA-HQ datasets. The multi-resolutional GA-CNTK employs 3 discriminators working at $256 \times 256$, 64×64, and 16×16 pixel resolutions, respectively. Figure 6 shows the results. To our surprise, GA-NTK (which models the discriminator as an ensemble of fully connected networks) suffices to generate recognizable faces. The images synthesized by GA-FNTK and GA-CNTK lack details and global coherence, respectively, due to

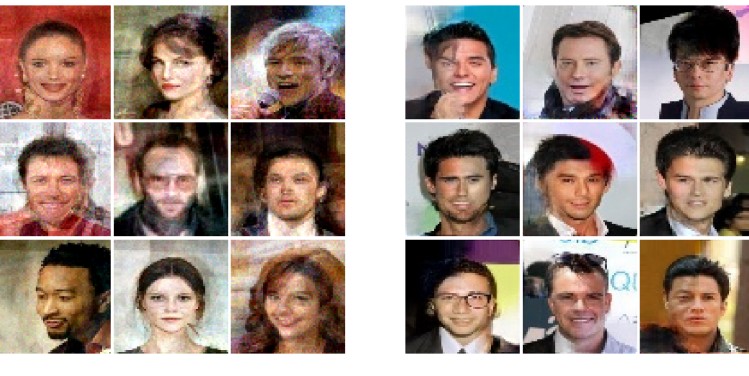

(a) GA-FNTK on CelebA        (b) GA-CNTK on CelebA

Figure 6: The images generated by (a) GA-FNTK and (b) GA-CNTK given 256 CelebA training images.

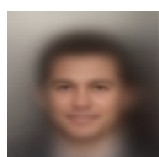

Figure 7: When $b = 1$, GA-NTK tends to generate a blurry mean image.

the characteristics of FNNs and CNNs. On the other hand, the multi-resolutional GA-CNTK gives both the details and global coherence thanks to the multiple discriminators working at different pixel resolutions. The results also demonstrate the potential of GA-NTK variants to generate high-quality data as there are many other techniques for GANs that could be adapted into GA-NTK.

## 9.2 BATCH-WISE GA-NTK

To work with a larger training set, we modify GA-CNTK by following the instructions in Section 3.2 to obtain the batch-wise GA-CNTK, which computes the gradients of $\boldsymbol{Z}^n$ in Eq. (4) from 256 randomly sampled training images during each gradient descent iteration. We train the batch-wise GA-CNTK on two larger datasets consisting of 2048 images from CelebA and 1300 images from ImageNet, respectively. Figure 8 shows the results, and the batch-wise GA-CNTK can successfully generate the "daisy" images on ImageNet.

Note that the batch-wise GA-CNTK solves a different problem than the original GA-CNTK—the former finds $\boldsymbol{Z}^n$ that deceives *multiple* discriminators, each trained on 256 examples, while the latter searches for $\boldsymbol{Z}^n$ that fools a single discriminator trained on 256 examples. We found that, when the batch size is small ($b = 1$), GA-NTK tends to generate a blurry mean image regardless of model architectures and initializations of model weights and $\boldsymbol{Z}^n$, as shown in Figure 7. This is because the mean image is the best for simultaneously fooling many NTK discriminators, each trained on a single example. However, in practice this setting is less common as one usually aims to use the largest $b$ possible (Brock et al., 2018). Figure 8 shows that a batch size of 256 suffices to give plausible results on the CelebA and ImageNet datasets. Comparing the images in Figure 1(f) with those in Figure 8(a), we can see that the batch-wise GA-CNTK gives a little more blurry images but the patterns in each synthesized image are more globally coherent, both due to the effect of multiple discriminators.

## 9.3 SENSITIVITY TO HYPERPARAMETERS

Here, we study how sensitive is the performance of WGAN, WGAN-GP, and GA-FNTK to their hyperparameters. We adjust the hyperparameters of different approaches using the grid search under a time budget of 3 hours, and then evaluate the quality of 2048 generated data points by the Wasser-

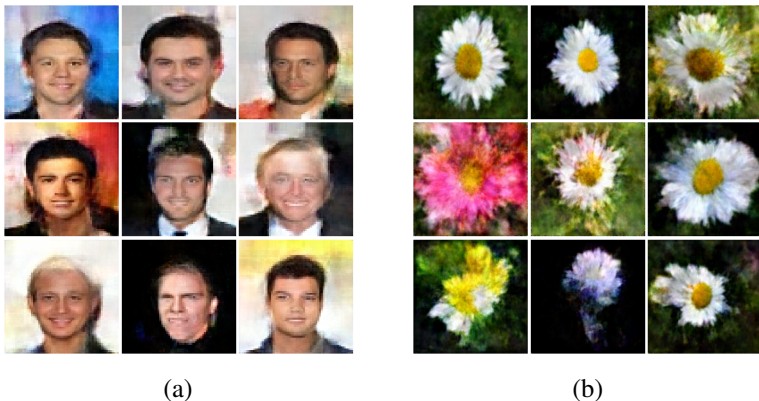

(a)               (b)

Figure 8: The images generated by batch-wise GA-CNTK on (a) CelebA dataset of 2048 randomly sampled images and (b) ImageNet dataset of 1300 randomly sampled images.

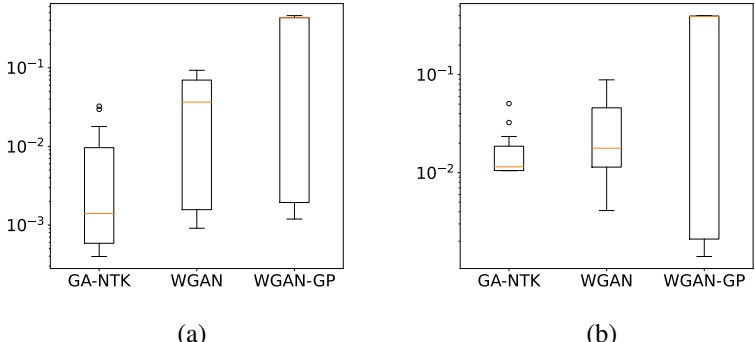

(a)               (b)

Figure 9: The distribution of Wasserstein distance between $\mathcal{P}_{\text{gen}}$ and $\mathcal{P}_{\text{data}}$ (used to measure the quality of the generated points) over the searched hyper-parameters on training sets of (a) 8- and (b) 25-modal Gaussian mixtures.

stein distance between $\mathcal{P}_{\text{gen}}$ and $\mathcal{P}_{\text{data}}$. We train different methods on two toy datasets consisting of 8- and 25-modal Gaussian mixtures following the settings described in Section 4.2. Figure 9 shows the results, and we can see that GA-FNTK achieves the lowest average Wasserstein distance in both cases. Moreover, its variances are smaller than the two other baselines, too. This shows that the performance of GA-FNTK is less sensitive to the hyperparameters and could be easier to tune in practice.

Note that, with 3-hour time budget, the hyperparameters we obtained through the grid search are good enough for reproducing the experiments conducted by Mao et al. (2017) on mode collapse. In the experiments, the $\mathcal{P}_{\text{gen}}$ of different methods aim to align a 2D 8-modal Gaussian mixtures in the ground truth. Our results are shown in Figure 10.

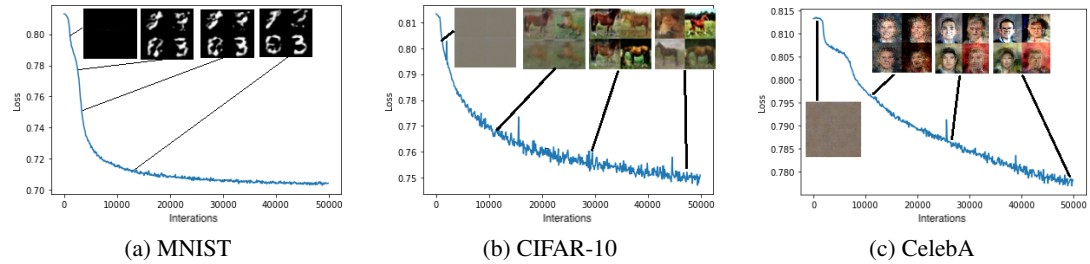

Figure 11: The learning curve of $\mathcal{G}$ in GA-CNTKg and the generated images $\mathcal{G}(z)$ at different stages of training given the same input $z$.

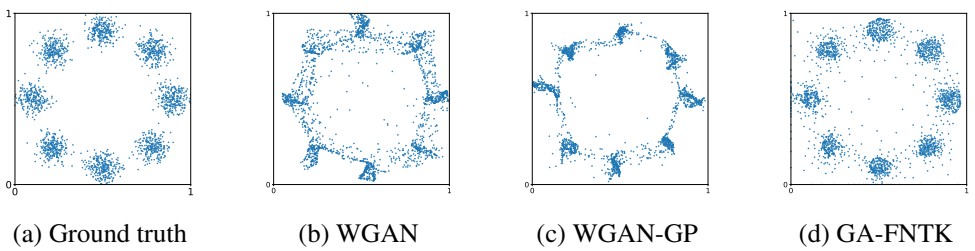

Figure 10: Visualization of distribution alignment and mode collapse on a 2D 8-modal Gaussian mixtures dataset.

## 9.4 Evolution of Images during Training

Figure 11 shows the learning curve of the generator in $\mathcal{G}$ in GA-CNTKg and the relationship between the quality of images output by $\mathcal{G}$ and the number of gradient descent iterations. The results show that the loss can be minimized even if it is an $f$-divergence, and a lower loss score implies higher image quality. This is consistent with the results of GA-CNTK (without a generator) shown in Figure 3.

**Source of creativity.** The diversity of our generated data not only comes from the randomness of an optimization algorithm (e.g., initialization of $Z$ or splitting of $X$ into batches, as discussed in Section 3.2) but also from the objective in Eq. (4) itself. To see this, observe in Figure 3 that the images generated at the later stage of training contain recognizable patterns that change constantly over training time, despite little change in the loss score. The reason is that, in Eq. (4), the $Z^n$ is optimized for a *moving* target—any change of $Z^n$ causes $\mathcal{D}$ to be "retrained" instantly. The training of the generator $\mathcal{G}$ in Eq. (6) also shares this nice property. In Figure 11, the patterns of a generated image $\mathcal{G}(z)$ change over training time even when the input $z$ is fixed. However, getting diverse artificial data through this property requires prolonged training time. In practice, we can simply initialize $Z$ differently to achieve diversity faster.

## 10 More Images Generated by GA-CNTK and GA-CNTKg

Figures 12–16 show more sample images synthesized by GA-CNTK and GA-CNTKg. All these images are obtained using the settings described in the main paper and the above.

We can see that the quality of the images synthesized by GA-CNTKg is worse than that of the images synthesized by GA-CNTK, as discussed in Section 4.1. Furthermore, recall from Table 1 that, without a generator network, the GA-NTK performs better when the date size increases. However, this is not the case for GA-NTKg having a generator network. We have resampled training data and rerun the experiments 5 times with different initial values of $Z^n$ but obtained similar results. Therefore, we believe the instability is due to the sample complexity of the generator network—256 examples or less are insufficient to train a stable, high-quality generator. This is evident in Figures 12(b)-15(b) where the generator outputs unrecognizable images more often.

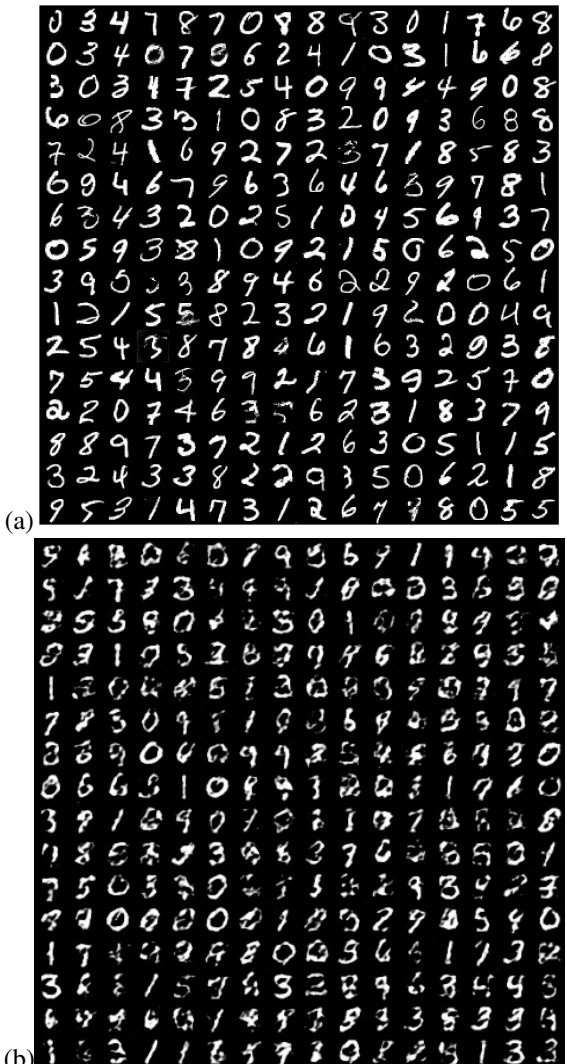

Figure 12: Sample images generated by GA-CNTK (a) without and (b) with generator on the MNIST dataset of 256 randomly sampled images.

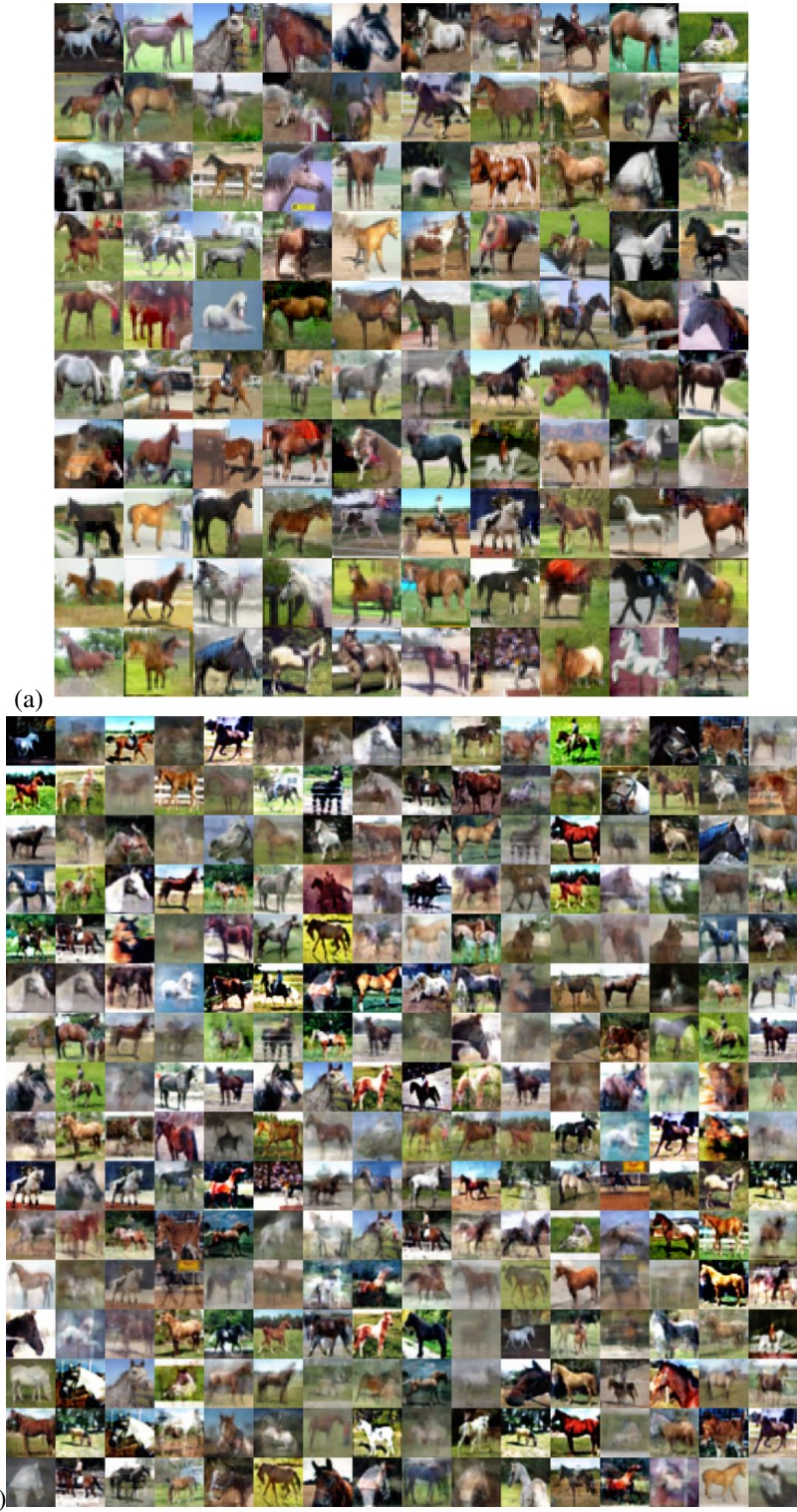

(a)

(b)

Figure 13: Sample images generated by GA-CNTK (a)without generator(b)with generator on the CIFAR-10 dataset of 256 randomly sampled images.

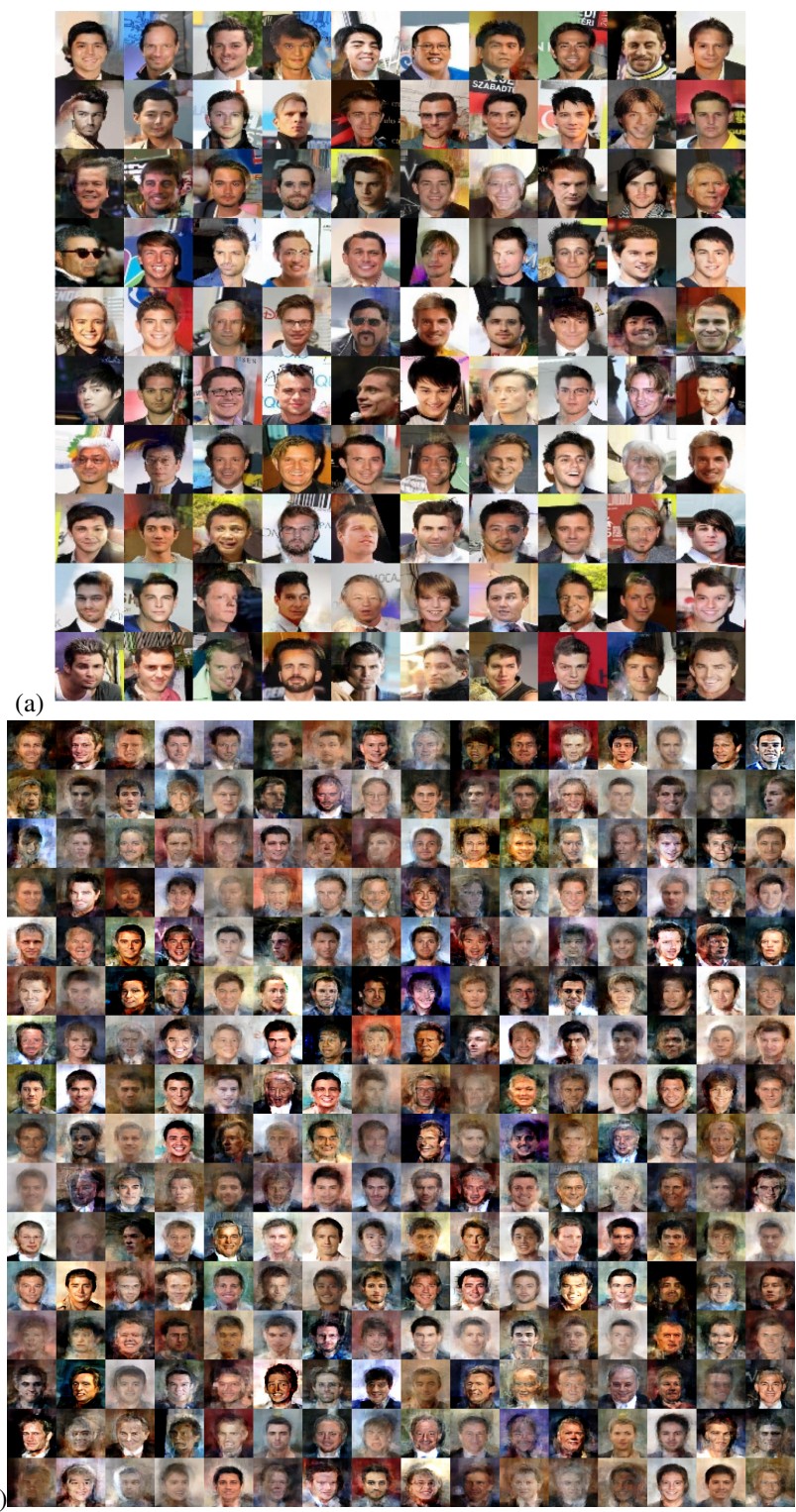

(a)

(b)

Figure 14: Sample images generated by GA-CNTK (a) without and (b) with generator on the CelebA dataset of 256 randomly sampled images.

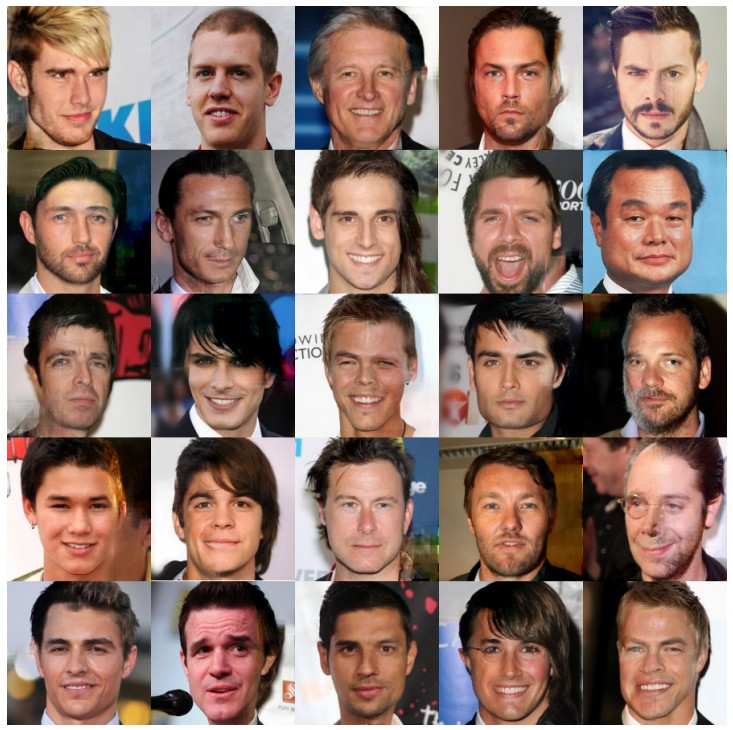

Figure 15: Sample images generated by multi-resolutional GA-CNTK on the CelebA-HQ dataset of 256 randomly sampled images.

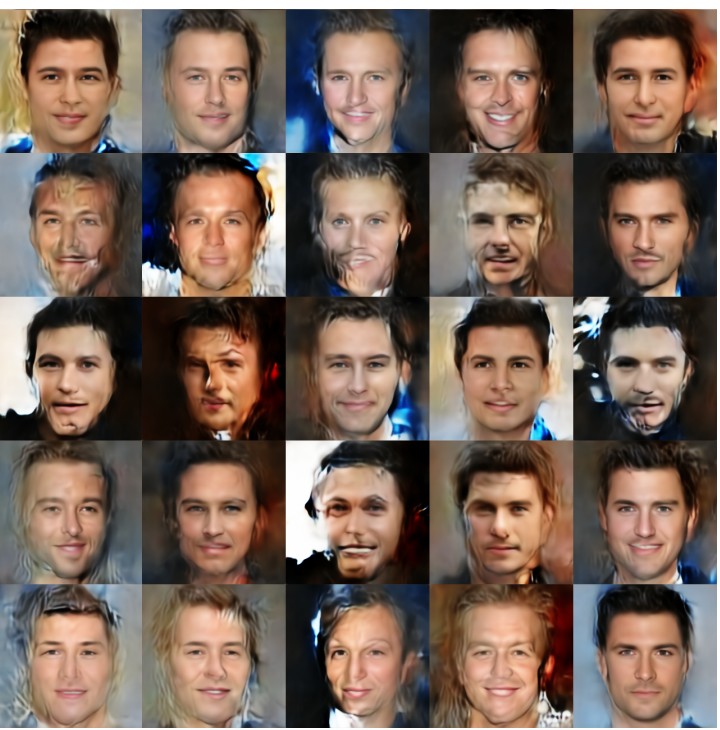

Figure 16: Sample images generated by GA-CNTKg on the CelebA-HQ dataset of 256 randomly sampled images.

## 11 DOWNGRADE IMAGES

As discussed in the main paper, we find that, when the size of training set is small, an image synthesis method may produce downgrade images that look almost identical to some images in the training set. This problem is less studied in the literature but important to applications with limited training data. We investigate this problem by showing the images from the training set that are the nearest to a generated image. We use the SSIM (Wang et al., 2004) as the distance measure. Figures 17, 18, and 19 show the results for some randomly sampled synthesized images. As compared to GANs, both GA-CNTK and batch-wise GA-CNTK can generate images that look less similar to the ground-truth images.

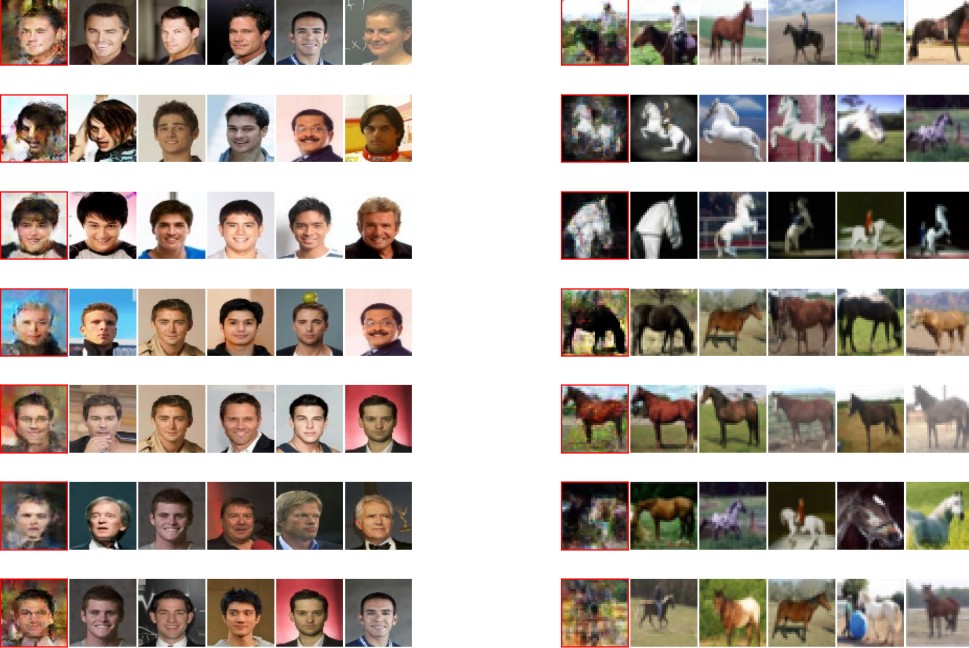

Figure 17: Comparison between the images generated by WGAN-GP trained on 256 images and the nearest neighbors (measured by SSIM) from the training set. Images with red bounding boxes are generated images.

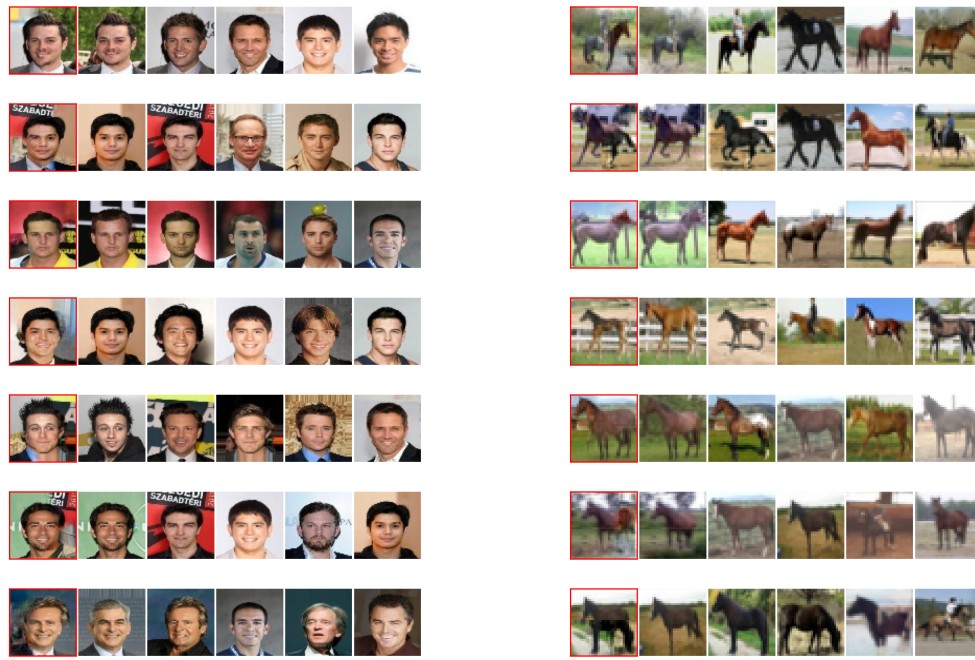

Figure 18: Comparison between the images generated by GA-CNTK trained on 256 images and the nearest neighbors (measured by SSIM) from the training set. Images with red bounding boxes are generated images.

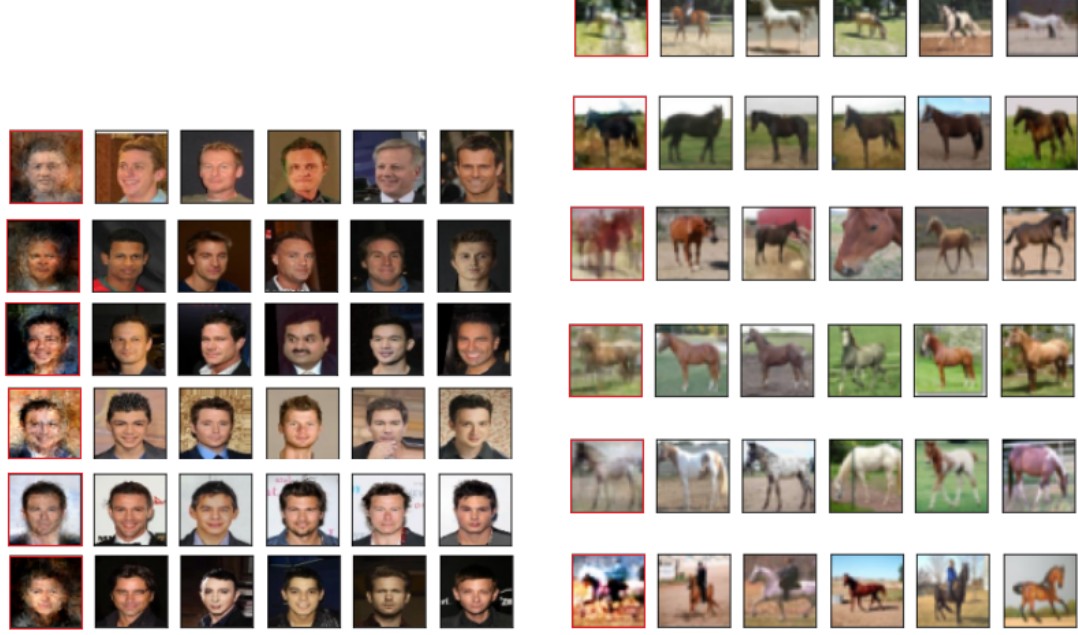

Figure 19: Comparison between the images generated by GA-CNTKg trained on 256 images and the nearest neighbors (measured by SSIM) from the training set. Images with red bounding boxes are generated images.

Table 9: The convergence speed and training time of different methods on a machine with a single NVIDIA Tesla V100 GPU given different datasets of 256 randomly sampled images. The GA-CNTK and GA-CNTKg are batch-wise, and the batch size $b$ is set to 64 for all methods.

| | Metric | DCGAN | LSGAN | WGAN | WGANGP | SNGAN | GACNTK | GACNTKg |
|---|---|---|---|---|---|---|---|---|
| **MNIST** | Iterations | 7400 | 5100 | 7000 | 3400 | 12800 | 500 | 1600 |
| | Iter. / sec. | 20 | 19 | 19 | 18 | 18 | 14 | 9 |
| | Seconds | 370 | 268 | 368 | 189 | 711 | 35 | 177 |
| **CIFAR-10** | Iterations | N/A | N/A | 14000 | 11100 | N/A | 600 | 6200 |
| | Iter. / sec. | 17 | 17 | 16 | 15 | 14 | 13 | 8 |
| | Seconds | N/A | N/A | 875 | 740 | N/A | 46 | 775 |
| **CelebA** | Iterations | N/A | N/A | 18800 | 11200 | N/A | 1200 | 5900 |
| | Iter. / sec. | 13 | 12 | 12 | 10 | 9 | 6 | 5 |
| | Seconds | N/A | N/A | 1566 | 1120 | N/A | 20 | 1180 |

## 12   SEMANTICS LEARNED BY GA-CNTKG

Here, we investigate whether the features learned by GA-NTK can encode high-level semantics. We plot "interpolated" images output by the generator $\mathcal{G}$ of GA-CNTKg taking equidistantly spaced $z$'s along a segment in $z$ space as the input. For ease of presentation, we consider a 2-dimensional $z$ space and train $\mathcal{G}$ on MNIST and CelebA datasets of 256 examples. Figure 20 shows the results, where the generated patterns transit smoothly across the 2D $z$ space, and neighboring images share similar looks. These similar-looking images are generated from adjacent but meaningless $z$'s, suggesting that the learned features encode high-level semantics.

## 13   CONVERGENCE SPEED AND TRAINING TIME

In this section, we study the time usage for training GA-NTK variants and compare it with the training of GANs. We conduct experiments to investigate the number of iterations and the wall-clock time required to train different methods on different datasets of 256 randomly sampled images. We use the batch-wise GA-CNTK and GA-CNTKg and set the batch size $b$ to 64 for all methods. We run the experiments on a machine with a single NVIDIA Tesla V100 GPU. For DCGAN and LSGAN whose loss scores do not reflect image quality, we monitor the training process manually and stop it as long as the generated images contain recognizable patterns. But these methods do not seem to converge. For other methods, we use the early-stopping with the patience of 10000 steps and delta of 0.05 to determine convergence. The results are shown in Table 9. As we can see, the number of iterations required by either batch-wise GA-CNTK or GA-CNTKg is significantly smaller than that used by GANs. This justifies our claims in Section 1. However, the batch-wise GA-CNTK and GA-CNTKg run fewer iterations per second than GANs because of the higher computation cost involved in back-propagating through $\boldsymbol{K}^{b,b}$. In terms of wall-clock time, the batch-wise GA-CNTK is the fastest while the GA-CNTKg runs as fast as WGAN-GP. We expect that, with the continuous optimization of the Neural Tangents library (Novak et al., 2019a) which our code is based on, the training speed of GA-NTK variants can be further improved.

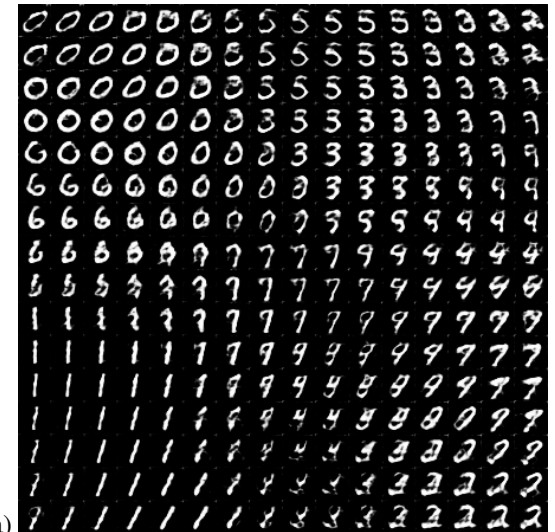

(a)

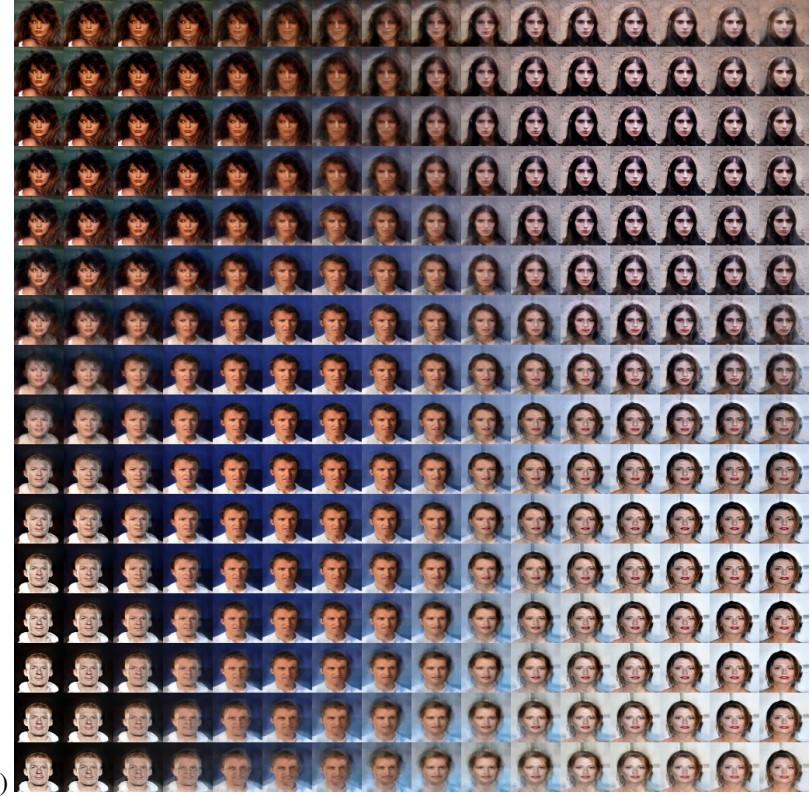

(b)

Figure 20: Interpolated images generated by GA-CNTKg, which is trained on (a) MNIST and (b) CelebA datasets of 256 randomly sampled examples from all classes. The $\mathcal{G}$ takes 2-dimensional $z$'s as input. For each dataset, we feed equidistantly spaced $z$'s along a segment in $z$ space to $\mathcal{G}$ to get the interpolated images.

