# OpenReview forum: "Single-level Adversarial Data Synthesis based on Neural Tangent Kernels"
_ICLR.cc/2023/Conference — Submitted to ICLR 2023_

### Official Review · Reviewer_qHF8 · 2022-10-23

**Confidence:** 3
**Correctness:** 3
**Technical Novelty And Significance:** 2
**Empirical Novelty And Significance:** 2
**Recommendation:** 3

**Clarity, Quality, Novelty And Reproducibility:**

- The paper's writing has good clarity. The method is novel as it provides a different training regime by introducing a new class of discriminators.
- code and network architectures are provided to improve reproducibility.

**Strength And Weaknesses:**

*Strengths*
- The motivation of the proposed method is clearly stated. The mathematical settings of the proposed method are clearly stated. The overall writing is fluent and easy to understand.
- Numerical experiments of generating new samples with the method are conducted. Also, the result is consistently evaluated against state-of-the-art methods.

*Weaknesses*
- The statement of the method could be made better if the training process of the proposed method is more clearly stated. In particular, I'm confused how the proposed discriminator is trained (or not trained) during the overall training process, e.g. what are the variables to be updated during training?
- Since the main advantage of the proposed method is to substitute traditional neural network-based discriminator with kernel methods, it would be plausible to include the comparison of the discriminator performance between the two methods.
- The scalability of the proposed method is one of the drawbacks of the proposed method. In particular, the proposed discriminator has a space complexity of O(n^2) where n is the number of data points.
- The paper makes the assumption that mode collapse is solely caused by min-max training regime in traditional GAN methods. But mode collapse can be caused by other factors as well, such as the structure of generators. A more thorough study of the causal relationship between min-max training and mode collapse could be plausible to provide.
- In the experiment part, generated images and their most similar images from the dataset are provided. It looks to me that the GA-CNTK method tries to memorize the training samples and the pixel level and not able to learn high-level semantics from the images.
- typo on page 4: "my be desirable" -> "might be desirable"

**Summary Of The Paper:**

This paper proposed a novel generative model GA-NTK that mitigates the drawbacks of traditional GANs trained with the alternative SGD method. This is achieved by using a closed-form discriminator based on a neural tangent kernel (NTK) instead of a neural network. The author proved the convergence of the proposed model during the training phase. Also, numerical experiments are conducted to show the efficacy of the proposed method.

**Summary Of The Review:**

Overall the paper provides a new angle of view for the GAN models. But there are still some points the paper needs to address to meet the bar of acceptance.

---

> ### Author Response · Authors · 2022-11-19
> **Reply to Reviewer qHF8 (Part 1 of 2)**
>
> Thank you for your valuable comments and suggestions. We appreciate that you find the proposed GA-NTK method interesting and novel, most parts of the paper well-written, our experiments convincing and reproducible, and our mathematical settings clear. We would like to point you to **[our general message to all reviewers](https://openreview.net/forum?id=_d2f3hRn0hT&noteId=jv7yTAeewGL)**. In what follows we reply to your particular comments.
>
> > I'm confused how the proposed discriminator is trained (or not trained) during the overall training process, e.g. what are the variables to be updated during training?
>
> Sorry for not making this clear enough. The discriminator is a closed form formula which is instantly “trained” once the training data $\boldsymbol{X}^{n}\oplus\boldsymbol{Z}^{n}$ change. It has no weights but a hyperparameter $\lambda$ that needs to be tuned before the training of $\boldsymbol{Z}^{n}$. We discuss in Section 3.1 that the value of $\lambda$ should be large enough but finite. Therefore, we search for the value that makes the discriminator nearly separate the real data from pure noises in an auxiliary learning task. The complete training procedure is as follows:
>
> 1. Tune the hyperparameter $\lambda$ by using an auxiliary task asking the discriminator to separate real data from pure noises, as described in Algorithm 1;
> 2. Fix $\lambda$ and solve $\boldsymbol{Z}^{n}$ in Eq. (4) by ordinary gradient descent.
>
> We have added a paragraph right before Section 3.1 in the revision to describe the complete training procedure. More details of step 1 can be found in Section 8.3.
>
> > It would be plausible to include the comparison of the discriminator performance between the two methods.
>
> Thanks for the great suggestion. The comparison between the performance of NTK-GPs and their finite-width counterparts (neural networks) has been conducted in the literature [1-4]. In summary, NTK-GPs perform similarly to neural networks in many situations and sometimes even better on small-data tasks [3-4]. We have added the above discussion to Section 2.2.
>
> [1] Novak et al., "Neural Tangents: Fast and Easy Infinite Neural Networks in Python," in Proc. of ICLR, 2019.
> [2] Lee et al., "Finite Versus Infinite Neural Networks: an Empirical Study," in Proc. of NeurIPS, 2020.
> [3] Arora et al., "Harnessing the Power of Infinitely Wide Deep Nets on Small-data Tasks," in Proc. of ICLR, 2020.
> [4] Geifman et al, "On the similarity between the laplace and neural tangent kernels," in Proc. of NeurIPS, 2020.
>
> > The proposed discriminator has a space complexity of $O(n^{2})$ where $n$ is the number of data points.
>
> This problem is rooted in NTKs, and many recent efforts [5-8] have been made to reduce their time/space complexity. Integrating these works into GA-NTK is an interesting direction to explore in the future. Meanwhile, we propose the batch-wise variant of GA-NTK, as described in Section 3.2, to addresses the scalability issue from the optimization perspective. In this variant, the $n$ represents the *batch size* instead of the size of an entire dataset. We show in Figure 8 that a batch size of 256 suffices to train a batch-wise GA-CNTK on larger CelebA and ImageNet datasets of 2048 and 1300 examples, respectively. We also show that one can train a generator $\mathcal{G}$ in GA-CNTKg with batches of size 256 and then, at inference time, use $\mathcal{G}$ to generate data regardless of the time/space limitations of NTKs.
>
> [5] Arora et al., "On Exact Computation with an Infinitely Wide Neural Net," in Proc. of NeurIPS, 2019.
> [6] Bietti et al., "On the Inductive Bias of Neural Tangent Kernels," in Proc. of NeurIPS, 2019.
> [7] Han et al., "Random Features for the Neural Tangent Kernel," CoRR abs/2104.01351, 2021.
> [8] Zandieh et al., "Scaling neural tangent kernels via sketching and random features," in Proc. of NeurIPS, 2021.
>
> > Mode collapse can be caused by other factors as well, such as the structure of generators.
>
> Thanks. We totally agree with you and have modified our claim in Section 1 to that GA-NTK avoids only the mode collapse problem caused by the alternating SGD.
>
> To be continued...

---

> > ### Author Response · Authors · 2022-11-19
> > **Reply to Reviewer qHF8 (Part 2 of 2)**
> >
> > > From figures in Section 11... It looks to me that the GA-CNTK method tries to memorize the training samples at the pixel level and not able to learn high-level semantics from the images.
> >
> > Thanks for this constructive comments. The purpose of Figures 17-19 in Section 11 is to demonstrate the downgrade effect under the limitation of training set size (256 examples). When the number of training examples increases to 2048, the memorization effect can be largely alleviated, as Figure 7 shows. Moreover, the images in Figures 17-19 are randomly sampled and may be far away from each other in the feature space, making it hard to evaluate whether the learned features encode high-level semantics. Hence, we investigate the “interpolated” images output by the generator $\mathcal{G}$ of GA-CNTKg taking equidistantly spaced $\boldsymbol{z}$'s along a segment in $\boldsymbol{z}$ space as the input. For ease of presentation, we consider a 2-dimensional $\boldsymbol{z}$ space and train $\mathcal{G}$ on MNIST and CelebA datasets of 256 sampled examples from all classes. The results are shown **[here](https://drive.google.com/file/d/1yXsW8zYuOcDqrrZU9iweCUtJ8luUBj2R/view?usp=share_link)** and **[there](https://drive.google.com/file/d/1vWkP1mjsOWWH1FWXYHNw7U1Nah8HdgMN/view?usp=share_link)**. We can see that the generated patterns transit smoothly across the 2D $\boldsymbol{z}$ space, and neighboring images share similar looks. These similar-looking images are generated from adjacent but meaningless $\boldsymbol{z}$'s, suggesting that the learned features encode high-level semantics. We have added the above discussion to Section 12.
> >
> > > Typos
> >
> > Thanks. We have fixed all the mentioned writing issues in the revision.
> >
> > We hope our reply above addresses your concerns. If so, we respectfully ask the reviewer to consider increasing your score. In any case, thanks again for your valuable comments that make this paper better.

---

> > > ### Author Response · Authors · 2022-11-30
> > > **Further experiment results for the novelty of generated images**
> > >
> > > Dear reviewer qHF8,
> > >
> > > Regarding to your last question asking whether GA-NTK can generate novel images, we have shown you in our last reply that the generator trained by Eq. (6) can learn a smooth space mapping that encodes high-level semantics.
> > >
> > > Now, we present further results to show that the downgrade effect you saw in Figures 18-19 is indeed due to the limited sized of training set. On a machine with NVIDIA RTX A6000 GPU of 48GB RAM (which is larger than the 32GB RAM in Tesla V100 GPU we originally used in the paper), we are able to train GA-NTK (defined in Eq. (4) *without* a generator) on MNIST dataset with $n=512$ training examples. The MNIST is a relatively simple task, so 512 examples might be sufficient. Here, we show the synthesized images given $b=n=$ **[64](https://drive.google.com/file/d/1KDOIyfEtedSjs9Kdn0jYq27ftb4kX3Y4/view?usp=share_link)**, **[256](https://drive.google.com/file/d/13Zouk69Sk6GFp-bg3SxBDYixKPjPcATx/view?usp=share_link)**, and **[512](https://drive.google.com/file/d/1RyNFcA1EoO6DDwF4EN0xf--AQ4KtQCnB/view?usp=share_link)**, where the leftmost image in each row is the image generated by GA-NTK and the rest images in the same row are the nearest neighbors (measured by SSIM) in the training set. It is clear that, when $b$ and $n$ increase, the downgrade effect is alleviated and GA-NTK tends to output more novel points.
> > >
> > > We hope the above addresses your concerns and are happy to answer your further questions.

---

### Official Review · Reviewer_ihk5 · 2022-10-28

**Confidence:** 3
**Correctness:** 3
**Technical Novelty And Significance:** 3
**Empirical Novelty And Significance:** 3
**Recommendation:** 3

**Clarity, Quality, Novelty And Reproducibility:**

Clarity: The presentation is clear and the manuscript is easy-to-read. But there are still minor flaws. The following list of the spots I realized:
******************************
Page 9: What is ``FNN''?
******************************

Quality: Although the presentation is good, I think the proposed method is not well supported by statistical background. The meaning is as follows. In the standard GANs framework, the generator tries to approximate the data distribution P_{data}. A common way to do so is [1] generating a hidden variable (noise) as z~P_Z (P_Z is usually chosen as Gaussian); [2] updating the generator G so that the distribution of G(z) is as close as possible to the data distribution. The distribution of G(z), P_g, resultantly approximates P_{data}. Hence we can use P_g as a generative model. The proposed method, however, has no such distribution. The author(s) do not discuss this point at all. Actually, the proposed way of generating artificial data is to find the minimizer of the loss defined by eq. (4); if the global minimizer of this minimization problem is obtained, then the generated artificial data is uniquely obtained as a function of the data and has no fluctuation, implying that it is not possible to generate diverse artificial data. Still, from another viewpoint, it is possible that a variety of artificial data can be obtained from optimization issues such as splitting the data into mini-batches and the local minimums of the loss. But the distribution of the variety caused by these issues is basically uncontrollable and cannot be expected to approximate the data distribution.
Hence, in the standard statistical meaning of the generative model, I think the proposed method does not provide any generative model. Due to this reason, I think the quality of the paper is not high.

Novelty: The base idea is novel and interesting, though its statistical justification is absent as I criticize above.

Reproducibility: Although I did not try to reproduce the result by myself, the algorithm is explained well and thus one can reproduce the result in principle. Hence there is no serious problem in reproducibility.


**Strength And Weaknesses:**

Strengths:
- The core part of the idea is novel.

- The computational cost of the proposed method is significantly smaller than other comparable methods.

Weaknesses:
- From a statistical viewpoint, there is no sufficient theoretical background for the proposed method, I think. The details are described in the Quality section below.


**Summary Of The Paper:**

This paper proposes a new framework, GA-NTK, of Generative adversarial networks (GANs) by constructing the discriminator based on the neural tangent kernel (NTK). The NTK allows to describe the discriminator output in a compact and simple function of the given data X and the auxiliary variables Z, and hence many computational issues involved in the GANs training, such as convergence problem, mode collapse, and gradient vanishing, are greatly suppressed. The proposed method's experimental results succeeded in producing artificial images of comparable qualities with other standard methods such as WGAN and SNGAN.


**Summary Of The Review:**

The core idea of the paper is novel and interesting but its statistical meaning is not clear as mentioned above. Taken as best one can, it can be said that the authors proposed an optimization framework of generative models rather than the statistical one. However, it is not possible for me to take it so favorably, and hence I vote for a rejection side about this paper.

---

> ### Author Response · Authors · 2022-11-19
> **Reply to Reviewer ihk5 (Part 1 of 2)**
>
> Thank you for your valuable comments and suggestions. We appreciate that you find the proposed GA-NTK method interesting and novel, and the paper well-written. We would like to point you to **[our general message to all reviewers](https://openreview.net/forum?id=_d2f3hRn0hT&noteId=jv7yTAeewGL)**. In what follows we reply to your particular comments.
>
> > Page 9: What is ''FNN''?
>
> Thanks, we have replace it with “fully-connected neural networks” in the revision.
>
> > The proposed method is not well supported by statistical background.
>
> We appreciate your constructive comment. Statistically, minimizing Eqs. (4) and (6) in the paper amounts to minimizing the Pearson $\chi^{2}$-divergence, a case of f-divergence, between $\mathcal{P}\_{\text{data}}+\mathcal{P}\_{\text{gen}}$ and $2\mathcal{P}\_{\text{gen}}$, where $\mathcal{P}\_{\text{data}}$ is the distribution of real data and $\mathcal{P}\_{\text{gen}}$ is the distribution of generated points. To see this, we first rewrite the loss of our discriminator $\mathcal{D}$, denoted by $\mathcal{L}(\mathcal{D})$, in expectation: $$\arg\min\_{\mathcal{D}}\mathcal{L}(\mathcal{D})=\arg\min\_{\mathcal{D}}\mathbb{E}\_{\boldsymbol{x}\sim\mathcal{P}\_{\text{data}}}\bigl[(\mathcal{D}(\boldsymbol{x})-1)^{2}\bigr]+\mathbb{E}\_{\boldsymbol{x}\sim\mathcal{P}\_{\text{gen}}}\bigl[(\mathcal{D}(\boldsymbol{x})-0)^{2}\bigr].$$
> Here, $\mathcal{P}\_{\text{gen}}$ can represent either $\boldsymbol{Z}$ in Eq. (4) or the output of the generator $\mathcal{G}$ in Eq. (6). Similarly, the loss function for our $\mathcal{P}\_{\text{gen}}$, denoted by $\mathcal{L}(\mathcal{P}\_{\text{gen}};\mathcal{D})$, can be written as follows:
> $$\arg\min\_{\mathcal{P}\_{\text{gen}}}\mathcal{L}(\mathcal{P}\_{\text{gen}};\mathcal{D})=\arg\min\_{\mathcal{P}\_{\text{gen}}}\mathbb{E}\_{\boldsymbol{x}\sim\mathcal{P}\_{\text{data}}}\bigl[(\mathcal{D}(\boldsymbol{x})-1)^{2}\bigr]+\mathbb{E}\_{\boldsymbol{x}\sim\mathcal{P}\_{\text{gen}}}\bigl[(\mathcal{D}(\boldsymbol{x})-1)^{2}\bigr].$$
> GA-NTK, in the form of the above two equations, is a special case of LSGAN [1]. Let $\mathcal{D}^*$ be the minimizer of the first equation. We can see that Eqs. (4) and (6) effectively solve the problem:
> $$\arg\min\_{\mathcal{P}\_{\text{gen}}}\mathcal{L}(\mathcal{P}\_{\text{gen}};\mathcal{D}^*)=\arg\min\_{\mathcal{P}\_{\text{gen}}}\mathbb{E}\_{\boldsymbol{x}\sim\mathcal{P}\_{\text{data}}}\bigl[(\mathcal{D}^* (\boldsymbol{x})-1)^{2}\bigr]+\mathbb{E}\_{\boldsymbol{x}\sim\mathcal{P}\_{\text{gen}}}\bigl[(\mathcal{D}^* (\boldsymbol{x})-1)^{2}\bigr].$$
> Mao et al. [1] show that, under mild relaxation, minimizing this objective yields minimizing the Pearson $\chi^{2}$-divergence between $\mathcal{P}\_{\text{data}}+\mathcal{P}\_{\text{gen}}$ and $2\mathcal{P}\_{\text{gen}}$:
> $$\arg\min\_{\mathcal{P}\_{\text{gen}}}\mathcal{L}(\mathcal{P}\_{\text{gen}};\mathcal{D}^{*})=\arg\min\_{\mathcal{P}\_{\text{gen}}}\chi\_{\text{Pearson}}^{2}(\mathcal{P}\_{\text{data}}+\mathcal{P}\_{\text{gen}}\Vert2\mathcal{P}\_{\text{gen}})$$
> $$=\arg\min\_{\mathcal{P}\_{\text{gen}}}\int(\mathcal{P}\_{\text{data}}(\boldsymbol{x})+\mathcal{P}\_{\text{gen}}(\boldsymbol{x}))\left(\frac{2\mathcal{P}\_{\text{gen}}(\boldsymbol{x})}{\mathcal{P}\_{\text{data}}(\boldsymbol{x})+\mathcal{P}\_{\text{gen}}(\boldsymbol{x})}-1\right)^{2}\textrm{d}\boldsymbol{x}.$$
> The loss becomes zero when $\mathcal{P}\_{\text{data}}(\boldsymbol{x})=\mathcal{P}\_{\text{gen}}(\boldsymbol{x})$ for all $\boldsymbol{x}$. Therefore, minimizing Eq. (4) or (6) brings $\mathcal{P}\_{\text{gen}}$ closer to $\mathcal{P}\_{\text{data}}$. We have added the above discussion to Section 6.
>
> [1] Mao et al., "Least Squares Generative Adversarial Networks," in Proc. of ICCV, 2017.
>
> To be continued...

---

> > ### Author Response · Authors · 2022-11-19
> > **Reply to Reviewer ihk5 (Part 2 of 2)**
> >
> > > If the global minimizer of this minimization problem is obtained, then the generated artificial data is uniquely obtained as a function of the data and has no fluctuation, implying that it is not possible to generate diverse artificial data.
> >
> > Sorry for not making this clear enough. The diversity of our generated data not only comes from the randomness of an optimization algorithm (e.g., initialization of $\boldsymbol{Z}$ or splitting of $\boldsymbol{X}$ into batches) but also from the objective in Eq. (4) itself. To show this, we update **[Figure 3](https://drive.google.com/file/d/13ElCJ2ju36GnDRnnUd4TfQR_lyimB7Yr/view?usp=share_link)** to include the images generated at the later stage of training. The patterns of these images keep changing over training time, despite little change in the loss score. This can be seen more clearly with the **[animation](https://drive.google.com/file/d/1LYMNNyBQ-mBQ-iNJNxXObg7Mgd-JtX0g/view?usp=share_link)**. The reason is that, in Eq. (4), the $\boldsymbol{Z}$ is optimized for a *moving* target—any change of $\boldsymbol{Z}$ causes $\mathcal{D}$ to be “retrained” instantly. Similarly, the training of $\mathcal{G}$ in Eq. (6) also shares this nice property. In Figure 11, the patterns of a generated image $\mathcal{G}(\boldsymbol{z})$ change over training time even when the input $\boldsymbol{z}$ is fixed. However, getting diverse artificial data through this property requires prolonged training time. In practice, we can simply initialize $\boldsymbol{Z}$ differently to achieve diversity faster. We have added the above discussion to Section 9.4.
> >
> > Furthermore, in Section 12 we study the “interpolated” images output by the generator $\mathcal{G}$ of GA-CNTKg taking equidistantly spaced $\boldsymbol{z}$'s along a segment in a two dimensional $\boldsymbol{z}$ space as the input. In the results shown **[here](https://drive.google.com/file/d/1yXsW8zYuOcDqrrZU9iweCUtJ8luUBj2R/view?usp=share_link)** and **[there](https://drive.google.com/file/d/1vWkP1mjsOWWH1FWXYHNw7U1Nah8HdgMN/view?usp=share_link)**, the generated patterns transit smoothly across the 2D $\boldsymbol{z}$ space, and neighboring images share similar looks. This provides another source of diversity (by varying $\boldsymbol{z}$) in our generator-based methods, like existing GANs.
> >
> > We hope our reply above addresses your concerns. If so, we respectfully ask the reviewer to consider increasing your score. In any case, thanks again for your valuable comments that make this paper better.

---

> > > ### Comment · Reviewer_ihk5 · 2022-11-23
> > > **a naive question**
> > >
> > > Thank you for your detailed reply. May be I do not understand some points of the paper and please let me ask a naive question.
> > >
> > > Let us focus on the simpler model (GA-NTK) defined through eq. (4), and suppose one has the minimizer (Z^n)^* of the loss in eq. (4). Then, how do you use (Z^n)^* to define your generator? Equivalently, how do you use (Z^n)^* to generate a new sample?

---

> > > > ### Author Response · Authors · 2022-11-23
> > > > **Thanks for the Question**
> > > >
> > > > We thank the reviewer for this question. In Eq. (4), there is no generator, and the $(\boldsymbol{Z}^{n})^*$ contains $n$ synthesized images directly. This setting allows us to study the essential elements for achieving single-level adversarial data synthesis. The Eq. (4) is made more complex (and practical) in Eqs. (5-7) to support batch-wise training, generator, and multiple pixel resolutions, respectively. Note that the minimizer $(\boldsymbol{Z}^{n})^*$ of Eq. (4) may bot be unique because $\boldsymbol{Z}^{n}$ is optimized for a *moving* target. During each SGD iteration for $\boldsymbol{Z}^{n}$, the kernel matrix $\boldsymbol{K}^{2n,2n}$ changes once $\boldsymbol{Z}^{n}$ is updated, which in turn changes the predictions and loss score of the discriminator $\mathcal{D}$. In other words, *both* $\boldsymbol{Z}^{n}$ and $\mathcal{D}$ are updated (against each other) in a single iteration. Figure 3 and the above animation confirm this by showing that $\boldsymbol{Z}^{n}$ continues evolving even after the loss score of $\boldsymbol{Z}^{n}$ converges.

---

> > > > > ### Comment · Reviewer_ihk5 · 2022-11-24
> > > > > **OK. But still not clear.**
> > > > >
> > > > > >  In Eq. (4), there is no generator, and it contains  synthesized images directly
> > > > >
> > > > > OK, so my first understanding was correct. My main concern in the review was this scheme could not be called  ``generator'' at least in the statistical sense. I now understand that the authors do not try to make any generator in GA-NTK, but just do make an adversarial data synthesis method. The generator is rather employed in GA-NTKg defined in eq. (6), but its mechanism as a generator is basically owing to the network constructing the generator and is independent of the loss proposed in the paper.
> > > > >
> > > > > Now the proposed framework itself becomes clear to me, but its statistical meaning is still not clear. The explanation about the chi^2 divergence is also not clear. In your formulas, what quantities correspond to P_{gen}?
> > > > >
> > > > > As a related point, in eq. (6) the dimension of Z is fixed to b/2 which is the same as the batch size, but in the most of generator the dimension of Z is fixed to an independent number from the batch size (e.g. 100 in CGAN and DCGAN). Why this is fixed to b/2? Referring the explanation just below eq. (6), there may be a misleading expression in eq. (6)...
> > > > >
> > > > > >Note that the minimizer  of Eq. (4) may bot be unique because  is optimized for a moving target. During each SGD iteration for , the kernel matrix  changes once  is updated, which in turn changes the predictions and loss score of the discriminator . In other words, both  and  are updated (against each other) in a single iteration. Figure 3 and the above animation confirm this by showing that  continues evolving even after the loss score of  converges.
> > > > >
> > > > > For this point, I think the authors misunderstand. Even if the kernel is the function of Z^n, the whole expression is a function of Z^n anyway (given other quantities such as X^n are fixed), and hence the global minimizer should be defined properly.

---

> > > > > > ### Author Response · Authors · 2022-11-25
> > > > > > **Thank You, and More Clarifications**
> > > > > >
> > > > > > We are delighted the GA-NTK objective makes sense to you as a means of adversarial data synthesis. Here are some further clarifications:
> > > > > >
> > > > > > > In your formulas, what quantities correspond to $\mathcal{P}_{\text{gen}}$?
> > > > > >
> > > > > > Let $n$ and $d$ be the number and dimension of data points, respectively. In Eq. (4), $\mathcal{P}\_{\text{gen}}$ is the empirical distribution of the rows of $\boldsymbol{Z}^{n}\in\mathbb{R}^{n\times d}$ (i.e., the $n$ synthesized points), while $\mathcal{P}\_{\text{data}}$ is the empirical distribution of the rows of $\boldsymbol{X}^{n}\in\mathbb{R}^{n\times d}$ (i.e., the $n$ example points). In Eq. (6), $\mathcal{P}\_{\text{gen}}$ is the empirical distribution of the output $\mathcal{G}(\boldsymbol{Z}^{n})\in\mathbb{R}^{n\times d}$ of the generator network taking $n$ noise vectors $\boldsymbol{Z}^{n}\in\mathbb{R}^{n\times l}$ as the input, where $l$ denotes the noise dimension.
> > > > > >
> > > > > > > In Eq. (6) the dimension of $\boldsymbol{Z}$ is fixed to $b/2$...
> > > > > >
> > > > > > Thank you very much for pointing out a typo: the “ $\boldsymbol{Z}\in\mathbb{R}^{l}$ ” should be “ $\boldsymbol{Z}\in\mathbb{R}^{\frac{b}{2}\times l}$ ” in the explanation just below Eq. (6). Each row of $\boldsymbol{Z}$ represents a noise vector of dimension $l$, which is independent with the batch size $b$. We will fix the typo in the next revision.
> > > > > >
> > > > > > > Even if the kernel is the function of $\boldsymbol{Z}^{n}$, the whole expression is a function of $\boldsymbol{Z}^{n}$ anyway, and hence the global minimizer should be defined properly.
> > > > > >
> > > > > > Yes, the global minimizer $(\boldsymbol{Z}^{n})^{* }$ is properly defined w.r.t. the loss. But this does not mean $(\boldsymbol{Z}^{n})^{* }$ is unique. Since the discriminator $\mathcal{D}$ (based on $\boldsymbol{K}$) changes with $\boldsymbol{Z}^{n}$, it is possible for two $\boldsymbol{Z}^{n}$ 's to give the same loss score *via fooling different* $\mathcal{D}$'s. As a result, these two $\boldsymbol{Z}^{n}$ 's may contain diverse synthesized points. The non-uniqueness allows us to obtain different $(\boldsymbol{Z}^{n})^{*}$ 's by starting gradient descent from differently initiated $\boldsymbol{Z}^{n}$ 's.
> > > > > >
> > > > > > We find that, during a single gradient descent process, the $\boldsymbol{Z}^{n}$ constantly changes even after the loss converges, as Figure 3 and the above animation show. On one hand, this observation justifies that $(\boldsymbol{Z}^{n})^{* }$ is not unique. On the other hand, it enables another approach to generating diverse artificial points: one can stop a gradient descent process at different time after convergence to get different $(\boldsymbol{Z}^{n})^{* }$ 's.
> > > > > >
> > > > > > We are happy to answer further questions if the above explanation is still not clear enough for you.

---

> > > > > > > ### Comment · Reviewer_ihk5 · 2022-12-01
> > > > > > > **I do not agree**
> > > > > > >
> > > > > > > I understand the typo around Eq. (6). That is understandable.
> > > > > > >
> > > > > > > But still, I think the authors misunderstand the meaning of Eq. (4). Consider a very simple situation where n=d=1. In that case, the loss in eq. (4) simply becomes $L=\sqrt{ (e^{-\lambda k(x,x)})^2+(1+e^{-\lambda k(x,z)})^2 }$ given kernel $k$. Minimizing this loss w.r.t. $z$ corresponds to bring $e^{-\lambda k(x,z)}$ closer to $-1$ if $\lambda>0$, which leads to maximizing $k(x,z)$. If our kernel is the standard RBF kernel, the maximizer of $k(x,z)$ thus becomes $z^*=x$, reproducing the datapoint $x$ as is. This is the unique minimizer of loss and is certainly producing a datapoint ``very very close'' to the given datapoint, but I think this kind of behavior is not the expected function of generator in the standard sense.

---

> > > > > > > > ### Author Response · Authors · 2022-12-01
> > > > > > > > **Further Clarification of Eq. (4)**
> > > > > > > >
> > > > > > > > Thanks for analyzing Eq. (4) using the case where $n=d=1$, which helps us better understand your concerns.
> > > > > > > >
> > > > > > > > > When $n=d=1$, Eq. (4) becomes $L=\cdots$, which leads to $z^{* }=x$.
> > > > > > > >
> > > > > > > > By definition, the exponential of a matrix
> > > > > > > >
> > > > > > > > $$\boldsymbol{A}=\begin{bmatrix}
> > > > > > > > a\_{1,1} & a\_{1,2} \\\ a\_{2,1} & a\_{2,2}
> > > > > > > > \end{bmatrix}\in\mathbb{R}^{2\times2}$$
> > > > > > > >
> > > > > > > > is
> > > > > > > > $$e^{\boldsymbol{A}}=\sum_{r=0}^{\infty}\frac{1}{r!}\boldsymbol{A}^{r}=\frac{1}{0!}\boldsymbol{I}+\frac{1}{1!}\boldsymbol{A}+\frac{1}{2!}\boldsymbol{A}^{2}+\cdots,$$
> > > > > > > >
> > > > > > > > which does *not* equal
> > > > > > > > $$\left[\begin{array}{cc}
> > > > > > > > e^{a_{1,1}} & e^{a_{1,2}} \\\ e^{a_{2,1}} & e^{a_{2,2}}
> > > > > > > > \end{array}\right]$$
> > > > > > > >
> > > > > > > > because each element $e_{i,j}^{\boldsymbol{A}}$  in the former matrix involves all $a_{s,t}$, $\forall s,t$, while an element $e^{a_{i,j}}$ in the latter involves only $a_{i,j}$.
> > > > > > > >
> > > > > > > > Now, let's go back to your case where $ n=d=1$. We have the kernel matrix
> > > > > > > >
> > > > > > > > $$\boldsymbol{K}=\left[\begin{array}{cc}
> > > > > > > > k(x,x) & k(x,z) \\\ k(z,x) & k(z,z)
> > > > > > > > \end{array}\right]\in\mathbb{R}^{2\times2}$$
> > > > > > > >
> > > > > > > > and thus
> > > > > > > >
> > > > > > > > $$e^{-\lambda\boldsymbol{K}}=\sum_{r=0}^{\infty}\frac{1}{r!}(-\lambda\boldsymbol{K})^{r}=\frac{1}{0!}\boldsymbol{I}+\frac{1}{1!}(-\lambda\boldsymbol{K})+\frac{1}{2!}(-\lambda\boldsymbol{K})^{2}+\cdots.$$
> > > > > > > >
> > > > > > > > If we replace $e^{-\lambda\boldsymbol{K}}$ with a finite approximation
> > > > > > > >
> > > > > > > > $$\frac{1}{0!}\boldsymbol{I}+\frac{1}{1!}(-\lambda\boldsymbol{K})+\cdots+\frac{1}{r!}(-\lambda\boldsymbol{K})^{r}$$
> > > > > > > >
> > > > > > > > for some finite $r$ in Eq. (4), then the loss $L$ becomes a polynomial of degree $r$ w.r.t. z (ignoring the effect of $k$). Analytically, we can obtain $z$ by solving the roots of the equation
> > > > > > > >
> > > > > > > > $$L'=0,$$
> > > > > > > >
> > > > > > > > where $L'$ is the derivative of $L$ w.r.t. $z$. Since $L'$ is a polynomial of degree $r-1$ w.r.t. $z$, there are $r-1$ roots (ignoring multiplicity). Generalizing this concept with $r\rightarrow\infty$, there could be an infinite number of distinct $z$'s minimizing Eq. (4). This is another way to explain why the $z$ solved by SGD keeps changing even after the loss converges, as Figure 3 and the above animation show.
> > > > > > > >
> > > > > > > > We hope the above addresses your concerns and are happy to answer your further questions.

---

> > > > > > > > > ### Comment · Reviewer_ihk5 · 2022-12-01
> > > > > > > > > **Sorry for my mistake. But the essential point is the same.**
> > > > > > > > >
> > > > > > > > > You are right. Sorry for my mistake. My previous formula was too simple, but the essential point I am saying is the same. The correct form of the loss is $L=\sqrt{ (e^{-\lambda k}_{11})^2 + (1+e^{-\lambda k}_{21})^2 }$.
> > > > > > > > >
> > > > > > > > > Now the exponential function of the matrix is correctly handled. And each term in the square root:
> > > > > > > > > $(e^{-\lambda k}_{11})^2$, and
> > > > > > > > >
> > > > > > > > > $(1+e^{-\lambda k}_{21})^2$
> > > > > > > > > can be analytically computed again, through assessing the eigenvalues of $e^{-\lambda k}$. If we denote those eigenvalues as $r_1,r_2$, the loss can be written as
> > > > > > > > > $L=\sqrt{ (\frac{1}{2}(e^{-\lambda r_1}+e^{-\lambda r_2}))^2 + (1+\frac{1}{2}(e^{-\lambda r_1}-e^{-\lambda r_2})  )^2 }$. Then, it is possible to analytically show that the minimum is obtained at $z^*=x$ if the kernel is RBF, though the computation is not simple hence I omit it here.

---

> > > > > > > > > > ### Author Response · Authors · 2022-12-02
> > > > > > > > > > **$z^*$ cannot be $x$ by definition**
> > > > > > > > > >
> > > > > > > > > > Please allow us to clarity one thing first. In Eq. (4), the kernel matrix $\boldsymbol{K}^{2n,2n}$ needs to be positive definite, implying that *a synthesized point $\boldsymbol{z}$ cannot be identical to any example $\boldsymbol{x}$*. After all, it's meaningless for a generative model to output example points.
> > > > > > > > > >
> > > > > > > > > > In the case where $n=d=1$, the $z=x$ is never a candidate solution. If your analysis is correct, then $z^{* }$ should be very close to $x$ in the feature space. Now, suppose $d>1$ and consider a particular minimizer $\boldsymbol{z}^{* }\in\mathbb{R}^{d}$ of Eq. (4), $\boldsymbol{z}^{* }\neq\boldsymbol{x}\in\mathbb{R}^{d}$. We show below that $\boldsymbol{z}^{* }$ is not unique. **Case 1: RBF kernel.** Let $\epsilon=\Vert\boldsymbol{z}^{* }-\boldsymbol{x}\Vert>0$. With an RBF kernel, any $\boldsymbol{z}$ residing on the sphere of an $\epsilon$-ball around $\boldsymbol{x}$ in the input space can also minimize Eq. (4). This is not very interesting because $\epsilon$ is small. **Case 2: NTK.** Let $c=\boldsymbol{x}^{\top}\boldsymbol{z}^{* }<1$ (assuming that both $\boldsymbol{x}$ and $\boldsymbol{z}$ are normalized). By definition of NTK described in Eq. (13) and (14), where
> > > > > > > > > >
> > > > > > > > > > $$k^{1}(\boldsymbol{x},\boldsymbol{x}')=\frac{\sigma_{w}^{2}}{d}\boldsymbol{x}^{\top}\boldsymbol{x}'+\sigma_{b}^{2}$$
> > > > > > > > > >
> > > > > > > > > > is the kernel function at the first network layer, we can see that any $\boldsymbol{z}$ having $\boldsymbol{x}^{\top}\boldsymbol{z}=c$ can minimize Eq. (4) as well. Furthermore, since NTK features can be exponentially more expressive in their depth [1], these different minimizers may be far away from each other in the input (pixel) space.
> > > > > > > > > >
> > > > > > > > > > We hope the above addresses your concerns and are happy to answer your further questions.
> > > > > > > > > >
> > > > > > > > > > [1] Schoenholz et al., "Deep Information Propagation," in Proc. of ICLR'17.

---

> > > > > > > > > > > ### Comment · Reviewer_ihk5 · 2022-12-02
> > > > > > > > > > > **confusing discussion**
> > > > > > > > > > >
> > > > > > > > > > > >the kernel matrix needs to be positive definite, implying that a synthesized point  cannot be identical to any example .
> > > > > > > > > > >
> > > > > > > > > > > Positive definiteness is just a designing principle of the kernel. It is different from the discussion about the loss property given kernel. Anyway once the kernel is given as the RBF one, the minimizer becomes $z^*=x$. This is sure. It is certain that at $z=x$ one of the eigenvalue, $r_2$, becomes zero and hence the corresponding matrix $k$ is not positive definite. But it does not produce any problem. The point I am saying through this analysis is that the global minimizer of eq. (4) certainly exists. And, as I have already pointed out, the statistical meaning of the solution is unclear. The discussions so far do not solve these points. Rather, for me, the authors seem to make the discussion confusing by putting irrelevant information on the table, as saying "NTK features can be exponentially more expressive in their depth" which is not related to my concern.
> > > > > > > > > > >
> > > > > > > > > > > Oveall, I think further discussion is pointless. I cannot raise the score but will raise my confidence level.

---

> > > > > > > > > > > > ### Author Response · Authors · 2022-12-02
> > > > > > > > > > > > **Our kernel is NTK, not RBF**
> > > > > > > > > > > >
> > > > > > > > > > > > Sorry for letting you feel this way. We have no intention to make things confusing. Let's explain our previous reply.
> > > > > > > > > > > >
> > > > > > > > > > > > First, our kernel is not RBF but NTK. We agree that with RBF kernel, the kernel matrix need not be positive definite. But the positive-definiteness is necessary for the NTK to be well defined (please see Jacot's paper “Neural Tangent Kernel: Convergence and Generalization in Neural Networks,” which shows that the convergence of the training of $\mathcal{D}$ is related to the positive-definiteness of NTK) and for our Theorem 3.1 to hold (please see the proof of Corollary 7.1). We are sorry for not making this precondition obvious for you and will state it more clearly in the next revision. With a positive definite NTK, the global minimizer of Eq. (4) still exists (we both agree on this), but is not unique.
> > > > > > > > > > > >
> > > > > > > > > > > > Second, we mentioned Schoenholz's paper to point out that different $\boldsymbol{z}$'s of the same (or similar) score may look very different in the pixel space. This is to address your concern that the generated points in $\boldsymbol{Z}^{n}$ in Eq. (4) may lack diversity if not relying on the randomness of the training algorithms.
> > > > > > > > > > > >
> > > > > > > > > > > > Our experimental results in Figure 3 and the uploaded animation align with our above arguments. You said
> > > > > > > > > > > >
> > > > > > > > > > > > > The point I am saying through this analysis is that the global minimizer of eq. (4) certainly exists.
> > > > > > > > > > > >
> > > > > > > > > > > > We totally agree with you. It is just that the global minimizer may not be unique. You also mentioned
> > > > > > > > > > > >
> > > > > > > > > > > > > The statistical meaning of the solution is unclear.
> > > > > > > > > > > >
> > > > > > > > > > > > Following your advice, we have shown that
> > > > > > > > > > > > 1. Minimizing Eqs. (4) and (6) amounts to minimizing the Pearson $\chi^{2}$-divergence between $\mathcal{P}\_{\text{data}}+\mathcal{P}\_{\text{gen}}$ and $2\mathcal{P}\_{\text{gen}}$.
> > > > > > > > > > > > 2. The global minimizer of Eq. (4) exists but is not unique. Yes, the feasible solution set of Eq. (4) excludes the sample points, but limiting the feasible solution set is quite normal (like introducing constraints) in an optimization problem.
> > > > > > > > > > > >
> > > > > > > > > > > > Could you please let us know what statistical meaning of the solution is still unclear to you?
> > > > > > > > > > > >
> > > > > > > > > > > > It does not seem fair to reject this paper because of the properties of RBF kernel that our GA-NTK isn't based on. We appreciate that the reviewer can tell us which part of our arguments is wrong or what still needs to be changed to address your concerns.

---

> > > > > > > > > > > > ### Author Response · Authors · 2022-12-05
> > > > > > > > > > > > **Your conclusion that $z^{*}=x$ might be incorrect**
> > > > > > > > > > > >
> > > > > > > > > > > > Dear reviewer ihk5:
> > > > > > > > > > > >
> > > > > > > > > > > > We took your comments seriously and tried to follow your analysis to get $\boldsymbol{z}^{*}=\boldsymbol{x}$ (ignoring its feasibility) for Eq. (4) in the case where $n=1$. However, we obtained a different result.
> > > > > > > > > > > >
> > > > > > > > > > > > ### Your analysis
> > > > > > > > > > > >
> > > > > > > > > > > > Consider the case where $n=1$. Let
> > > > > > > > > > > > $$\boldsymbol{K}=\left[\begin{array}{cc}
> > > > > > > > > > > > 1 & k(\boldsymbol{x},\boldsymbol{z}) \\\ k(\boldsymbol{x},\boldsymbol{z}) & 1
> > > > > > > > > > > > \end{array}\right]\in\mathbb{R}^{2\times2}$$
> > > > > > > > > > > >
> > > > > > > > > > > > The eigen-decomposition gives
> > > > > > > > > > > > $$\boldsymbol{K}=\boldsymbol{Q}\left[\begin{array}{cc}
> > > > > > > > > > > > r_{1} & 0 \\\ 0 & r_{2}
> > > > > > > > > > > > \end{array}\right]\boldsymbol{Q}^{\top}\quad\text{ and }\quad\boldsymbol{Q}=\left[\begin{array}{cc}
> > > > > > > > > > > > \sqrt{\frac{1}{2}} & -\sqrt{\frac{1}{2}} \\\ \sqrt{\frac{1}{2}} & \sqrt{\frac{1}{2}}
> > > > > > > > > > > > \end{array}\right],$$
> > > > > > > > > > > >
> > > > > > > > > > > > where $\boldsymbol{Q}$ is an orthogonal matrix. Since $0\leq k(\boldsymbol{x},\boldsymbol{z})\leq1$ for any $\boldsymbol{z}$, we have
> > > > > > > > > > > > $$r\_{1}+r\_{2}=2\quad\text{ and }\quad r\_{1}\rightarrow2\text{ as }k(\boldsymbol{x},\boldsymbol{z})\rightarrow1.$$
> > > > > > > > > > > >
> > > > > > > > > > > > The $\boldsymbol{z}$ affects the values of $r\_{1}$ and $r\_{2}$. In Eq. (4), we have
> > > > > > > > > > > > $$e^{-\lambda\boldsymbol{K}}=\boldsymbol{Q}\left[\begin{array}{cc}
> > > > > > > > > > > > e^{-\lambda r\_{1}} & 0 \\\ 0 & e^{-\lambda r_{2}}
> > > > > > > > > > > > \end{array}\right]\boldsymbol{Q}^{\top},$$
> > > > > > > > > > > >
> > > > > > > > > > > > Therefore, the square loss of Eq. (4) can be expressed as
> > > > > > > > > > > > $$L^{2}=(e\_{1,1}^{-\lambda\boldsymbol{K}})^{2}+(1+e_{2,1}^{-\lambda\boldsymbol{K}})^{2}$$
> > > > > > > > > > > > $$\quad=(\frac{1}{2}(e^{-\lambda r\_{1}}+e^{-\lambda(2-r\_{1})}))^{2}+(1+\frac{1}{2}(e^{-\lambda r\_{1}}-e^{-\lambda(2-r_{1})}))^{2}$$
> > > > > > > > > > > >
> > > > > > > > > > > > We can find the optimum $r_{1}^*$ by solving $\partial L^{2}/\partial r_{1}=0$, which gives
> > > > > > > > > > > > $$e^{-\lambda(2-r\_{1})}-e^{-\lambda r\_{1}}=1.$$
> > > > > > > > > > > >
> > > > > > > > > > > > This shows that:
> > > > > > > > > > > >
> > > > > > > > > > > > 1. The optimum $r\_{1}^{*}$ depends on the value of $\lambda$, which indicates how well the discriminator learns from the training set $(\boldsymbol{x}\oplus\boldsymbol{z},[1,0]^{\top})$.
> > > > > > > > > > > >
> > > > > > > > > > > > 2. The $r_{1}^*$ is not necessarily equal to 2 (happening only when $k(\boldsymbol{x},\boldsymbol{z})=1$). That is, the global optimal solution $\boldsymbol{z}^*$ of Eq. (4) needs *not* be $\boldsymbol{x}$.
> > > > > > > > > > > >
> > > > > > > > > > > > Note that the above analysis holds for *any* kernel, including RBF and NTK. We appreciate it if the reviewer can point out any mistake we made when reproducing your analysis.
> > > > > > > > > > > >
> > > > > > > > > > > > ### Reply to your previous comments
> > > > > > > > > > > >
> > > > > > > > > > > > If our above analysis is acceptable to you. Maybe we can continue the discussion, shall we?
> > > > > > > > > > > >
> > > > > > > > > > > > > Positive definiteness is just a designing principle of the kernel... It does not produce any problem.
> > > > > > > > > > > >
> > > > > > > > > > > > Considering the case where $n=1$. Suppose $\boldsymbol{K}$ is not positive definite, and the solution $\boldsymbol{z}^{* }=\boldsymbol{x}$ is admitted. Then, the $\mathcal{D}$ in Eq. (4) is asked to output
> > > > > > > > > > > > $$\mathcal{D}(\boldsymbol{x})=1\quad\text{ and }\quad\mathcal{D}(\boldsymbol{z^{*}})=\mathcal{D}(\boldsymbol{x})=0,$$
> > > > > > > > > > > >
> > > > > > > > > > > > which is an ill-defined function.
> > > > > > > > > > > >
> > > > > > > > > > > > Let's consider more a general case where $\mathcal{D}$ is specified by Eq. (3):
> > > > > > > > > > > > $$\mathcal{D}(\boldsymbol{X}^{n},\boldsymbol{Z}^{n})=(\boldsymbol{I}^{2n}-e^{-\lambda\boldsymbol{K}^{2n,2n}})(\boldsymbol{1}^{n}\oplus\boldsymbol{0}^{n})\in\mathbb{R}^{2n},$$
> > > > > > > > > > > >
> > > > > > > > > > > > where $\lambda$ controls, intuitively, how many gradient descent steps to take to train $\mathcal{D}$ on $2n$ examples $(\boldsymbol{X}^{n}\oplus\boldsymbol{Z}^{n},\boldsymbol{1}^{n}\oplus\boldsymbol{0}^{n})$. For $\mathcal{D}$ to be well-defined, it is necessary that $\mathcal{D}(\boldsymbol{X}^{n},\boldsymbol{Z}^{n})\rightarrow\boldsymbol{1}^{n}\oplus\boldsymbol{0}^{n}$ as $\lambda\rightarrow\infty$. This happens only when $\boldsymbol{K}^{2n,2n}$ is positive definite, since $(\boldsymbol{I}^{2n}-e^{-\lambda\boldsymbol{K}^{2n,2n}})\rightarrow\boldsymbol{I}^{2n}$ when $\lambda\rightarrow\infty$.
> > > > > > > > > > > >
> > > > > > > > > > > > > Anyway once the kernel is given as the RBF one, the minimizer becomes $z^{*}=x$. The point I am saying through this analysis is that the global minimizer of Eq. (4) certainly exists.
> > > > > > > > > > > >
> > > > > > > > > > > > Our above analysis show that $\boldsymbol{z}^*$ is not necessarily equal to $\boldsymbol{x}$ given either an RBF kernel or NTK. We can also see that $\boldsymbol{z}^*$ is not unique. **Case 1: RBF kernel.** Let $\epsilon=\Vert\boldsymbol{z}^{* }-\boldsymbol{x}\Vert>0$. With an RBF kernel, any other $\boldsymbol{z}$ residing on the sphere of an $\epsilon$-ball around $\boldsymbol{x}$ in the input space can also minimize Eq. (4). **Case 2: NTK.** Let $c=\boldsymbol{x}^{\top}\boldsymbol{z}^{* }<1$ (assuming that both $\boldsymbol{x}$ and $\boldsymbol{z}$ are normalized). By definition of NTK described in Eq. (13) and (14), any other $\boldsymbol{z}$ having $\boldsymbol{x}^{\top}\boldsymbol{z}=c$ can minimize Eq. (4) as well. In summary, the global optima of Eq. (4) exist but not unique.
> > > > > > > > > > > >
> > > > > > > > > > > > We hope our above explanation can address your concerns and are happy to answer your further questions. Thanks again for your inspiring comments that help us better understand the loss properties of GA-NTK.

---

### Official Review · Reviewer_rAgg · 2022-10-28

**Confidence:** 4
**Clarity, Quality, Novelty And Reproducibility:** The paper is well-written and the pro…
**Correctness:** 3
**Technical Novelty And Significance:** 4
**Empirical Novelty And Significance:** 3
**Recommendation:** 8

**Strength And Weaknesses:**

In general, the paper is well-written and the proposed GA-NTK method is very interesting and novel. Other strengths are listed below:

1. GA-NTK converts the original bilevel optimization problem of training GAN to a single-level problem, which significantly reduces the hardness of training GAN.

2. Experiment results on real-world datasets show that GA-NTK can generate images that align well with human perception.

3. Experiment results on synthesized data are strong enough to justify that GA-NTK can avoid mode collapse.


Other comments / questions:

1. I wonder the relationship between the batch size and the generative performance of the proposed method. For example, based on the FID results in Table 1, for GACNTKg, one can find a positive correlation on MNIST, a negative correlation on CIFAR-10, and no correlation on CIFAR-100. Any insight about that?

2. What is the time usage for training GA-NTK and its variants?

3. In the second paragraph of page 2, I disagree with the claim that gradients of $\theta_g$ can not be back-propagated through the inner maximization problem. Actually, once you find the solution of the inner maxization problem in which we denoting it as $\theta_D^*$, then the gradient for the current $\theta_g$ can be written in an analytical form with respect to $\theta_D^*$.

4. The comparison between GA-NTK and existing approaches may be a little unfair, as GA-NTK does not use the same model architectures as that in existing approaches. A more fair comparison for me may be comparing existing approaches with its GA-NTK version in which the finite-wide discrimators are replaced with their corresponded infinite-wide linear conterparts.

5. It would be interesing to compare the convergence speed of GA-NTK with previous GAN training algorithms.

**Summary Of The Paper:**

This paper proposes a novel NTK-based GAN named GA-NTK. Specifically, the authors use an infinite-wide linear neural network (i.e., NTK model) as the discriminator for GAN, which thus enables one to directly obtain a closed-form solution for the inner maximization problem in GAN training. Compared with vanilla GAN, GA-NTK enjoys better convergence and less mode collapse and can avoid gradient vanishing. Experiments are conducted on both synthesized and real-world datasets to justify the effectiveness of the proposed method.

**Summary Of The Review:**

The proposed GA-NTK is novel and the experiment results are strong enough to justify the effectiveness of GA-NTK. Besides, the paper also extends the application domain of NTK and opens up new research direction of GAN training. Therefore, I suggest to accept this paper.

---

> ### Author Response · Authors · 2022-11-19
> **Reply to Reviewer rAgg (Part 1 of 2)**
>
> Thank you for your valuable comments and suggestions. We appreciate that you find the proposed GA-NTK method interesting and novel, most parts of the paper well-written, our experiments convincing and reproducible, and our mathematical settings clear. We would like to point you to **[our general message to all reviewers](https://openreview.net/forum?id=_d2f3hRn0hT&noteId=jv7yTAeewGL)**. In what follows we reply to your particular comments.
>
> > 1. Any insights about the relationship between the training data size and the generative performance of GA-NTK?
>
> Good question. We can see from Table 1 that, without a generator network, the GA-NTK performs better when the date size increases. However, this is not the case for GA-NTKg having a generator network. We have resampled training data and rerun the experiments 5 times with different initial values of $\boldsymbol{Z}^{n}$ but obtained similar results. Therefore, we believe the instability is due to the sample complexity of the generator network—256 examples or less are insufficient to train a stable, high-quality generator. This is evident in Figures 12(b)-16(b) where the generator outputs unrecognizable images more often. We have add the above discussion to Section 10 in the revision.
>
> > 2. What is the time usage for training GA-NTK variants and how does it compare with training GANs?
>
> Thanks for the constructive comment. The following shows the number of iterations and the wall-clock time required to train different methods on different datasets of 256 randomly sampled images:
>
> | MNIST | #iterations | iterations per second | wall clock time |
> | -------- | -------- | -------- | -------- |
> | DCGAN    | 7400     | 20 it/s     |  370 secs    |
> | LSGAN    | 5100     | 19 it/s     |  268 secs    |
> | WGAN     | 7000     | 19 it/s     |  368 secs    |
> | WGAN-GP  | 3400     | 18 it/s     |  189 secs    |
> | SNGAN    | 12800    | 18 it/s     |  711 secs    |
> | GA-NTK   | 500      | 14 it/s     |  35  secs    |
> | GA-NTKg  | 1600     | 9 it/s      |  177 secs    |
>
> | CIFAR-10 | #iterations | iterations per second | wall clock time |
> | -------- | -------- | -------- | -------- |
> | DCGAN    |  NA    | 17 it/s     |   not converged      |
> | LSGAN    |  NA    | 17 it/s     |   not converged      |
> | WGAN     |  14000    |  16 it/s     | 875  secs        |
> | WGAN-GP  |  11100    |  15 it/s     | 740  secs        |
> | SNGAN    |  NA       |  14 it/s     |   not converged  |
> | GA-NTK   |  600      |  13 it/s     |  46 secs         |
> | GA-NTKg  |  6200     |  8 it/s      |  775 secs        |
>
> | CelebA |#iterations | iterations per second | wall clock time |
> | -------- | -------- | -------- | -------- |
> | DCGAN    |  NA      | 13 it/s  |   not converged    |
> | LSGAN    |  NA      | 12 it/s  |   not converged    |
> | WGAN     |  18800   | 12 it/s  |    1566  secs      |
> | WGAN-GP  |  11200   | 10 it/s  |    1120  secs      |
> | SNGAN    |  NA      | 9 it/s   |   not converged    |
> | GA-NTK   |  1200    | 6 it/s   |     20 secs        |
> | GA-NTKg  |  5900    | 5 it/s   |  1180 secs         |
>
> We use the batch-wise GA-CNTK and GA-CNTKg and set the batch size b to 64 for all methods. The experiments run on a machine with a single NVIDIA Tesla V100 GPU. For DCGAN and LSGAN whose loss scores do not reflect image quality, we monitor the training process manually and stop it as long as the generated images contain recognizable patterns. But these methods do not seem to converge. For other methods, we use the early-stopping with the patience of 10000 steps and delta of 0.05 to determine convergence. The above results show that the number of iterations required by either batch-wise GA-CNTK or GA-CNTKg is significantly smaller than that used by GANs. This justifies our claims in Section 1. However, the batch-wise GA-CNTK and GA-CNTKg run fewer iterations per second than GANs because of the higher computation cost involved in back-propagating through the kernel matrix $\boldsymbol{K}^{b,b}$. In terms of wall-clock time, the batch-wise GA-CNTK is the fastest while the GA-CNTKg runs as fast as WGAN-GP. We expect that, with the continuous optimization of the Neural Tangents library [1] which our code is based on, the training speed of GA-NTK variants can be further improved. We have added the above discussion to Section 13.
>
> [1] Novak et al., "Neural Tangents: Fast and Easy Infinite Neural Networks in Python," in Proc. of ICLR, 2019.
>
> To be continued...

---

> > ### Author Response · Authors · 2022-11-19
> > **Reply to Reviewer rAgg (Part 2 of 2)**
> >
> > > 3. In the second paragraph of page 2, I disagree with the claim that gradients of $\boldsymbol{\theta}\_{\mathcal{G}}$ can not be back-propagated through the inner maximization problem. Actually, once you find the solution of the inner maximization problem in which we denoting it as $\boldsymbol{\theta}\_{\mathcal{D}}^*$, then the gradient for the current $\boldsymbol{\theta}\_{\mathcal{G}}$ can be written in an analytical form with respect to $\boldsymbol{\theta}\_{\mathcal{D}}^{*}$.
> >
> > In traditional GANs, the problem is that $\boldsymbol{\theta}\_{\mathcal{D}}^{*}$ are *unknown* before running SGD. Let $\boldsymbol{\theta}\_{\mathcal{D}}^{(s)}$ be the weights of the discriminator at the $s$-th SGD steps for solving the inner maximization problem. To solve the outer minimization problem, we need to compute
> > $$\nabla_{\boldsymbol{\theta}\_{\mathcal{G}}}\mathcal{J}(\mathcal{D}(\boldsymbol{X}^{n}\oplus\mathcal{G}(\boldsymbol{Z}^{n});\boldsymbol{\theta}\_{\mathcal{D}}^{(s)}),\boldsymbol{1}^{n}\oplus\boldsymbol{0}^{n}),$$
> > where $\mathcal{J}$ is the GAN objective. We have
> > $$\boldsymbol{\theta}\_{\mathcal{D}}^{(s)}=\boldsymbol{\theta}\_{\mathcal{D}}^{(s-1)}+\eta\cdot\nabla\_{\boldsymbol{\theta}\_{\mathcal{D}}}\mathcal{J}(\mathcal{D}(\boldsymbol{X}^{n}\oplus\mathcal{G}(\boldsymbol{Z}^{n});\boldsymbol{\theta}\_{\mathcal{D}}^{(s-1)}),\boldsymbol{1}^{n}\oplus\boldsymbol{0}^{n}),$$
> > $$\boldsymbol{\theta}\_{\mathcal{D}}^{(s-1)}=\boldsymbol{\theta}\_{\mathcal{D}}^{(s-2)}+\eta\cdot\nabla_{\boldsymbol{\theta}\_{\mathcal{D}}}\mathcal{J}(\mathcal{D}(\boldsymbol{X}^{n}\oplus\mathcal{G}(\boldsymbol{Z}^{n});\boldsymbol{\theta}\_{\mathcal{D}}^{(s-2)}),\boldsymbol{1}^{n}\oplus\boldsymbol{0}^{n}),$$
> > $$\boldsymbol{\theta}\_{\mathcal{D}}^{(s-2)}=\cdots$$
> > Each $\boldsymbol{\theta}\_{\mathcal{D}}^{(i)}$, $i=0,1,\cdots,s$, depends on $\mathcal{G}(\boldsymbol{Z}^{n})$ parametrized by $\boldsymbol{\theta}\_{\mathcal{G}}$. Replacing $\boldsymbol{\theta}\_{\mathcal{D}}^{(s)}$ with $\boldsymbol{\theta}\_{\mathcal{D}}^{(s-1)},\cdots,\boldsymbol{\theta}\_{\mathcal{D}}^{(0)}$ in the gradient $\nabla\_{\boldsymbol{\theta}\_{\mathcal{G}}}\mathcal{J}(\mathcal{D}(\boldsymbol{X}^{n}\oplus\mathcal{G}(\boldsymbol{Z}^{n});\boldsymbol{\theta}\_{\mathcal{D}}^{(s)}),\boldsymbol{1}^{n}\oplus\boldsymbol{0}^{n})$, we can see that evaluating the gradient requires computing high-order derivatives w.r.t. $\boldsymbol{\theta}\_{\mathcal{G}}$. If we treat $\boldsymbol{\theta}\_{\mathcal{D}}^{(s)}$ as a constant in $\nabla\_{\boldsymbol{\theta}\_{\mathcal{G}}}\mathcal{J}(\mathcal{D}(\boldsymbol{X}^{n}\oplus\mathcal{G}(\boldsymbol{Z}^{n});\boldsymbol{\theta}\_{\mathcal{D}}^{(s)}),\boldsymbol{1}^{n}\oplus\boldsymbol{0}^{n})$ instead, then we effectively run the alternating SGD.
> >
> > > 4. A more fair comparison for me may be comparing existing approaches with its GA-NTK version in which the finite-wide discriminators are replaced with their corresponded infinite-wide linear counterparts.
> >
> > To the best of our knowledge, so far NTK can only approximate (an ensemble of) neural networks with the MSE loss in closed form. In contrast, existing discriminators of GANs usually use more complex loss functions such as the KL-divergence and Wasserstein distance. Some discriminator losses even have regularization terms. For example, the decision boundary of the WGAN-GP discriminator is regularized to be Lipchitz continuous. Kernelizing these discriminators is not a trivial task and is out of the scope of this paper. While this is an interesting direction to explore in the future, our contribution in this regard is to show that the kernelization of (an ensemble of) neural networks with the MSE loss suffices to guide $\boldsymbol{Z}^{n}$ or $\mathcal{G}$ to synthesize plausible images. On the other hand, using the exact GAN discriminators allows us to study the practical limitations of GANs such as the failure of convergence, mode collapse, and vanishing gradients.
> >
> > We hope our reply above is satisfactory to you. If so, we humbly ask the reviewer to champion this paper. Thanks again for your valuable comments that make this paper better.

---

> > > ### Comment · Reviewer_rAgg · 2022-11-24
> > > **Good job!**
> > >
> > > Thanks to the authors for their response. The comments basically address my questions. Please include them in the final version of the paper.
> > >
> > > Besides, regarding the weaker performance of GA-NTKg, I think one way to improve it could be *obtaining the closed-form NTK discriminator with a small batch of randomly sampled data while training the generator with the full dataset*. Nevertheless, this is out of the scope of this paper. The authors could study this in their future works.
> > >
> > > In general, I like this paper and look forward it to appearing in ICLR.

---

> > > > ### Author Response · Authors · 2022-11-25
> > > > **Thank You**
> > > >
> > > > We will include all the modifications in the next revision and study the training technique you proposed. Thanks again for your positive comments and support.

---

### Official Review · Reviewer_LpjQ · 2022-10-30

**Confidence:** 2
**Correctness:** 3
**Technical Novelty And Significance:** 4
**Empirical Novelty And Significance:** 4
**Recommendation:** 8

**Clarity, Quality, Novelty And Reproducibility:**

Clarity:

I do not understand the points [3]--[6*].
Other parts have a good clarity.

---

Quality:

I think that the quality of the paper is not low, but I do not understand the points [3]--[6*].
So I will reserve a definite evaluation until the response.

---

Novelty:

As I wrote in [1*], I give a good evaluation regarding the novelty.

---

Reproducibility:

I took a glance of GitHub repository.
I think that experimental part of this study is reproducible.
However, I want you to cope with [6*], to improve the understandability (a basis of the reproducibility).

**Strength And Weaknesses:**

I place special emphasis on comments with the mark *.

---

Strength:

[1*] As the authors also wrote at the end of Section 1, the main significance of this study is not performance improvement but novelty of the research direction.
This study is an interesting attempt that uses an interesting idea to solve the problems of GANs, which are the motivation for the study.

---

Weakness:

[2] There are writing misses.

[2-1] You should use \citep as well as \citet.
For example, you can write "theoretical interpretations Arjovsky & Bottou (2017)." in p.3 as "theoretical interpretations (Arjovsky & Bottou, 2017)." by using \citep.
Also, "Franceschi et al. Franceschi et al. (2021)" in p.3 is strange for me; you can write "Franceschi et al. (2021)" by using \citet only.
Please look up the use of \citep and \citet and rewrite the relevant parts.

[2-2] In l.12 in p.4, value each element -> value of each element.
Check your writing (including other parts) again.

[2-3] In eq.(4), $\arg\min\_{Z^n} L(Z^n)=\arg\min\_{Z^n} ||\cdot||_2$ -> $\arg\min\_{Z^n} L(Z^n)$, where $L(Z^n)=||\cdot||^2$.
You should define $L(Z^n)$ explicitly, and unify $||\cdot||\_2$ and $||\cdot||$ into one if they are the seme (see also norm in other parts).

[2-4] The $f$-divergence appears in places, but what function was used for $f$?

[2-5] In Figure 3, title is partially obscured.

[2-6] In Figure 5, what is "real" in the legend?

[2-7] In the last equation in p.14, absolute value symbol is missing:
for example, $\nabla_{z_j} L(Z^n)\le$ -> $|\nabla_{z_j} L(Z^n)|\le$.

[2-8] You should centralize Figure 14.

[2-9] In Section 10, ?? -> 18.

---

Question:

Regarding the following points, it may be that the authors' description is appropriate and there is no problem, just because I have not understood it correctly.
I do not reflect these points in my current recommendation score.
Depending on the authors' response, I may change my recommendation score.

[3] Is it necessary to solve an optimization problem like eq.(4) every time to generate pseudo examples?
If so, that is a demerit compared with GANs, which can generate an infinite number of pseudo examples without additional training from a trained model.
This disadvantage should be written more clearly.
(Even so, I will not lower the score.)

[4*] From the experimental results (see especially Figure 3) and my understanding of the formulas, it seems to me that GA-NTK that is sufficiently trained with a large $t$ would simply return an training example (or mixture of training examples) as a pseudo example.
Is this question correct?
If correct, such data synthesis would have low practical value.
It does not seem to me that the use of a generator model adequately solves such a problem.
Also, the authors should explain when (i.e., at what $t$) to stop learning.

[5] This question relates to [4*].
I am interested in the relationship between the batch size $b$ and the novelty of the pseudo example (dissimilarity to the training examples $X^{b/2}$).
I suspect that if $b$ is small, each generated pseudo example will be fairly close to one of the training examples $X^{b/2}$.
Is this question correct?
If correct, I think that the authors should mention this issue.

[6*] It is difficult for me to understand the whole algorithm.
I could not understand the relation between the index $t$ in $\lambda=\eta \cdot t$ in eq.(3), the index $j$ in Theorem 3.1, "Epochs" in Figure 3, and "Iterations" in Figure 5, and the evolution of $Z^n$ and $Z^{n,(j)}$.
Please write a pseudo code (or modify the text) so that the relation of these objects become clear.

Also, how does $t$ change when $Z^n$ changes?
I think this question is relevant to the validity of applying NTK theory.

**Summary Of The Paper:**

In this paper, the authors propose generative adversarial neural tangent kernel (GA-NTK) and its variants.
They model the discriminator as a Gaussian process whose mean (and covariance) is governed by NTK.
Also, they can be performed via a single-level training process as a whole, because the training dynamics of their discriminators can be evaluated in closed form.
The authors claim that this property allows them to avoid the training difficulties often encountered with GANs, and experimentally confirmed this claim and their effectiveness for data synthesis.

**Summary Of The Review:**

I give this study a positive score now, on the basis of the novelty and interest of the idea [1*].
However, the description of the proposed method is insufficient and not clear [3]--[6*].
These points could be improved!
I will change the score up or down, depending on the responses to [3]--[6*].

---

I changed the score from 6 to 8.

---

> ### Author Response · Authors · 2022-11-19
> **Reply to Reviewer LpjQ**
>
> Thank you for valuable comments and suggestions. We appreciate that you find the proposed GA-NTK method interesting and novel, our experiments convincing and reproducible, and our direction significant for future research. We would like to point you to **[our general message to all reviewers](https://openreview.net/forum?id=_d2f3hRn0hT&noteId=jv7yTAeewGL)**. In what follows we reply to your particular comments.
>
> > [2] Writing misses...
>
> Thank you. We have fixed all the issues and added necessary citations to clarify terms in the revision.
>
> > [3] Is it necessary to solve Eq. (4) every time to generate pseudo examples?
>
> GA-NTK can have a generator network $\mathcal{G}$, as Eq. (6) shows. The generator network maps a random vector $\boldsymbol{z}$ to a point $\mathcal{G}(\boldsymbol{z};\boldsymbol{\theta}\_{\mathcal{G}})$ in data space just like that of GANs. So, it's not necessary to solve Eq. (4) every time to generate new data. Please see the images synthesized by $\mathcal{G}$ in Figures 2 and 12(b)–16(b).
>
> > [4*] What value of t to use when training GA-NTK?
>
> Thank you for the constructive comment. The $t$ (and $\lambda$) is a hyperparameter that needs to be tuned before training. We discuss in Section 3.1 that its value should be large enough but finite. Therefore, we search for the value that makes the discriminator nearly separate the real data from pure noises in an auxiliary learning task. The complete training procedure is as follows:
>
> 1. Tune the hyperparameter $t$ (and $\lambda$) by using an auxiliary task asking the discriminator to separate real data from pure noises, as described in Algorithm 1;
> 2. Fix $t$ (and $\lambda$) and solve $\boldsymbol{Z}^{n}$ in Eq. (4) by ordinary gradient descent.
>
> We have added a paragraph right before Section 3.1 to describe the complete training procedure. More details of step 1 can be found in Section 8.3.
>
> > [5] The impact of small batch size $b$.
>
> We found that a very small $b$ (e.g., $b=1$) results in a **[blurry mean image](https://drive.google.com/file/d/1nR0269tFxosyQFYaiXK-hOAiMTTXFPtl/view?usp=share_link)** regardless of model architectures and initializations of model weights and $\boldsymbol{Z}^{n}$ because the mean image is the best for simultaneously fooling many NTK discriminators, each trained on a single example. However, in practice this setting is less common as one usually aims to use the largest $b$ possible (Brock et al., 2018). Figure 7 shows that a batch size of 256 suffices to give plausible results and has a positive impact on the global coherence of image patterns. We have extended Section 9.2 to discuss the above.
>
> Brock et al., "Large scale GAN training for high fidelity natural image synthesis," in Proc. of ICLR, 2019.
>
> > [6*] What's the relation between $t$ in Eq. (3), index $s$ in Theorem 3.1, “Iterations” in Figure 5, and “Epochs” in Figure 3?
>
> The $t$ is a hyperparameter controlling how well the discriminator should be “analytically trained” on ($\boldsymbol{X}^{n}\oplus\boldsymbol{Z}^{n}$, $\boldsymbol{1}^{n}\oplus\boldsymbol{0}^{n}$) in Eq. (4). The index $s$ is the number of gradient descent steps actually taken to solve $\boldsymbol{Z}^{n}$ in Eq. (4), which is the same as the “Iterations” in Figure 5 and the “Epochs” in Figure 3. We have changed the “Epochs” in Figure 3 to “Iterations” to make our presentation consistent.
>
> We hope our reply above addresses your concerns. If so, we respectfully ask the reviewer to consider increasing your score. In any case, thanks again for your valuable comments that make this paper better.

---

> > ### Comment · Reviewer_LpjQ · 2022-11-26
> > **2nd Review by Reviewer LpjQ**
> >
> > Sorry I have misunderstood the discussion.
> > I got that $t$ (and $\lambda$) is a hyperparameter and $s$ is a learning iteration (with fixed $t$).
> > I had thought that $t$ is increased in the learning process.
> > Thus, the comments [2], [3*], [6] are resolved.
> >
> > ---
> >
> > However, I'd like to comment on [4*] and [5] additionally.
> >
> > I ask you whether you examined the effect of $\lambda$ (or $t$) to the performance of data generation (you answered the effect of small $b$ to the performance).
> > I recognize that Algorithm 1 determines $\lambda$ heuristically.
> > Did you try other $\lambda$.
> > I am worried about that GA-NTK with large $\lambda$ that is sufficiently trained with a large $s$ would simply return an training example (or mixture of training examples) as a pseudo example if $b$ is not large, as I wrote in [4*] and [5].
> > You may reply that using small $\lambda$ and $s$ can inject noise, but that noise would be not designed for the task (it is merely random).
> > Thus, I think that your proposal must be designed to work with large $t$ and $s$.
> >
> > GANs expect a discriminator to learn the intrinsic (low-dimensional) difference between true and pseudo examples from large examples (and memorize large examples), but GA-NTK does not.
> > I recognize that NTK-based discriminator is a non-parametric estimator like smoothing based on small (like 256) training examples.
> > Your that choice is not bad, but the implications of it should be carefully discussed.
> > I think that by that choice, the generated data would be very relevant to small training examples, and diversity would be lost.
> > You may reply that $Z^{b/2}\sim\mathcal{N}(0,1)$ in (6) increase the diversity, but I think that it is just interpolation of training examples.
> > This thought is also supported by Figures 18--19, and your additional experiment for Figure 7.
> >
> > You discussed the situation where the number of available examples is small, in Section 11.
> > But, proper data generation in this situation is impossible in the first place.
> > I am interested in the situation where the number of available examples are large.
> > For proper data generation, I think that NTK-based discriminator, a non-parametric estimator, requires to use quite large $b$ (e.g., $b=n$ with large $n$).
> >
> > Statisticians often say "generalization error = approximation error + estimation error + optimization error".
> > I think non-parametric NTK-based discriminator is worse than parametric GANs estimator in estimation error, but that NTK-based discriminator is better than GANs estimator in optimization error, if NTK-based discriminator uses $b=n$ in the large-sample limit.
> > Considered in this way, your proposal may be attractive.
> >
> > However, I am still interested in whether using large $b=n$ makes GA-NTK generate novel examples (non-similar to training examples), and relation between $b$ and novelty.
> > Can we use GA-NTK with large $b=n$ in computational perspective?
> > Can we set $b=2^{10}, 2^{15}, 2^{20}, 2^{25}$?
> >
> > I do not require conference papers to discuss all relevant issues completely.
> > This comment may be of my personal interest.
> > If GA-NTK cannot be computed with large $b$, I do not reduce the score.
> > I wish you consider this question and that this comment leads to improve the paper.

---

> > > ### Author Response · Authors · 2022-11-30
> > > **Reply to 2nd Review by Reviewer LpjQ**
> > >
> > > Thanks for your insightful comments. We are delighted you find the proposed GA-NTK attractive in some ways. Following replies to your questions:
> > >
> > > > The effect of $\lambda$ and $s$.
> > >
> > > The $\lambda$ controls how well the discriminator $\mathcal{D}$ is analytically trained in Eq. (4). A too large $\lambda$ yields a very small $e^{-\lambda\boldsymbol{K}^{2n,2n}}$ (as $\boldsymbol{K}^{2n,2n}$ is positive definite) and therefore
> > > $$\mathcal{D}(\boldsymbol{X}^{n},\boldsymbol{Z}^{n})=(\boldsymbol{I}^{2n}-e^{-\lambda\boldsymbol{K}^{2n,2n}})(\boldsymbol{1}^{n}\oplus\boldsymbol{0}^{n})\approx(\boldsymbol{1}^{n}\oplus\boldsymbol{0}^{n})\in\mathbb{R}^{2n}$$
> > >
> > > To minimize Eq. (4), the synthesized $\boldsymbol{Z}^{n}$ needs to be very similar to $\boldsymbol{X}^{n}$ to fool $\mathcal{D}$. We call this the downgrade effect. On the other hand, a too small $\lambda$ introduces noise and thus yields unrecognizable images in $\boldsymbol{Z}^{n}$. In practice, we find that the value returned by Algorithm 1, denoted by $\hat{t}$, gives acceptable image quality while avoiding the downgrade effect. We also find the performance of GA-CNTK insensitive to the values around $\hat{t}$. This can be verified by comparing the generated images on CelebA dataset (of 256 samples) when **[$\boldsymbol{t=\hat{t}/2}$](https://drive.google.com/file/d/14BrGzL2n-jqJKWxtPb-sLOv5T4kPH8kN/view?usp=share_link)**, **[$\boldsymbol{t=\hat{t}}$](https://drive.google.com/file/d/14E2mlwuflyKahcP-_lYAZlguv_eCJk4Z/view?usp=share_link)**, and **[$\boldsymbol{t=2\hat{t}}$](https://drive.google.com/file/d/14KQa5hFz0bPaoPv_WwnxGgEucyRVFwCq/view?usp=share_link)**.
> > >
> > > The $s$ denotes the number of SGD steps to run to solve $\boldsymbol{Z}^{n}$ in Eq. (4). A larger $s$ brings $\boldsymbol{Z}^{n}$ closer to an optimum. In practice, we use the early-stopping with the patience of 10000 steps and delta of 0.05 to determine the best value of $s$ at training time. Nevertheless, a larger $s$ is acceptable, because Eq. (4) has an infinite number of optima. In **[Figure 3](https://drive.google.com/file/d/13ElCJ2ju36GnDRnnUd4TfQR_lyimB7Yr/view?usp=share_link)**, we can see that the images generated at the later stage of the SGD training are constantly changing, despite little change in the loss score. This can be seen more clearly with the **[animation](https://drive.google.com/file/d/1LYMNNyBQ-mBQ-iNJNxXObg7Mgd-JtX0g/view?usp=share_link)**. The reason is that, in Eq. (4), the $\boldsymbol{Z}^{n}$ is optimized for a moving target—any change of $\boldsymbol{Z}^{n}$ causes $\mathcal{D}$ to be “retrained” instantly. So, it is possible for two $\boldsymbol{Z}^{n}$ 's to give the same loss score via fooling different $\mathcal{D}$'s.
> > >
> > > In summary, the GA-NTK can work with a large $s$. But the value of $\lambda$ should be tuned to avoid both noise and downgrade effect.
> > >
> > > > Whether using large $b=n$ makes GA-NTK generate novel examples?
> > >
> > > Thanks for this constructive comment. We have tried many ways to increase the batch size and, in the end, $b=n=256$ is the best we can get on a machine with NVIDIA Tesla V100 GPU of 32GB RAM. We believe scaling up $b$ to a much larger number like 1024 or 2048 requires non-trivial effort, and we leave it as a future work.
> > >
> > > However, on a machine with NVIDIA RTX A6000 GPU of 48GB RAM, we are able to train GA-NTK on MNIST dataset with $b=n=512$. The MNIST is a relatively simple task, so 512 examples might be sufficient. Here, we show the synthesized images given $b=n=$ **[64](https://drive.google.com/file/d/1KDOIyfEtedSjs9Kdn0jYq27ftb4kX3Y4/view?usp=share_link)**, **[256](https://drive.google.com/file/d/13Zouk69Sk6GFp-bg3SxBDYixKPjPcATx/view?usp=share_link)**, and **[512](https://drive.google.com/file/d/1RyNFcA1EoO6DDwF4EN0xf--AQ4KtQCnB/view?usp=share_link)**, where the leftmost image in each row is the image generated by GA-NTK and the rest images in the same row are the nearest neighbors (measured by SSIM) in the training set. It is clear that, when $b$ and $n$ increase, the downgrade effect is alleviated and GA-NTK tends to output more novel points.
> > >
> > > We hope the above addresses your concerns, and thanks again for your valuable suggestions that make this paper better.

---

> > > > ### Comment · Reviewer_LpjQ · 2022-12-08
> > > > **3rd Review by Reviewer LpjQ**
> > > >
> > > > I here summarize my final opinion briefly.
> > > >
> > > > - **The advantage of this study is the novelty and interestingness of idea.**
> > > > By modeling the discriminator based on NTK, the learning of the discriminator is omitted, and the learning of the generator is simplified.
> > > > GANs expect a discriminator to learn the intrinsic (low-dimensional) difference between true and pseudo examples from large examples (and memorize large examples), but GA-NTK does not.
> > > > I recognize that NTK-based discriminator is a non-parametric estimator like smoothing based on small (like 256) training examples.
> > > > Statisticians often say "generalization error = approximation error + estimation error + optimization error".
> > > > I think non-parametric NTK-based discriminator is worse than parametric GANs estimator in estimation error, but that NTK-based discriminator is better than GANs estimator in optimization error.
> > > > **Considered in this way, the proposal may be attractive.**
> > > >
> > > > - **Data generated by GA-TNK lack diversity and novelty, and hence GA-TNK is currently not practical, when $b$ is small.**
> > > > The relation $z^*\neq x$ shown by another reviewer does not completely guarantee diversity and novelty in practical meaning.
> > > > I believe that $b$ must be increased.
> > > > **However, this limitation should not lead to rejection of this paper.**
> > > > Increase of $b$ can be a future work.
> > > >
> > > > - **However, authors should describe this limitation in the paper more clearly.**
> > > > The authors describe the proposal too positively, and users of data augmentation (often unfamiliar with the theoretical arguments) will not be aware of this limitation.
> > > > **If the authors can fully reflect this request, I will give the paper a score of 8.**

---

> > > > > ### Author Response · Authors · 2022-12-09
> > > > > **Reply to 3rd Review by Reviewer LpjQ**
> > > > >
> > > > > We thank the reviewer for the insightful comments and summarization of the discussion. Following your advice, we will make the following changes in the next revision:
> > > > >
> > > > > 1. In Section 1, we will modify the last point of our claimed contributions by clearly stating that the batch-wise GA-NTK training technique does not completely solve the memory issue, which limits the values of $n$ and $b$.
> > > > >
> > > > > 2. In Section 11, we will conduct new experiments to study how the value of $b$ (or $n$) could negatively impact the diversity of synthesized points and how the novelty could be improved by increasing $b$. We will also summarize the results in the main paper in Section 4.1.
> > > > >
> > > > > 3. In Conclusion, we will review the current limitations of GA-NTK. In particular, we will emphasize that the current maximum value of $b$ (or $n$) that we achieved is still not large enough for complex tasks in the real world. Meanwhile, we will highlight the importance of scale-up techniques in the future.
> > > > >
> > > > > With the above changes, the readers of this paper could easily spot the bottlenecks of GA-NTK and are encouraged to investigate possible improvements. We hope these changes address the reviewer's concerns and are happy to follow any specific modifications suggested by the reviewer.

---

> > > > > > ### Comment · Reviewer_LpjQ · 2022-12-09
> > > > > > **4th Review to Reviewer LpjQ**
> > > > > >
> > > > > > I am satisfied with the modification.
> > > > > > I raise the score.
> > > > > >
> > > > > > ---
> > > > > >
> > > > > >  AC and other reviewers know this change.

---

> > > > > > > ### Author Response · Authors · 2022-12-09
> > > > > > > **Reply to 4th Review to Reviewer LpjQ**
> > > > > > >
> > > > > > > We appreciate your support! Thanks again for all your valuable suggestions that help make this paper better.

---

> > > > > > > ### Author Response · Authors · 2022-12-09
> > > > > > > **Score not changed**
> > > > > > >
> > > > > > > Dear reviewer,
> > > > > > >
> > > > > > > We notice that your overall Recommendation score has not changed. Could you please double check in case the AC and other reviewers miss your opinion?

---

### Author Response · Authors · 2022-11-19
**Reply to All Reviewers (Part 1 of 2)**

We thank all the reviewers for their valuable feedback and comments. We are delighted the reviewers find the proposed GA-NTK method interesting and novel (LpjQ, rAgg, ihk5, qHF8), most parts of the paper well-written (rAgg, ihk5, gHF8), and our experiments convincing and reproducible (LpjQ, rAgg, qHF8). Similarly, we appreciate you mention that GA-NTK has clear mathematical settings (rAgg, qHF8) and significance for future research (LpjQ).

We have submitted a revision following your suggestions. To favor an open discussion, we summarize the major concerns brought up by the reviewers, our responses, and the corresponding text/modifications in the revision below:

### LpjQ

> Is it necessary to solve Eq. (4) every time to generate pseudo examples?

GA-NTK can work with a generator network of finite width, as Eq. (6) shows. So it's not necessary.

> What value of t to use when training GA-NTK?

The $t$ (and $\lambda$) is a hyperparameter tuned before GA-NTK training. The complete training procedure is as follows:

1. Tune the hyperparameter $t$ (and $\lambda$) by using an auxiliary task asking the discriminator to separate real data from pure noises, as described in Algorithm 1;
2. Fix $t$ (and $\lambda$) and solve $\boldsymbol{Z}^{n}$ in Eq. (4) by ordinary gradient descent.

We add a paragraph right before Section 3.1 to describe the complete training procedure. More details of step 1 can be found in Section 8.3.

> What's the relation between $t$ in Eq. (3), index $s$ in Theorem 3.1, “Iterations” in Figure 5, and “Epochs” in Figure 3?

The $t$ is a hyperparameter controlling how well the discriminator should be “analytically trained” on ($\boldsymbol{X}^{n}\oplus\boldsymbol{Z}^{n}$, $\boldsymbol{1}^{n}\oplus\boldsymbol{0}^{n}$) in Eq. (4). The index $s$ is the number of gradient descent steps actually taken to solve $\boldsymbol{Z}^{n}$ in Eq. (4), which is the same as the “Iterations” in Figure 5 and the “Epochs” in Figure 3. We have changed the “Epochs” in Figure 3 to “Iterations” to make our presentation consistent.

### rAgg

> What is the time usage for training GA-NTK variants and how does it compare with training GANs?

We add Section 13 to discuss the convergence and training time. In summary, we found that GA-NTK variants require significantly fewer numbers of SGD iterations to converge than GANs. However, GA-NTK variants run fewer iterations per second because of the higher computation cost involved in back-propagating through the kernel matrix $\boldsymbol{K}$. In terms of wall-clock time, the batch-wise GA-CNTK (without a generator) is faster than GANs while the GA-CNTKg (with a generator) runs as fast as WGAN-GP.

### ihk5

> Statistical background of aligning $\mathcal{P}\_{\text{data}}$ and $\mathcal{P}\_{\text{gen}}$.

We add Section 6 to discuss this. It can be shown that minimizing the GA-NTK and GA-NTKg objectives (Eqs. (4) and (6)) amounts to minimizing the Pearson $\chi^{2}$-divergence between $\mathcal{P}\_{\text{data}}+\mathcal{P}\_{\text{gen}}$ and $2\mathcal{P}\_{\text{gen}}$, where $\mathcal{P}\_{\text{data}}$ is the distribution of real data and $\mathcal{P}\_{\text{gen}}$ is the distribution of generated points.

> How does minimizing Eq. (4) give fluctuation and diverse artificial data?

The diversity of our generated data not only comes from the randomness of an optimization algorithm (e.g., initialization of $\boldsymbol{Z}$ or splitting of $\boldsymbol{X}$ into batches) but also from the objective in Eq. (4) itself. To show this, we update **[Figure 3](https://drive.google.com/file/d/13ElCJ2ju36GnDRnnUd4TfQR_lyimB7Yr/view?usp=share_link)** to include the images generated at the later stage of training. The patterns of these images keep changing over training time, despite little change in the loss score. This can be seen more clearly with the **[animation](https://drive.google.com/file/d/1LYMNNyBQ-mBQ-iNJNxXObg7Mgd-JtX0g/view?usp=share_link)**. The reason is that, in Eq. (4), the $\boldsymbol{Z}$ is optimized for a *moving* target—any change of $\boldsymbol{Z}$ causes $\mathcal{D}$ to be “retrained” instantly.

To be continued...

---

> ### Author Response · Authors · 2022-11-19
> **Reply to All Reviewers (Part 2 of 2)**
>
> ### qHF8
>
> > Is the proposed discriminator trained (or not trained) during the overall training process?
>
> The discriminator is a closed form formula which is instantly “trained” once the training data $\boldsymbol{X}^{n}\oplus\boldsymbol{Z}^{n}$ change. It has no weights but a hyperparameter $\lambda$ that needs to be tuned before the training of $\boldsymbol{Z}^{n}$. The complete training procedure is as follows:
>
> 1. Tune the hyperparameter $\lambda$ by using an auxiliary task asking the discriminator to separate real data from pure noises, as described in Algorithm 1;
> 2. Fix $\lambda$ and solve $\boldsymbol{Z}^{n}$ in Eq. (4) by ordinary gradient descent.
>
> We add a paragraph right before Section 3.1 to describe the complete training procedure. More details of step 1 can be found in Section 8.3.
>
> > Comparison between the traditional neural network-based discriminator and kernel methods.
>
> This has been actively studied in the NTK literature [1-4]. In summary, NTK-GPs perform similarly to neural networks in many situations and sometimes even better on small-data tasks [3-4].
>
> [1] Novak et al., "Neural Tangents: Fast and Easy Infinite Neural Networks in Python," in Proc. of ICLR, 2019.
> [2] Lee et al., "Finite Versus Infinite Neural Networks: an Empirical Study," in Proc. of NeurIPS, 2020.
> [3] Arora et al., "Harnessing the Power of Infinitely Wide Deep Nets on Small-data Tasks," in Proc. of ICLR, 2020.
> [4] Geifman et al, "On the similarity between the laplace and neural tangent kernels," in Proc. of NeurIPS, 2020.
>
> > The proposed discriminator has a space complexity of $O(n^{2})$ where $n$ is the number of data points.
>
> This problem is rooted in NTKs, and many recent efforts [5-8] have been made to reduce their time/space complexity. Integrating these works into GA-NTK is an interesting direction to explore in the future. Meanwhile, we propose the batch-wise variant of GA-NTK, as described in Section 3.2, to addresses the scalability issue from the optimization perspective. In this variant, the $n$ represents the *batch size* instead of the size of an entire dataset. We show in Figure 8 that a batch size of 256 suffices to train a batch-wise GA-CNTK on larger CelebA and ImageNet datasets of 2048 and 1300 examples, respectively. We also show that one can train a generator $\mathcal{G}$ in GA-CNTKg with batches of size 256 and then, at inference time, use $\mathcal{G}$ to generate data regardless of the time/space limitations of NTKs.
>
> [5] Arora et al., "On Exact Computation with an Infinitely Wide Neural Net," in Proc. of NeurIPS, 2019.
> [6] Bietti et al., "On the Inductive Bias of Neural Tangent Kernels," in Proc. of NeurIPS, 2019.
> [7] Han et al., "Random Features for the Neural Tangent Kernel," CoRR abs/2104.01351, 2021.
> [8] Zandieh et al., "Scaling neural tangent kernels via sketching and random features," in Proc. of NeurIPS, 2021.
>
> > Mode collapse can be caused by other factors as well, such as the structure of generators.
>
> We totally agree with you and have modified our claim in Section 1 to that GA-NTK avoids only the mode collapse problem caused by the alternating SGD.
>
> > Can GA-CNTK learn high-level semantics?
>
> In Section 12, we study the “interpolated” images output by the generator $\mathcal{G}$ of GA-CNTKg taking equidistantly spaced $\boldsymbol{z}$'s along a segment in a two dimensional $\boldsymbol{z}$ space as the input. Looking at the results **[here](https://drive.google.com/file/d/1yXsW8zYuOcDqrrZU9iweCUtJ8luUBj2R/view?usp=share_link)** and **[there](https://drive.google.com/file/d/1vWkP1mjsOWWH1FWXYHNw7U1Nah8HdgMN/view?usp=share_link)**, we can see that the generated patterns transit smoothly across the 2D $\boldsymbol{z}$ space, and neighboring images share similar looks. Since the input $\boldsymbol{z}$'s are meaningless, the learned features encode high-level semantics.
>
> Pleas find our detailed responses in the comments to be given to individual reviewers. We hope the new revision addresses your concerns, and thank you again for making this paper better.

---

### Author Response · Authors · 2022-12-07
**Gentle nudge to reviewers**

Dear reviewers and AC,

The discussion period is coming to an end, and we would like to use this forum one last time to kindly ask the reviewers rAgg and LpjQ to voice their support for the paper once again.

We have received many constructive comments in the discussion period. However, it seems that the final decision of the paper will heavily rely on the engagement of the two positive reviewers. In this regard, and to facilitate their involvement in the discussion we would like to use this opportunity to make a summary of the discussion and updates to our work:

* We have compared the convergence speed and training time usage between GA-NTK and GAN variants as requested by reviewer rAgg.

* We have discussed the effect of $\lambda$ (controlling how well the discriminator $\mathcal{D}$ is analytically trained) and $s$ (the number of SGD steps to run to get generated points $\boldsymbol{Z}$) as demanded by reviewer LpjQ.

* We have conducted new experiments to show that as the number of training examples $n$ or the batch size $b$ increases (up to 512), GA-NTK can generate more novel artificial points as suggested by reviewer LpjQ.

* We have shown that minimizing the GA-NTK and GA-NTKg objectives (Eqs. (4) and (6)) amounts to minimizing the Pearson $\chi^{2}$-divergence between $\mathcal{P}\_{\text{data}}+\mathcal{P}\_{\text{gen}}$ and $2\mathcal{P}\_{\text{gen}}$ as demanded by reviewer ihk5.

* We have added two plots to show that the generator in GA-NTKg can learn high-level semantics of images and can map from the space of $\boldsymbol{z}$ to that of $\boldsymbol{x}$ smoothly as requested by reviewer qHF8.

*  We have discussed the correspondence between the performance of NTK-GPs and finite-width neural networks in the literature as requested by reviewer qHF8.

* The only remaining concern of reviewer ihk5 seems to be that, when $n=1$ and an RBF kernel is used, the global optimum $\boldsymbol{z}^{* }$ becomes $\boldsymbol{x}$ in the reviewer's analysis. Therefore, the reviewer concluded that the points generated by GA-NTK may lack diversity. However, we obtained opposite results following the reviewer's analysis:
    1. The spectral properties of the kernel matrix $\boldsymbol{K}$ shows that $\boldsymbol{z}^{* }$ needs not be $\boldsymbol{x}$. This result holds for any kernel, including the RBF kernel and NTK.
    2. The $\boldsymbol{z}=\boldsymbol{x}$ is not a feasible solution to Eq. (4) because otherwise it will lead to an ill-defined discriminator $\mathcal{D}$ giving $$\mathcal{D}(\boldsymbol{x})=1\quad\text{and}\quad\mathcal{D}(\boldsymbol{z})=\mathcal{D}(\boldsymbol{x})=0.$$
    3. When $\boldsymbol{z}^{*}\neq\boldsymbol{x}$, there can be an infinite number of optima minimizing Eq. (4). Furthermore, with NTK, these minima can look very different in the input (pixel) space thanks to the well-known non-linearity of NTK features [1].

    Although the reviewer did not provide feedback on our results, we would be totally open to discuss the loss properties in the camera-ready of our work and we hope that this will not be a reason for rejection of the paper.

We thank the reviewers and AC once again for the active conversation, and we hope that we can use these final days of the discussion period to address any remaining concerns.

[1] Schoenholz et al., "Deep Information Propagation," in Proc. of ICLR'17.

---

### Decision · Program_Chairs · 2023-01-20

**Decision:**

Reject

**Justification For Why Not Higher Score:**

Although this paper proposes an interesting idea, the weaknesses make it not reach acceptance at the present stage (see the meta-review).

**Justification For Why Not Lower Score:**

N/A

**Metareview: Summary, Strengths And Weaknesses:**

This paper proposes a novel NTK-based GAN named GA-NTK, which converts the bilevel optimization in training GANs to a single-level problem. GA-NTK can reduce computational costs and make the training of GANs more stable. Experiment results on multiple datasets demonstrate the superiority of GA-NTK.

The strengths of this paper contain: (a) This paper gives a clear description of the motivation. The proposed idea is insightful and interesting, which might inspire follow-up research in efficient GAN training. (b) The mathematical presentation is clear.  Overall, readers find that it is easy to follow this paper.

The main weaknesses of this paper include: (a) The proposed method is not well justified from a statistical perspective, in particular the statistical meaning of the solution. (b) Experiments are not convincing. Although the paper provides a series of empirical evaluations to support its claims, the reported results are found to lack diversity. The issue makes the current GA-NTK not practical. (c) More discussions on the method design are needed. For example, the complexity of the method is still worried. The causes of the mode collapse should be further discussed.

In the period of rebuttal, the authors actively communicate with reviewers to address their concerns. Reviewers acknowledge that partial concerns mainly caused by confusing descriptions are addressed. After rebuttal, however, some pivotal problems are not well handled. In particular, (1) there should be more convincing explanations of the method's effectiveness. (2) there should be more discussions on empirical results. (3) the contributions could be stated more clearly, e.g., by moderately discussing the limitations of the method.

The AC considers that the mentioned but unresolved issues are important before acceptance. Hence, a "reject" recommendation is made at this stage. However, AC and reviewers believe this work will be a very stronger submission after revision.